# Why Fine-grained Labels in Pretraining Benefit Generalization?

**Guan Zhe Hong**      *hong288@purdue.edu*
*Purdue University*

**Yin Cui**[*]      *yinc@nvidia.com*
*NVIDIA*

**Ariel Fuxman**      *afuxman@google.com*
*Google Research*

**Stanley H. Chan**      *stanchan@purdue.edu*
*Purdue University*

**Enming Luo**      *enming@google.com*
*Google Research*

**Reviewed on OpenReview:** *https://openreview.net/forum?id=FojAV72owK*

## Abstract

Recent studies show that pretraining a deep neural network with fine-grained labeled data, followed by fine-tuning on coarse-labeled data for downstream tasks, often yields better generalization than pretraining with coarse-labeled data. While there is ample empirical evidence supporting this, the theoretical justification remains an open problem. This paper addresses this gap by introducing a "hierarchical multi-view" structure to confine the input data distribution. Under this framework, we prove that: 1) coarse-grained pretraining only allows a neural network to learn the common features well, while 2) fine-grained pretraining helps the network learn the rare features in addition to the common ones, leading to improved accuracy on hard downstream test samples.

## 1 Introduction

We consider the theory of label granularity in deep learning. By label granularity, we mean a hierarchy of training labels specifying how detailed each label subclass needs to be (See Figure 1).

Having access to different granularity of labels offers us the freedom of training a classifier using a different level of precision. For example, instead of differentiating between `dogs` and `cats`, we can train a classifier to differentiate a `Poodle` dog and a `Persian` cat. The latter classification task is undoubtedly harder. However, recent studies found that if one uses fine-grained labels to *pre-train* a backbone, the pre-trained backbone will help the downstream neural networks generalize better (Chen et al., 2018). Vision transformers, for example, are well-known to require pretraining on large datasets with thousands of classes for effective downstream generalization (Dosovitskiy et al., 2021; He et al., 2016; Krizhevsky et al., 2012).

To convince readers who are less familiar with this particular training strategy, we conduct an experiment on ImageNet with details described in Appendix A.2 (we also include experiments on iNaturalist 2021 in Appendix A). Our experiment is limited in scale due to its high demand on the computing resources. Figure 2 shows an experiment of pre-training on ImageNet21k and fine-tuning the pre-trained network using ImageNet1k. The labels used in the ImageNet21k is based on WordNet Hierarchy. The downstream task

---

[*]Work done at Google Research.

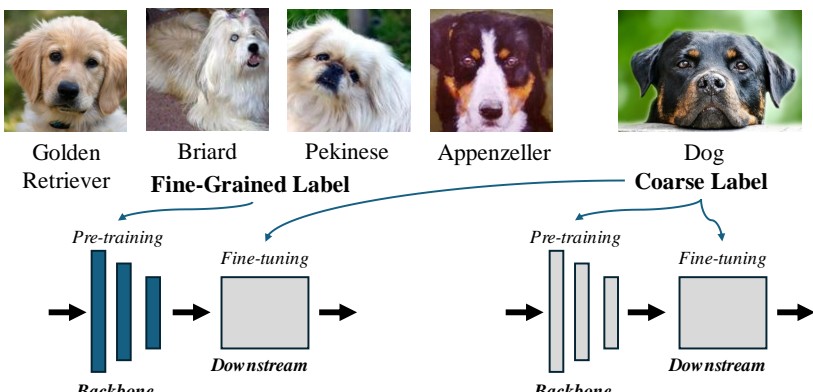

Figure 1: The goal of this paper is to provide a theoretical justification of why fine-grained labels in pre-training benefit generalization.

is ImageNet1k classification. The $x$-axis of this plot indicates the number of pre-training classes whereas the $y$-axis shows the validation accuracy for the ImageNet1k classification task. It is evident from the plot that as we increase the number of classes (hence a finer label granularity in pre-training), the downstream classification task's performance is improved.

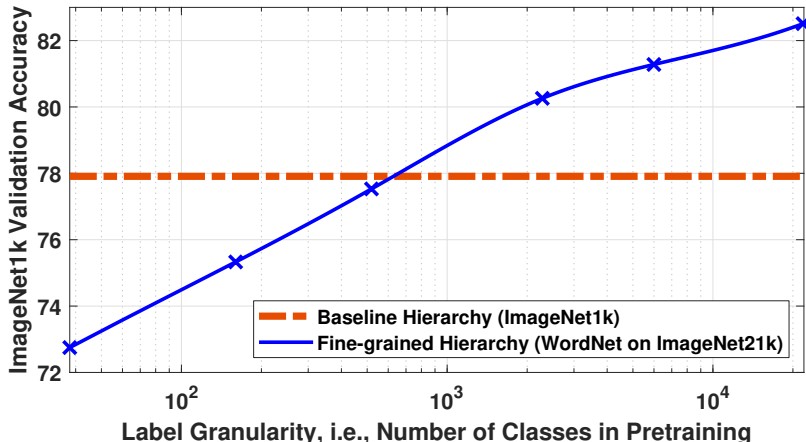

Figure 2: ImageNet21k→ImageNet1k transfer using a ViT-B/16 model. [Blue]: pretrained on the WordNet hierarchy of ImageNet21k, *finetuned* on ImageNet1k. [Red]: baseline, trained and evaluated on ImageNet1k.

## 1.1    Goal of this paper

The above experimental finding may sound familiar to practitioners who frequently train large models. In fact, experimental evidence on this subject is abundant (Mahajan et al., 2018; Singh et al., 2022; Yan et al., 2020; Shnarch et al., 2022; Juan et al., 2020; Yang et al., 2021; Chen et al., 2018; Ridnik et al., 2021; Son et al., 2023; Ngiam et al., 2018; Cui et al., 2018; 2019a). However, the theoretical explanation remains an open problem. Our goal in this paper is to provide a *theoretical* justification. The core question we ask is:

---
*Theoretical Question*

**Why** does pretraining at a high label granularity benefit generalization?

---

Certainly, this grand challenge can be impossible to answer in full because of the uncontrollable complexity of the practical situations. To say something concrete, we focus on a tractable (sub-)problem under a controlled setting:

- *Simple* scheme: We pretrain a backbone on a classification task and then finetune it for a target problem;

- Assume *negligible distribution shift* between the input distributions of the source and target datasets;

- The label functions for both datasets *align* well in terms of the features which they consider discriminative;

- The labels are error-free.

## 1.2 Main results and theoretical contributions

Our main result is based on analyzing a two-layer convolutional neural network with ReLU activation. We assume that the data distribution satisfies a certain *hierarchical multi-view* condition (to be discussed in Section 4.1). The optimization algorithm is stochastic gradient descent. Such problem settings are consistent with published works on this subject (Allen-Zhu & Li, 2023b; 2022; Shen et al., 2022b; Jelassi & Li, 2022). Our conclusions are as follows.

---
*Theoretical results*

1. *Coarse-grained* pretraining *only* allows the neural network to learn the common features well. Therefore, when testing, the test error on easy samples is $o(1)$ (i.e., small) whereas the error on hard samples is $\Omega(1)$ (i.e., large).
2. *Fine-grained* pretraining helps the network learn the rare features *in addition* to the common ones, thus improving its test error on *hard* samples. In particular, the test error rate on both easy and hard test samples are $o(1)$ (i.e., small).

---

To our knowledge, a precise characterization of the test error presented in this paper has never been reported in the literature. The key enablers of our theoretical finding are the concepts of *hierarchical multi-view* and *representation-label correspondence*. We summarize these two concepts below:

1. *Hierarchical multi-view*. To understand the label granularity problem, we argue that it is necessary for coarse and fine-grained classes to be distinguished by their corresponding input features. This is consistent with the *multi-view* data property pioneered by Allen-Zhu & Li (2023b). We call this a *hierarchical multi-view* structure. The hierarchical multi-view structure on the data makes us different from many other deep learning theory works that assume simple or no structure in the input data (Kawaguchi, 2016; Allen-Zhu & Li, 2023a; Ba et al., 2022; 2023; Damian et al., 2022; Kumar et al., 2023; Ju et al., 2021).
2. *Representation-label correspondence*. Representation learning aims to recognize features in the input data. As will be shown later in the paper, under the hierarchical multi-view data assumption, label complexity (i.e., how complex the labels are) during training influences the representation complexity (i.e., how many and what types of features are learnt), which further influences the model's generalization performance. Studying label granularity through understanding the neural network's feature-learning process is a departure from the literature which focuses on *feature selection* (Jacot et al., 2018; Ju et al., 2021; 2022; Pezeshki et al., 2021; Arora et al., 2019), i.e., selecting a subset of pre-determined features.

## 2 Related Work

### 2.1 Our theoretical setting compared to the literature

The subject of label granularity is immensely related to how to make a deep neural network (DNN) generalize better. In the existing literature, this is mostly explained through the lens of implicit regularization and bias towards simpler solutions to prevent overfitting even when DNNs are highly overparameterized (Lyu et al., 2021; Kalimeris et al., 2019; Ji & Telgarsky, 2019; De Palma et al., 2019; Huh et al., 2017). An alternative approach is the concept of shortcut learning which argues that deep networks can learn overly simple solutions. As such, deep networks achieve high training and testing accuracy on in-distribution data but generalize poorly to challenging downstream tasks (Geirhos et al., 2020; Shah et al., 2020; Pezeshki et al., 2021).

By examining these papers, we believe that Shah et al. (2020); Pezeshki et al. (2021) are the closest to ours because they demonstrate that DNNs perform shortcut learning and respond weakly to features that have a weak presence in the training data. However, our work departs from Shah et al. (2020); Pezeshki et al. (2021) in several key ways.

1. We focus on how the pretraining label space affects classification generalization, while Shah et al. (2020); Pezeshki et al. (2021) primarily focus on demonstrating that simplicity bias can be harmful to generalization.
2. The core theoretical tool used by Pezeshki et al. (2021) is the neural tangent kernel (NTK) model, which is unsuitable for analyzing the label granularity problem because the feature extractor of an NTK model barely changes after pretraining.
3. The theoretical setting in Shah et al. (2020) is limited because they use the hinge loss while we use a more standard exponential-tailed cross-entropy loss.
4. Our data distribution assumptions are more realistic, as they capture feature hierarchies in natural images, which has direct impact on the downstream generalization power of the pretrained model.

## 2.2 Our analytic tool compared to literature

Our theoretical analysis is inspired by a recent line of work by Allen-Zhu & Li (2022; 2023b); Shen et al. (2022b). These papers analyze the feature learning dynamics of neural networks by tracking how the hidden neurons of shallow nonlinear neural networks evolve to solve dictionary-learning-like problems. We adopt a *multi-view* approach to the data distribution which was first proposed in Allen-Zhu & Li (2023b). However, the learning problems we analyze and the results we aim to show are fundamentally different. As such, we derive the gradient descent dynamics of the neural network from scratch.

## 2.3 Consistency with existing empirical results

We stress that our theoretical findings are consistent with the reported empirical results in the literature, especially those that aim to improve classification accuracy by manipulating the pre-training label space (Mahajan et al., 2018; Singh et al., 2022; Yan et al., 2020; Shnarch et al., 2022; Juan et al., 2020; Yang et al., 2021; Chen et al., 2018; Ridnik et al., 2021; Son et al., 2023; Ngiam et al., 2018; Cui et al., 2018; 2019a). For example, Mahajan et al. (2018); Singh et al. (2022) use hashtags from Instagram as pretraining labels, Yan et al. (2020); Shnarch et al. (2022) apply clustering on the data first and then treat the cluster IDs as pretraining labels, Juan et al. (2020) use the queries from image search results, Yang et al. (2021) apply image transformations such as rotation to augment the label space, and Chen et al. (2018); Ridnik et al. (2021) include fine-grained manual hierarchies in their pretraining processes. Our results corroborate the utility of pretraining on fine-grained label space.

On the empirical end, there is also work focusing on exploiting the hierarchical structures present in (human-generated) label space to improve classification accuracy (Yan et al., 2015; Zhu & Bain, 2017; Goyal & Ghosh, 2020; Sun et al., 2017; Zelikman et al., 2022; Silla & Freitas, 2011; Shkodrani et al., 2021; Bilal et al., 2017; Goo et al., 2016). For example, Yan et al. (2015) adapt the network architecture to learn super-classes at each hierarchical level, Zhu & Bain (2017) add hierarchical losses in the hierarchical classification task, Goyal & Ghosh (2020) propose a hierarchical curriculum loss for curriculum learning. Our results do not directly validate these practices because we are more interested in understanding the influence of label granularity on model generalization.

# 3 Notations and Intuitions

## 3.1 Notations and training schemes

For a DNN-based classifier, given input image $\boldsymbol{X}$, we can write its (pre-logit) output for class $c$ as

$$\underbrace{F_c(\boldsymbol{X})}_{\substack{\text{pre-logit} \\ \text{output for class } c}} = \Big\langle \underbrace{\boldsymbol{a}_c}_{\substack{\text{linear} \\ \text{classifier}}}, \underbrace{\boldsymbol{h}}_{\substack{\text{backbone} \\ \text{network}}} (\underbrace{\boldsymbol{\Theta}}_{\substack{\text{network} \\ \text{parameter}}}; \boldsymbol{X}) \Big\rangle, \tag{1}$$

where $\boldsymbol{a}_c$ is the linear classifier for class $c$, $\boldsymbol{h}(\boldsymbol{\Theta}; \cdot)$ is the network backbone with parameter $\boldsymbol{\Theta}$.

Referring to Figure 1, label granularity concerns about two datasets: $\mathcal{X}^{\text{src}}$ for the source (typically fine-grained) and $\mathcal{X}^{\text{tgt}}$ for the target (typically coarse-grained). The corresponding labels are $\mathcal{Y}^{\text{src}}$ and $\mathcal{Y}^{\text{tgt}}$, respectively. A dataset can be represented as $\mathcal{D} = (\mathcal{X}, \mathcal{Y})$. For instance, the source training dataset is $\mathcal{D}_{\text{train}}^{\text{src}} = (\mathcal{X}_{\text{train}}^{\text{src}}, \mathcal{Y}_{\text{train}}^{\text{src}})$.

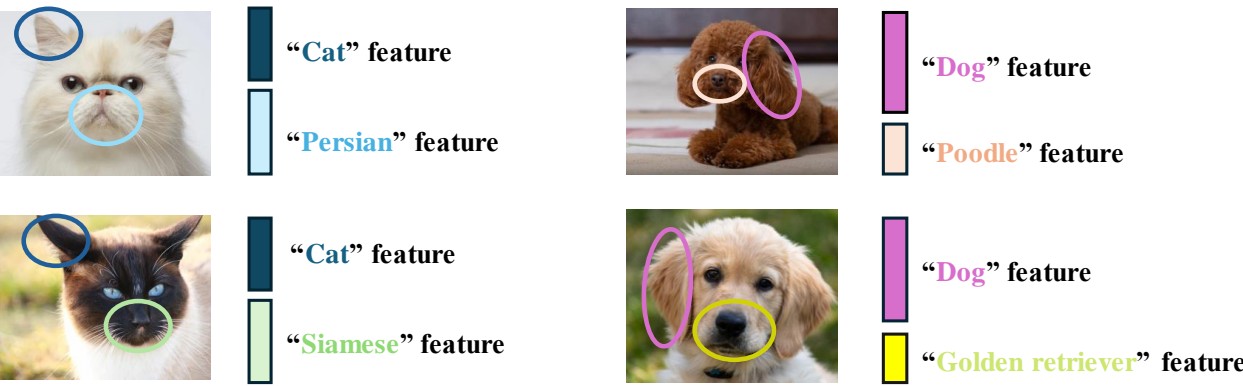

Figure 3: A simplified symbolic representation of the cat versus dog problem.

The relevant training and testing datasets are denoted as $\mathcal{D}_{\text{train}}^{\text{src}}, \mathcal{D}_{\text{train}}^{\text{tgt}}, \mathcal{D}_{\text{test}}^{\text{tgt}}$. Finally, the granularity of a label set is denoted as $\mathcal{G}(\mathcal{Y})$, which represents the total number of classes.

The two learning methodologies of interest are as follows.

1. Baseline: Train $F_c(\cdot)$ using $\mathcal{D}_{\text{train}}^{\text{tgt}}$. Test $F_c(\cdot)$ using $\mathcal{D}_{\text{test}}^{\text{tgt}}$.
2. Fine-to-coarse: Train $F_c(\cdot)$ using $\mathcal{D}_{\text{train}}^{\text{src}}$. This gives us the pretrained feature extractor $\boldsymbol{h}(\boldsymbol{\Theta}_{\text{train}}^{\text{src}}; \cdot)$. Then finetune $F_c(\cdot)$ using $\mathcal{D}_{\text{train}}^{\text{tgt}}$. Test the resulting $F_c(\cdot)$ using $\mathcal{D}_{\text{test}}^{\text{tgt}}$.

### 3.2 Intuition: why higher granularity improves generalization

Consider the following toy example. There are two classes: cat and dog. Our goal is to build a binary classifier. Let's discuss how the two training schemes would work, with an illustration shown in Figure 3.

1. **Baseline**. The baseline method tries to identify the common features that can distinguish most of the cats from dogs, for instance, the shape of the animal's ear as shown in Figure 3. These features are often the most noticeable ones because they appear the most frequently. Of course, there are hard samples, e.g., a close-up shot of a cat's fur. They pose limited influence during training because they are relatively rare in natural images.
2. **Fine-to-coarse**. With fine-grained labels, each subclass has its own unique visual features that are only dominant within that subclass. However, fine-grained features are not as common in the dataset, hence making them more difficult to be noticed. Therefore, if we only present the coarse labels in the pre-training stage, the learner is allowed to take shortcuts by learning only the common features to achieve low training loss. One strategy to force the learner to learn the rarer features is to explicitly label the fine-grained classes. This means that within each fine-grained class, the fine-grained features become as easy to notice as the common features. As a result, even if common features are weakly present or missing in a hard test sample, the network can still be reasonably robust to distracting irrelevant patterns due to its ability to recognize (some of) the finer-grained features.

## 4 Problem Formulation

Our first theoretical contribution is a new data model, the hierarchical multi-view model. This model consists of four definitions. Compared to existing theories studying feature learning of neural networks in the literature (Allen-Zhu & Li, 2023b; 2022; Shen et al., 2022b; Jelassi & Li, 2022), these four definitions are better formulated to the label granularity problem. For the sake of brevity, we present the core concepts of our data model here, and delay its full specification to Appendix B. Following data model specifications, we also discuss characteristics of the learner, a two-layer nonlinear convolutional neural network.

### 4.1 New data model: hierarchical multi-view

We consider the setting where an input sample $\boldsymbol{X} \in \mathbb{R}^{dP}$ consists of $P$ patches $\boldsymbol{x}_1, \boldsymbol{x}_2, ..., \boldsymbol{x}_P$ with $\boldsymbol{x}_p \in \mathbb{R}^d$, where $d$ is sufficiently large, and all our asymptotic statements are made with respect to $d$.

For analytic tractability, we consider two levels of label hierarchy. The root of this hierarchy has two superclasses $+1$ and $-1$. The superclass $+1$ has $k_+$ subclasses. We denote these $k_+$ subclasses as $(+1, c_1), \dots, (+1, c_{k_+})$. We can do the same for the superclass $-1$ which has $k_-$ subclasses. Each subclass has two types of features: the common features and the fine-grained features. The two types of features are sufficiently different in the sense they have zero correlation and equal magnitude. This leads to the following definition.

**Definition 4.1** (Features). We define **features** as elements of a fixed orthonormal dictionary $\mathcal{V} = \{\boldsymbol{v}_i\}_{i=1}^d \subset \mathbb{R}^d$. The common and fine-grained features are

- Common feature: $\boldsymbol{v}_+ \in \mathcal{V}$ and $\boldsymbol{v}_- \in \mathcal{V}$
- Fine-grained feature of subclass $c$: $\boldsymbol{v}_{+,c}$ and $\boldsymbol{v}_{-,c} \in \mathcal{V}$

The usage of an orthonormal dictionary is again a choice of our model. We choose so because it is more tractable. With features defined, we can now specify patches in an input sample.

**Definition 4.2** (Input patches). We define three types of patches for $y \in \{+, -\}$:

- (Common-feature patches) are defined as $\boldsymbol{x}_p = \alpha_p \boldsymbol{v}_y + \boldsymbol{\zeta}_p$, where $\alpha_p \approx 1$, and $\boldsymbol{\zeta}_p \sim \mathcal{N}(\boldsymbol{0}, \sigma_\zeta^2 \boldsymbol{I}_d)$.
- (Subclass-feature patches) are defined as $\boldsymbol{x}_p = \alpha_p \boldsymbol{v}_{y,c} + \boldsymbol{\zeta}_p$, where $\alpha_p \approx 1$, and $\boldsymbol{\zeta}_p \sim \mathcal{N}(\boldsymbol{0}, \sigma_\zeta^2 \boldsymbol{I}_d)$.
- (Noise patches) are defined as $\boldsymbol{x}_p = \boldsymbol{\zeta}_p$.

Within an input sample $\boldsymbol{X} = (\boldsymbol{x}_1, \boldsymbol{x}_2, ..., \boldsymbol{x}_p)$, there are approximately $s^*$ common-feature patches and $s^*$ subclass-feature patches, the rest are all noise patches. Moreover, within a sample, the choice of $y$ has to be consistent across the feature patches. Lastly, the positions of the features patches are random.

These definitions of the input patches are illustrated in Figure 4.

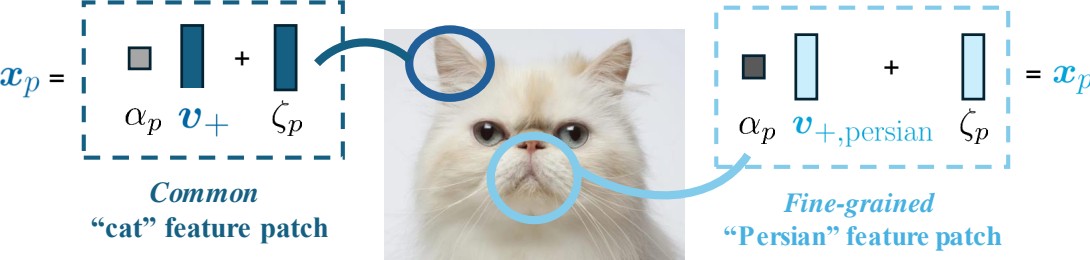

Figure 4: Illustration of features and patches.

Some comments: An **easy sample** is generated according to Definition 4.2. A **hard sample** is generated in the same way as easy samples, except the common-feature patches are replaced by noise patches, and we replace a small number of noise patches by "feature-noise" patches, which are of the form $\boldsymbol{x}_p = \alpha_p^\dagger \boldsymbol{v}_- + \boldsymbol{\zeta}_p$, where $\alpha_p^\dagger \in o(1)$, and set one of the noise patches to $\boldsymbol{\zeta}^* \sim \mathcal{N}(\boldsymbol{0}, \sigma_{\zeta^*}^2 \boldsymbol{I}_d)$ with $\sigma_{\zeta^*} \gg \sigma_\zeta$; these patches serve the role of "distracting patterns".

**Definition 4.3** (Source dataset's label mapping). We say a sample $\boldsymbol{X}$ belongs to the $+1$ superclass if any one of its common- or subclass-feature patches contains $\boldsymbol{v}_+$ or $\boldsymbol{v}_{+,c}$ for any $c \in [k_+]$. It belongs to the $(+, c)$ subclass if any one of its subclass-feature patches contains $\boldsymbol{v}_{+,c}$.

**Definition 4.4** (Source training set). We assume the input samples of the source training set as $\mathcal{X}_{\text{train}}^{\text{src}}$ are generated as in Definition 4.2; the corresponding labels are generated following Definition 4.3. Overall, we denote the source training dataset $\mathcal{D}_{\text{train}}^{\text{src}}$.

**Relation to multi-view**. Our data model is inspired by the multi-view concept first proposed in Allen-Zhu & Li (2023b), as we (1) use an orthonormal dictionary to define the features, (2) define an input consisting

of many disjoint high-dimensional patches, and (3) assume the existence of *multiple* discriminative features per class. The reason why the original multi-view property is insufficient for our problem is that it does not consider any label hierarchy nor its link to the input structure. We resolve this issue by following our intuition that classes at different hierarchy levels should be distinguished by their corresponding features: this naturally defines a feature hierarchy, with an exact correspondence with the label hierarchy.

**Target dataset**. To ensure that baseline and fine-grained training have no unfair advantage over each other, we post a set of new characterizations on the *target* dataset:

1. The input samples in the target dataset is generated according to Definition 4.2.
2. The true label function is identical across the source and target datasets.
3. Since we are studying the "fine-to-coarse" transfer direction, the target problem's label space is the *root* of the hierarchy, meaning that any element of $\mathcal{Y}_{\text{train}}^{\text{tgt}}$ or $\mathcal{Y}_{\text{test}}^{\text{tgt}}$ must belong to the label space $\{+1, -1\}$.

Therefore, in our setting, only $\mathcal{Y}^{\text{src}}$ and $\mathcal{Y}^{\text{tgt}}$ can differ (in distribution) due to different choices in the label granularity level. In this idealized setting, we have essentially made baseline training and coarse-grained pretraining the *same* procedure. Therefore, an equally valid way to view our theory's setting is to consider $\mathcal{D}_{\text{train}}^{\text{tgt}}$ the same as $\mathcal{D}_{\text{train}}^{\text{src}}$ except with coarse-grained labels. In other words, we pretrain the network on two versions of the source dataset $\mathcal{D}_{\text{train}}^{\text{src,coarse}}$ and $\mathcal{D}_{\text{train}}^{\text{src,fine}}$, and then compare the two models on $\mathcal{D}_{\text{test}}^{\text{tgt}}$ (which has coarse-grained labels).

### 4.2 Characteristics about the learner

Our model about the learner is consistent with Allen-Zhu & Li (2023b; 2022); Shen et al. (2022b). The learner is a two-layer average-pooling convolutional ReLU network:

$$F_c(\boldsymbol{X}) = \sum_{r=1}^{m} a_{c,r} \sum_{p=1}^{P} \sigma(\langle \boldsymbol{w}_{c,r}, \boldsymbol{x}_p \rangle + b_{c,r}), \tag{2}$$

where $m$ is a low-degree polynomial in $d$ and denotes the width of the network, $\sigma(\cdot) = \max(0, \cdot)$ is the ReLU nonlinearity, and $c$ denotes the class. We perform a *random initialization* of $\boldsymbol{w}_{c,r}^{(0)} \sim \mathcal{N}(\boldsymbol{0}, \sigma_0^2 \boldsymbol{I}_d)$ with $\sigma_0^2 = 1/\text{poly}(d)$; we set $b_{c,r}^{(0)} = -\Theta\left(\sigma_0 \sqrt{\ln(d)}\right)$ and manually tune it, similar to Allen-Zhu & Li (2022). Cross-entropy is the training loss for both baseline and transfer training. To simplify analysis and to focus solely on the learning of the feature extractor, we freeze $a_{c,r} = 1$ during all baseline and transfer training phases, and we use the fine-grained model for binary classification as follows: $\widehat{F}_+(\boldsymbol{X}) = \max_{c \in [k_+]} F_{+,c}(\boldsymbol{X})$, $\widehat{F}_-(\boldsymbol{X}) = \max_{c \in [k_-]} F_{-,c}(\boldsymbol{X})$. See Appendix B.2 and the beginning of Appendix G for details of learner characteristics and training algorithm.

## 5 Theoretical results and proof strategy

Our second theoretical contribution lies in establishing a correspondence between the *complexity of the labels* and *complexity of the network's representations*. Under the assumption of the hierarchical multi-view data structure, the following are true:

1. If trained with coarse-grained labels (i.e. overly simple labels), the network only learns the common features well, so its representations of the data is overly simple;
2. In contrast, training with fine-grained labels helps the network learn the fine-grained features well in addition to the common ones, so its representation of the data is more complex.

The difference in representation complexity leads to the difference in the network's downstream test accuracy.

### 5.1 Main results

**Theorem 5.1** (Coarse-label training: baseline)**.** *(Summary). Let the number of subclasses be lower-bounded: $k_y \geq polylog(d)$. With high probability, with proper choice of step size, there exists a time $T^* \in poly(d)$ such*

*that for any $T \in [T^*, poly(d)]$, the training loss is upper bounded according to*

$$\mathcal{L}(F^{(T)}) \leq o(1) \tag{3}$$

*Moreover, for an **easy** test sample $(\boldsymbol{X}_{easy}, y)$, the probability of making a classification mistake is small:*

$$\mathbb{P}\left[F_y^{(T)}(\boldsymbol{X}_{easy}) \leq F_{y'}^{(T)}(\boldsymbol{X}_{easy})\right] \leq o(1), \quad for\ y' \neq y. \tag{4}$$

*However, for all $t \in [0, poly(d)]$, given a **hard** test sample $(\boldsymbol{X}_{hard}, y)$, the probability of making a classification mistake is large:*

$$\mathbb{P}\left[F_y^{(t)}(\boldsymbol{X}_{hard}) \leq F_{y'}^{(t)}(\boldsymbol{X}_{hard})\right] \geq \Omega(1), \quad for\ y' \neq y. \tag{5}$$

This theorem essentially says that, with a mild lower bound on the number of fine-grained classes, if we only train on the *easy* samples with *coarse* labels, it is virtually impossible for the network to learn the fine-grained features even if we give it as much practically reachable amount of time and training samples as possible. Consequently, the network would perform poorly on the hard downstream test samples: if the sample is missing the *common* features, then the network can be easily misled by the noise present in the sample. To see the full setup and statement of this theorem, please see Appendix B and E. Its proof spans Appendix C to E.

**Theorem 5.2** (Fine-grained-label training)**.** *(Summary). Assume the same setting as in Theorem 5.1, except let the labels be fine-grained and $k_y \leq d^{0.4}$ (number of subclasses not pathologically large; see Section 7 for its discussion). Within $poly(d)$ time, the probability of making a classification mistake is small:*

$$\mathbb{P}\left[\widehat{F}_y^{(T)}(\boldsymbol{X}) \leq \widehat{F}_{y'}^{(T)}(\boldsymbol{X})\right] \leq o(1)\ for\ y' \neq y, \tag{6}$$

*on the target binary problem on both **easy** and **hard** test samples.*

The full version of this result is presented in Appendix G.4, and its proof in Appendix G. After fine-grained pretraining, the network's feature extractor gains a strong response to the fine-grained features, therefore its accuracy on the downstream hard test samples increases significantly.

*Remark.* One concern about the above theorems is that the neural networks are trained only on easy samples. As noted in Sections 1 and 3.2, *easy* samples should make up the *majority* of the training and testing samples. Pretraining at higher label granularities only improves network performance on *rare* samples. Our theoretical result presents the feature-learning bias of a neural network in an exaggerated fashion. Therefore, it is natural to start with the case of no hard training samples. In reality, even if a small portion of hard training samples is present, finite-sized training datasets can have many flaws that can cause the network to overfit severely before learning the fine-grained features, especially since rarer features are learnt more slowly and corrupted by greater amount of noise. We leave these deeper considerations for future theoretical work.

## 5.2 Proof strategy: representation-label correspondence

The key idea of the proof is to establish a correspondence between the *complexity of the labels* and *complexity of the network's representations*. We show that when trained on coarse-grained labels (i.e. overly simple labels), the network only learns the common features well, so its representations of the data is overly simple. In contrast, training with fine-grained labels helps the network learn the fine-grained features well in addition to the common ones, so its representations are more complex.

We first sketch the proof of **baseline training** which uses *coarse-grained* labels.

**Feature detector neurons**. We show that, at initialization, with high probability, for every feature $\boldsymbol{v} \in \mathcal{V}$, there exists a small group of "lucky" neurons, denoted $S_y^{*(0)}(\boldsymbol{v})$ (with $y$ indicating the superclass), that only activate on $\boldsymbol{v}$-dominated feature patches. We prove that if $\boldsymbol{v}$ is a feature of class $y$, then with high probability,

the lucky neurons will remain activated on $\boldsymbol{v}$-dominated patches throughout training, and dominate the feature extractor's response to the feature $\boldsymbol{v}$. In particular, given any $\boldsymbol{v}$-dominated patch $\boldsymbol{x}_p = \alpha_p \boldsymbol{v} + \boldsymbol{\zeta}_p$,

$$\underbrace{\sum_{r=1}^{m} \sigma\left(\left\langle \boldsymbol{w}_{y,r}^{(t)}, \boldsymbol{x}_p \right\rangle + b_{y,r}^{(t)}\right)}_{\substack{\text{network representation of} \\ \boldsymbol{v}\text{-dominated patch } \boldsymbol{x}_p}} \approx \underbrace{\sum_{r \in S_y^{*(0)}(\boldsymbol{v})}^{m} \sigma\left(\left\langle \boldsymbol{w}_{y,r}^{(t)}, \boldsymbol{x}_p \right\rangle + b_{y,r}^{(t)}\right)}_{\substack{\text{detector neurons'} \\ \text{response to } \boldsymbol{v}\text{-dominated patch } \boldsymbol{x}_p}}, \quad t \in [0, \operatorname{poly}(d)]. \tag{7}$$

Therefore, we call neurons in $S_y^{*(0)}(\boldsymbol{v})$ the *detector neurons* of feature $\boldsymbol{v}$.

The significance of equation 7 is that, we may now argue about the *network's representation of the input data* solely based on the *behavior of the feature detector neurons.*

**Impartial representation at initalization**. At initialization, the feature extractor's response to common and fine-grained features are *very close*. The reason is that, $\left| S_y^{*(0)}(\boldsymbol{v}) \right| \approx \left| S_{y'}^{*(0)}(\boldsymbol{v}') \right|$ for all superclasses $y, y'$ and features $\boldsymbol{v}, \boldsymbol{v}'$, and they all have a similar magnitude of activation strength. Written explicitly, given any common-feature patch $\boldsymbol{x}_{\mathrm{com}} = \alpha \boldsymbol{v}_y + \boldsymbol{\zeta}$ and subclass-feature patch $\boldsymbol{x}_{\mathrm{sub}} = \alpha' \boldsymbol{v}_{y,c} + \boldsymbol{\zeta}'$ (from the training or testing distribution), with high probability,

$$\underbrace{\sum_{r \in S_y^{*(0)}(\boldsymbol{v}_y)}^{m} \sigma\left(\left\langle \boldsymbol{w}_{y,r}^{(0)}, \alpha \boldsymbol{v}_y + \boldsymbol{\zeta} \right\rangle + b_{y,r}^{(0)}\right)}_{\substack{\text{network representation of} \\ \text{common-feature patch } \boldsymbol{x}_{\mathrm{com}} \text{ at } t=0}} \approx \underbrace{\sum_{r \in S_y^{*(0)}(\boldsymbol{v}_{y,c})}^{m} \sigma\left(\left\langle \boldsymbol{w}_{y,r}^{(0)}, \alpha' \boldsymbol{v}_{y,c} + \boldsymbol{\zeta}' \right\rangle + b_{y,r}^{(0)}\right)}_{\substack{\text{network representation of} \\ \text{subclass-feature patch } \boldsymbol{x}_{\mathrm{sub}} \text{ at } t=0}} \tag{8}$$

So what happened during training which caused a strong *imbalance* of representation of the common and fine-grained features in the end? The answer below is the core of the proof.

***Overly simple labels $\Longrightarrow$ overly simple representations***. The imbalance of growth is a result of the subclass-feature patches occurring with less frequency in the training set than the common-feature patches. Recall that the number of subclasses is $k_y$: for any subclass $(y, c)$, subclass-feature patches dominated by $\boldsymbol{v}_{y,c}$ are about $k_y$ times *rarer* than the common feature patches. This has a direct impact on the growth speed of the common and fine-grained detector neurons: for any neuron $r_{\mathrm{com}} \in S_y^{*(0)}(\boldsymbol{v}_y)$ and any $r_{\mathrm{fine}} \in S_y^{*(0)}(\boldsymbol{v}_{y,c})$, $\langle \Delta \boldsymbol{w}_{y,r_{\mathrm{com}}}^{(t)}, \boldsymbol{v}_y \rangle \approx \Theta\left(k_y\right) \times \langle \Delta \boldsymbol{w}_{y,r_{\mathrm{fine}}}^{(t)}, \boldsymbol{v}_{y,c} \rangle.$

With careful arguments on the influence of noise and bias on the activation values, we can show that, for $t$ *sufficiently large*, the fine-grained detector neurons are about $\Theta(k_y)$ times *weaker* in strength:

$$\underbrace{\sum_{r \in S_y^{*(0)}(\boldsymbol{v}_y)}^{m} \sigma\left(\left\langle \boldsymbol{w}_{y,r}^{(t)}, \alpha \boldsymbol{v}_y + \boldsymbol{\zeta} \right\rangle + b_{y,r}^{(t)}\right)}_{\substack{\text{network representation of} \\ \text{common-feature patch } \boldsymbol{x}_{\mathrm{com}} \text{ at } \textbf{large } t}} \approx \Theta\left(k_y\right) \times \underbrace{\sum_{r \in S_y^{*(0)}(\boldsymbol{v}_{y,c})}^{m} \sigma\left(\left\langle \boldsymbol{w}_{y,r}^{(t)}, \alpha' \boldsymbol{v}_{y,c} + \boldsymbol{\zeta}' \right\rangle + b_{y,r}^{(t)}\right)}_{\substack{\text{network representation of} \\ \text{subclass-feature patch } \boldsymbol{x}_{\mathrm{sub}} \text{ at } \textbf{large } t}} \tag{9}$$

Furthermore, we prove that, due to the exponential tail of cross-entropy, by the end of training,

$$\sum_{r \in S_y^{*(0)}(\boldsymbol{v}_y)}^{m} \sigma\left(\left\langle \Delta \boldsymbol{w}_{y,r}^{(t)}, \alpha \boldsymbol{v}_y + \boldsymbol{\zeta} \right\rangle + b_{y,r}^{(t)}\right) = \Theta\left(\log(d)\right) \tag{10}$$

which causes the representation of subclass-feature patches to be *vanishing* in strength:

$$\sum_{r \in S_y^{*(0)}(\boldsymbol{v}_{y,c})}^{m} \sigma\left(\left\langle \boldsymbol{w}_{y,r}^{(t)}, \alpha' \boldsymbol{v}_{y,c} + \boldsymbol{\zeta}' \right\rangle + b_{y,r}^{(t)}\right) \le O\left(\frac{\log(d)}{k_y}\right) < o(1), \quad t \le \operatorname{poly}(d). \tag{11}$$

In other words, the neural network almost cannot detect subclass features by the end of baseline training. Therefore, even though it can classify the easy test samples correctly since it learned the common features

well, it simply cannot classify the hard ones, which requires the model to solely rely on subclass-feature patches for inference.

**Fine-grained** training alleviates this issue.

***Complex labels $\implies$ complex representations***. The proof of fine-grained training proceeds in a very similar fashion as the case of coarse-grained training. The main difference lies in the gradient updates. During training, for any neuron $r_{\text{com}} \in S_{(y,c)}^{*(0)}(\boldsymbol{v}_y)$ and any $r_{\text{fine}} \in S_{(y,c)}^{*(0)}(\boldsymbol{v}_{y,c})$,

$$\left\langle \Delta \boldsymbol{w}_{(y,c),r_{\text{com}}}^{(t)}, \boldsymbol{v}_y \right\rangle \approx \left\langle \Delta \boldsymbol{w}_{(y,c),r_{\text{fine}}}^{(t)}, \boldsymbol{v}_{y,c} \right\rangle. \tag{12}$$

In other words, the common and fine-grained detector neurons for each subclass grow at similar speeds now, because the common- and subclass-feature patches occur with similar frequency in each subclass. Again with careful analysis of how the noise and bias influence the activation values, we arrive at

$$\underbrace{\sum_{r \in S_{(y,c)}^{*(0)}(\boldsymbol{v}_y)}^{m} \sigma\left(\left\langle \boldsymbol{w}_{(y,c),r}^{(t)}, \alpha \boldsymbol{v}_y + \boldsymbol{\zeta}\right\rangle + b_{(y,c),r}^{(t)}\right)}_{\substack{\text{network representation of} \\ \text{common-feature patch } \boldsymbol{x}_{\text{com}}, \text{ end of training} \\ \geq \Omega(1)}} \approx \underbrace{\sum_{r \in S_{(y,c)}^{*(0)}(\boldsymbol{v}_{y,c})}^{m} \sigma\left(\left\langle \boldsymbol{w}_{(y,c),r}^{(t)}, \alpha' \boldsymbol{v}_{y,c} + \boldsymbol{\zeta}'\right\rangle + b_{(y,c),r}^{(t)}\right)}_{\substack{\text{network representation of} \\ \text{subclass-feature patch } \boldsymbol{x}_{\text{sub}}, \text{ end of training} \\ \geq \Omega(1)}} \tag{13}$$

Therefore, both the common and fine-grained features are learnt well. It follows that the model can correctly utilize the common- and subclass-feature patches in the input, so it can classify easy and hard test samples with high accuracy.

## 6 Empirical Results

Building on our theoretical analysis in an idealized setting, this section discusses conditions on the source and target label functions that we observed to be important for fine-grained pretraining to work *in practice*, while remaining in the controlled setting described in Section 1 for the sake of tractability. We present the core experimental results obtained on ImageNet21k and iNaturalist 2021 in the main text, and leave the experimental details and ablation studies to Appendix A.

### 6.1 ImageNet21k→ImageNet1k transfer experiment

This subsection provides more details about the experiment shown in Figure 2. Specifically, we show that the common practice of pretraining on ImageNet21k using leaf labels is indeed better than pretraining at lower granularities in the manual hierarchy.

**Hierarchy definition**. The label hierarchy in ImageNet21k is based on WordNet Miller (1995); Deng et al. (2009). To define fine-grained labels, we first define the leaf labels of the dataset as Hierarchy level 0. For each image, we trace the path from the leaf label to the root using the WordNet hierarchy. We then set the $k$-th synset (or the root synset, if it is higher in the hierarchy) as the level-$k$ label of this image. This procedure also applies to the multi-label samples. This is how we generate the hierarchies shown in Figure 2.

**Network choice and training**. For this dataset, we use the more recent Vision Transformer ViT-B/16 Dosovitskiy et al. (2021). Our pretraining pipeline is almost identical to the one in Dosovitskiy et al. (2021). For fine-tuning, we experimented with several strategies and report only the best results in the main text; the finer details are discussed in Appendix A.1.2 and A.2. To ensure a fair comparison, we also used these strategies to find the best baseline result by using $\mathcal{D}_{\text{train}}^{\text{tgt}}$ for pretraining.

### 6.2 Transfer experiment on iNaturalist 2021

We conduct a systematic study of the transfer method *within* the label hierarchies of iNaturalist 2021 (Horn & macaodha, 2021). This dataset is well-suited for our analysis because it has a manually defined label

hierarchy that is based on the biological traits of the creatures in the images. Additionally, the large sample size of this dataset reduces the likelihood of sample-starved pretraining on reasonably fine-grained hierarchy levels.

Our experiments on this dataset again demonstrate that, as long as the finer-grained labels contain little noise, are well-aligned with the target label space, and sample count per subclass is not too limited, then we observe improvement in the model's generalization performance. However, we also show negative results outside of the aforementioned "nice regime": when sample count per sub-class is limited, or the fine-grained labels are *noisy*, or potentially *misaligned* with the target label space's, finer-grained labels do not necessarily improve generalization significantly.

**Relevant datasets**. We perform transfer experiments within iNaturalist2021. More specifically, we set $\mathcal{X}^{\mathrm{src}}_{\mathrm{train}}$ and $\mathcal{X}^{\mathrm{tgt}}_{\mathrm{train}}$ both equal to the training split of the input samples in iNaturalist2021, and set $\mathcal{X}^{\mathrm{tgt}}_{\mathrm{train}}$ to the testing split of the input samples in iNaturalist2021. To focus on the "fine-to-coarse" transfer setting, the *target problem* is to classify the root level of the manual hierarchy, which contains 11 superclasses. To generate a greater gap between the performance of different hierarchies and to shorten training time, we use the mini version of the training set in all our experiments.

**Alternative hierarchies generation**. To better understand the transfer method's operating regime, we experiment with different ways of generating the fine-grained labels for pretraining: we perform kMeans clustering on the ViT-L/14-based CLIP embedding Radford et al. (2021); Dehghani et al. (2022) of every sample in the training set and use the cluster IDs as pretraining class labels. We carry out this experiment in two ways. The green curve in Figure 5 comes from performing kMeans clustering on the embedding of each superclass *separately*, while the purple one's cluster IDs are from performing kMeans on the *whole dataset*. The former way preserves the implicit hierarchy of the superclasses in the cluster IDs: samples from superclass $k$ cannot possibly share a cluster ID with samples belonging to superclass $k' \neq k$. Therefore, its label function is forced to align better with that of the 11 superclasses than the purple curve's. We also assign random class IDs to samples.

**Network choice and training**. We experiment with ResNet 34 and 50 on this dataset. For pretraining on $\mathcal{D}^{\mathrm{src}}_{\mathrm{train}}$ with fine-grained labels, we adopt a standard 90-epoch large-batch-size training procedure commonly used on ImageNet He et al. (2016); Goyal et al. (2017). Then we finetune the network for 90 epochs and test it on the 11-superclass $\mathcal{D}^{\mathrm{tgt}}_{\mathrm{train}}$ and $\mathcal{D}^{\mathrm{tgt}}_{\mathrm{test}}$, respectively, using the pretrained backbone $h(\boldsymbol{\Theta}_{\mathrm{src}}; \cdot)$. To ensure a fair comparison, we trained the baseline model using exactly the same training pipeline, except that the pretraining stage uses $\mathcal{D}^{\mathrm{tgt}}_{\mathrm{train}}$. We observed that this "retraining" baseline consistently outperformed the naive one-pass 90-epoch training baseline on this dataset. Due to space limitations, we leave the results of ResNet50 to the appendix.

**Interpretation of results**. Figure 5 shows the validation errors of the resulting models on the 11-superclass problem. We make the following observations.

*Reasonably fine-grained labels benefit generalization, but there is a catch*. We can observe that in the blue curve of Figure 5 that, as long as the number of subclasses is less than $10^3$, we see obvious decline in the validation error on the target labels. In other words, reasonably fine-grained pretraining is indeed beneficial in this setting. We should note, however, the overall curve exhibits a U shape: overly fine-grained labels are not beneficial to downstream generalization. This is intuitive. If the pretraining granularity is too close to the target one, we should not expect improvement. On the other extreme, if we assign a unique label to *every* sample in the training data, it is highly likely that the only *differences* a model can find between each class would be frivolous details of the images, which would not be considered discriminative by the label function of the target coarse-label problem. In this case, the pretraining stage is almost meaningless and can be misleading, as evidenced by the very high label-per-sample error (red star in Figure 5).

*High granularity can be helpful, but label-assignment consistency is critical*. Random class ID pretraining (orange curve) performs the worst of all the alternatives. The label function of this type does not generate a *meaningful hierarchy* because it has no consistency in the features it considers discriminative when decomposing the superclasses. This is in stark contrast to the manual hierarchies, which decompose the superclasses based on the finer biological traits (mostly visual in nature) of the creatures in the image.

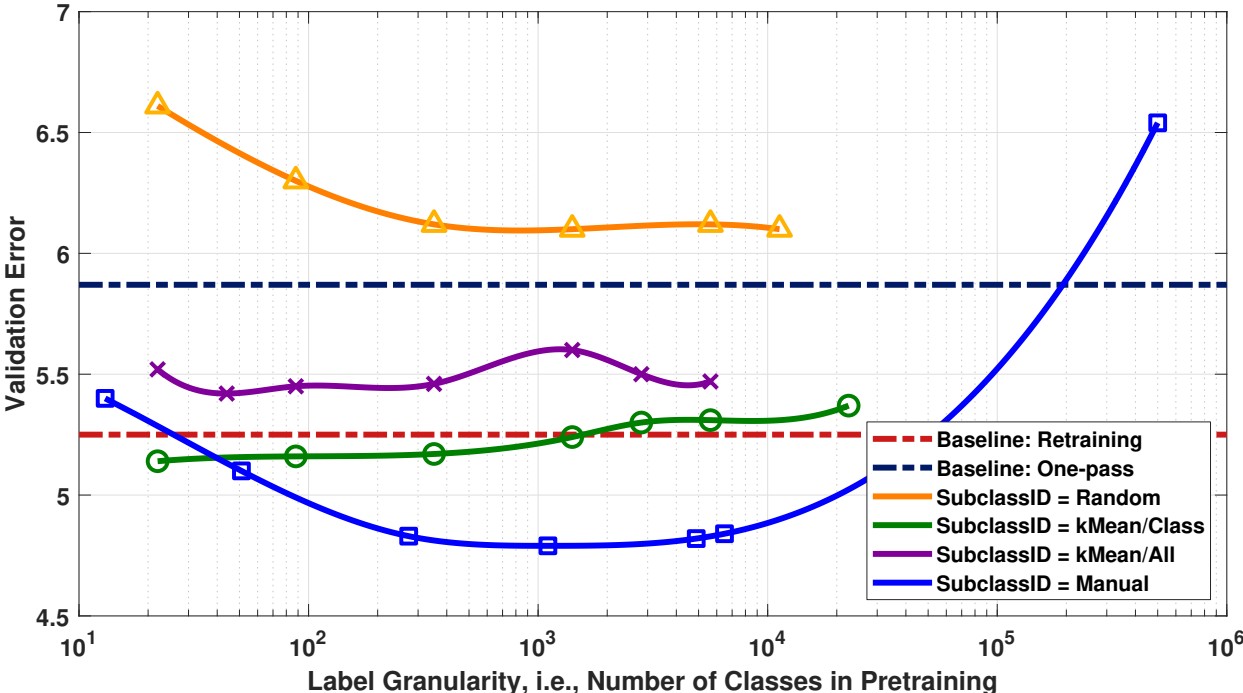

Figure 5: **In-dataset transfer**. ResNet34 validation error (with standard deviation) of finetuning on 11 superclasses of iNaturalist 2021, pretrained on various label hierarchies. The manual hierarchy outperforms the baseline and every other hierarchy, and exhibits a U-shaped curve.

*Alignment between fine-grained and target label spaces are important.* For fine-grained pretraining to be effective, the features that the pretraining label function considers discriminative must *align* well with those valued by the label function of the 11-superclass hierarchy. To see this point, observe that for models trained on cluster IDs obtained by performing kMeans on the CLIP embedding samples in each superclass *separately* (green curve in Figure 5), their validation errors are much lower than those trained on cluster IDs obtained by performing kMeans on the whole dataset (purple curve in Figure 5). As expected, the manually defined fine-grained label functions align best with that of the 11 superclasses, and the results corroborate this view.

## 7 Discussion

*Q: Are there other reasons why fine-grained labels benefit neural network generalization?*

A: Yes, it is possible, e.g., the optimization landscape induced by finer-grained labels could contain less saddle points, making it friendlier to SGD. We did not analyze this because our focus is primarily on the generalization instead of optimization aspect of the problem.

*Q: Does a higher label granularity always imply better generalization?*

A: No. There is an operating regime. Training a model with *pathologically* high label granularity is harmful. For example, if we assign a unique class to every sample in the dataset, the model will be forced to rely on the frivolous differences between each sample. We verify this intuition in Figure 5 in Appendix A.1.1 on the iNaturalist 2021 dataset. These extreme scenarios do not arise in common practice, so we do not focus on them in this paper.

*Q: The theoretical setting appears restrictive.*

A: Our theoretical setting is consistent with Cao et al. (2022); Allen-Zhu & Li (2023b; 2022); Shen et al. (2022b); Jelassi & Li (2022). With a limited number of available analytic tools in the literature, we believe these settings are necessary to keep things tractable.

# 8 Conclusion

In this paper, we formally studied the influence of pretraining label granularity on the generalization of DNNs, and performed large-scale experiments to complement our theoretical results. Under the new data model, hierarchical multi-view, we theoretically showed that higher label complexity leads to higher representation complexity, through which we explained why pretraining with fine-grained labels is beneficial to generalization. We complement our theory with experiments on ImageNet and iNaturalist, demonstrating that in the controlled setting of this paper, pretraining on reasonably fine-grained labels indeed benefits generalization.

**Broader Impact Statement**

This paper presents work whose goal is to advance the theory of deep learning. There are potential societal consequences of our work, none which we feel must be specifically highlighted here.

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

# Appendix

## A  Additional Experimental Results

In this section, we present the full details of our experiments and relevant ablation studies. All of our experiments were performed using tools in the Scenic library Dehghani et al. (2022).

### A.1  In-dataset transfer results

To clarify, in this transfer setting, we are essentially transferring *within* a dataset. More specifically, we set $\mathcal{X}^{\text{src}} = \mathcal{X}^{\text{tgt}}$ and only the label spaces $\mathcal{Y}^{\text{src}}$ and $\mathcal{Y}^{\text{tgt}}$ may differ (in distribution). The baseline in this setting is clear: train on $\mathcal{D}^{\text{tgt}}_{\text{train}}$ and test on $\mathcal{D}^{\text{tgt}}_{\text{test}}$. In contrast, after pretraining the backbone network $\boldsymbol{h}(\boldsymbol{\Theta}; \cdot)$ on $\mathcal{Y}^{\text{src}}$, we finetune or linear probe it on $\mathcal{D}^{\text{tgt}}_{\text{train}}$ using the backbone and then test on $\mathcal{D}^{\text{tgt}}_{\text{test}}$.

#### A.1.1  iNaturalist 2021

iNaturalist 2021 is well-suited for our analysis because it has a high-quality, manually defined label hierarchy that is based on the biological traits of the creatures in the images. Additionally, the large sample size of this dataset reduces the likelihood of sample-starved pretraining on reasonably fine-grained hierarchy levels. We use the mini training dataset with size 500,000 instead of the full training dataset to show a greater gap between the results of different hierarchies and speed up training.

We use the architectures ResNet 34 and 50 He et al. (2016).

**Training details**. Our pretraining pipeline on iNaturalist is essentially the same as the standard large-batch-size ImageNet-type training for ResNets He et al. (2016); Goyal et al. (2017). The following pipeline applies to model pretraining on any hierarchy.

- Optimization: SGD with 0.9 momentum coefficient, 0.00005 weight decay, 4096 batch size, 90 epochs total training length. We perform 7 epochs of linear warmup in the beginning of training until the learning rate reaches $0.1 \times 4096/256 = 1.6$, and then apply the cosine annealing schedule. Each training instance is run on 16 TPU v4 chips, taking around 2 hours per run.

- Data augmentation: subtracting mean and dividing by standard deviation, image (original or its horizontal flip) resized such that its shorter side is 256 pixels, then a $224 \times 224$ random crop is taken.

For finetuning, we keep everything in the pipeline the same except setting the batch size to $4096/4 = 1024$ and base learning rate $1.6/4 = 0.4$. We found that finetuning at higher batch size and learning rate resulted in training instabilities and severely affected the final finetuned model's validation accuracy, while finetuning at lower batch size and learning rate than the chosen one resulted in lower validation accuracy at the end even though their training dynamics was stabler.

For the baseline accuracy, as mentioned in the main text, to ensure fairness of comparison, in addition to only training the network on the target 11-superclass problem for 90 epochs (using the same pretraining pipeline), we also perform "retraining": follow the exact training process of the models trained on the various hierarchies, but use $\mathcal{D}^{\text{tgt}}_{\text{train}}$ as the training dataset in both the pretrianing and finetuning stage. We observed consistent increase in the final validation accuracy of the model, so we report this as the baseline accuracy. Without retraining (so naive one-pass 90-epoch training on 11 superclasses), the average accuracy with standard deviation is $94.13, 0.025$.

**Clustering**. To obtain the cluster-ID-based labels, we perform the following procedure.

1. For every sample $\boldsymbol{X}_n$ in the mini training dataset of iNaturalist 2021, obtain its ViT-L/14 CLIP embedding $\boldsymbol{E}_n$.

2. Per-superclass kMeans clustering. Let $C$ be the predefined number of clusters per class.

Table 1: **In-dataset transfer, iNaturalist 2021**. ResNet34 average finetuning validation error and standard deviation on 11 superclasses in iNaturalist 2021, pretrained on various label hierarchies with different label granularity. Baseline (11-superclass) and best performance are highlighted.

| Manual Hierarchy | $\mathcal{G}(\mathcal{Y}^{\mathrm{src}})$ | 11 | 13 | 51 | 273 | 1103 | 4884 | 6485 |
|---|---|---|---|---|---|---|---|---|
| | Validation error | **5.25±0.051** | 5.40±0.075 | 5.10±0.038 | 4.83±0.041 | **4.79±0.045** | 4.82±0.056 | 4.84±0.033 |
| Random class ID | $\mathcal{G}(\mathcal{Y}^{\mathrm{src}})$ | 22 | 88 | 352 | 1,408 | 5,632 | 11,264 | 500,000 |
| | Validation error | 6.61±0.215 | 6.30±0.070 | 6.12±0.77 | 6.10±0.053 | 6.12±0.042 | 6.10±0.057 | 6.54±0.758 |
| CLIP+kMeans | $\mathcal{G}(\mathcal{Y}^{\mathrm{src}})$ | 22 | 88 | 352 | 1408 | 2816 | 5632 | 22528 |
| per superclass | Validation error | 5.14±0.049 | 5.16±0.033 | 5.17±0.027 | 5.24±0.029 | 5.30±0.029 | 5.31±0.077 | 5.37±0.032 |
| C+k per supclass | $\mathcal{G}(\mathcal{Y}^{\mathrm{src}})$ | 88 | 218 | 320 | 608 | 1040 | 1984 | |
| Class rebalanced | Validation error | 5.18±0.054 | 5.17±0.038 | 5.23±0.052 | 5.28±0.045 | 5.26±0.035 | 5.21±0.040 | |
| CLIP+kMeans | $\mathcal{G}(\mathcal{Y}^{\mathrm{src}})$ | 22 | 44 | 88 | 352 | 1408 | 2816 | 5632 |
| whole dataset | Validation error | 5.52±0.015 | 5.42±0.047 | 5.45±0.049 | 5.46±0.019 | 5.60±0.029 | 5.50±0.029 | 5.47±0.029 |

    (a) For every superclass $k$, for the set of embedding $\{(\boldsymbol{E}_n, y_n = k)\}$ belonging to that superclass, perform kMeans clustering with cluster size set to $C$.

    (b) Given a sample with superclass ID $k \in \{1, 2, ..., 11\}$ and cluster ID $c \in \{1, 2, ..., C\}$, define its fine-grained ID as $C \times k + c$.

3. Whole-dataset kMeans clustering. Let $C$ be the predefined number of clusters on the whole dataset.

    (a) Perform kMeans on the embedding of all the samples in the dataset, with the number of clusters set to $C$. Set the fine-grained class ID of a sample to its cluster ID.

Some might have the concern that having the same number of kMeans clusters per superclass could cause certain classes to have too few samples, which could be a reason for why the cluster ID hierarchies perform worse than the manual hierarchies. Indeed, the number of samples per superclass on iNaturalist is different, so in addition to the above "uniform-number-of-cluster-per-superclass" hierarchy, we add an extra label hierarchy by performing the following procedure to balance the sample size of each cluster:

1. Perform kMeans for each superclass with number of clusters set to 2, 8, 32, 64, 128, 256, 512, 1024 and save the corresponding image-ID-to-cluster-ID dictionaries (so we are basically reusing the clustering results of the CLIP+kMeans per superclass experiment)

2. For each superclass, find the image-ID-to-cluster-ID dictionary with the highest granularity while still keeping the minimum number of samples for each cluster > predefined threshold (e.g. 1000 samples per subclass)

3. Now we have nonuniform granularity for each superclass while ensuring that the sample count per cluster is above some predefined threshold.

This simple procedure somewhat improves the balance of sample count per cluster, for example, Figure 6 shows the sample count per cluster for the cases of total number of clusters = 608 and 1984. Unfortunately, we do not observe any meaningful improvement on the model's validation accuracy trained on this more refined hierarchy.

**Experimental procedures**. All the validation accuracies we report on ResNet34 are the averaged results of experiments performed on at least 6 random seeds: 2 random seeds for backbone pretraining and 3 random seeds for finetuning. We report the average accuracies with their standard deviation on various hierarchies in Table 1.

An additional experiment we performed with ResNet34 is a small grid search over what checkpoint of a pretrained backbone we should use for finetuning on the 11-superclass method; we tried the 50-, 70- and 90-epoch checkpoints of the backbone on the manual hierarchies. We report these results in Table 2. As we can see, 90-epoch checkpoints performs almost equally well as the 70-epoch checkpoints and better than the 50-epoch ones by a nontrivial margin. With this observation, we chose to use the end-of-pretraining 90-epoch checkpoints in all our other experiments without further ablation studies on those hierarchies.

Table 2: **In-dataset transfer, iNaturalist 2021**. ResNet34 average finetuned validation error and standard deviation on 11 superclasses in iNaturalist 2021, pretrained on the manual hierarchies, with different backbone checkpoints.

| 90-Epoch ckpt | $\mathcal{G}(\mathcal{Y}^{\mathrm{src}})$ | 13 | 51 | 273 | 1103 | 4884 | 6485 |
|---|---|---|---|---|---|---|---|
| | Validation error | 5.40±0.075 | 5.10±0.038 | 4.83±0.041 | 4.79±0.045 | 4.82±0.056 | 4.84±0.033 |
| 70-Epoch ckpt | $\mathcal{G}(\mathcal{Y}^{\mathrm{src}})$ | 13 | 51 | 273 | 1103 | 4884 | 6485 |
| | Validation error | 5.43±0.055 | 5.08±0.029 | 4.86±0.037 | 4.82±0.034 | 4.83±0.064 | 4.85±0.018 |
| 50-Epoch ckpt | $\mathcal{G}(\mathcal{Y}^{\mathrm{src}})$ | 13 | 51 | 273 | 1103 | 4884 | 6485 |
| | Validation error | 5.53±0.036 | 5.2±0.031 | 4.90±0.038 | 4.9±0.042 | 4.91±0.020 | 4.95±0.026 |

Table 3: **In-dataset transfer, iNaturalist 2021**. ResNet50 finetuned average validation error and standard deviation on 11 superclasses in iNaturalist 2021, pretrained on label hierarchies with different label granularity.

| Manual Hierarchy | $\mathcal{G}(\mathcal{Y}^{\mathrm{src}})$ | 11 | 13 | 51 | 273 | 1103 | 4884 | 6485 |
|---|---|---|---|---|---|---|---|---|
| | Validation error | **4.43±0.029** | 4.44±0.063 | 4.36±0.062 | 4.22±0.021 | **4.20±0.035** | 4.23±0.054 | 4.33±0.037 |
| Random class ID | $\mathcal{G}(\mathcal{Y}^{\mathrm{src}})$ | 22 | 88 | 352 | 1,408 | 5,632 | 11,264 | 500,000 |
| | Validation error | 5.36±0.111 | 5.31±0.079 | 5.24±0.093 | 5.38±0.052 | 5.37±0.033 | 5.40±0.033 | 5.13±0.072 |

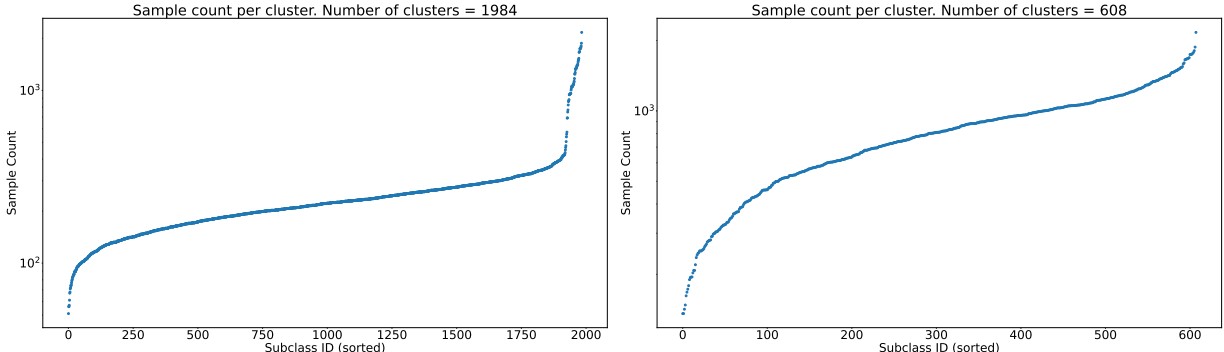

Figure 6: **In-dataset transfer, iNaturalist 2021**. Number of samples per cluster in the case of 608 and 1984 total clusters, after applying the sample size rebalancing procedure described in subsection A.1.1. Observe that the sample sizes are reasonably balanced across almost all the subclasses.

Table 4: **In-dataset transfer**. ViT-B/16 validation error on the binary problem "is this object a *living thing*?" of ImageNet21k. Pretrained on various hierarchy levels of ImageNet21k, finetuned on the binary problem. Observe that the maximal improvement appears at the leaf labels, and as $\mathcal{G}(\mathcal{Y}^{\mathrm{src}})$ approaches 2, the percentage improvement approaches 0.

| Hierarchy level | $\mathcal{G}(\mathcal{Y}^{\mathrm{src}})$ | Validation error |
|---|---|---|
| Baseline | 2 | **7.90** |
| 0 (leaf) | 21843 | **6.56** |
| 1 | 5995 | 6.76 |
| 2 | 2281 | 6.70 |
| 4 | 519 | 6.97 |
| 6 | 160 | 7.31 |
| 9 | 38 | 7.55 |

Our ResNet50 results are not as extensive as those on ResNet34. We present the average accuracies and standard deviations in Table 3.

### A.1.2  ImageNet21k

The ImageNet21k dataset we experiment on contains a total of 12,743,321 training samples and 102,400 validation samples, with 21843 leaf labels. A small portion of samples have multiple labels.

Caution: due to the high demand on computational resources of training ViT models on ImageNet21k, all of our experiments that require (pre-)training or finetuning/linear probing on this dataset were performed with one random seed.

**Hierarchy generation**. To define fine-grained labels, we start by defining the leaf labels of the dataset to be Hierarchy level 0. For every image, we trace from the leaf synset to the root synset relying on the WordNet hierarchy, and set the $k$-th synset (or the root synset, whichever is higher in level) as the level-$k$ label of this image; this procedure also applies to the multi-label samples. This is the way we generate the manual hierarchies shown in the main text.

Due to the lack of a predefined coarse-label problem, we manually define our target problem to be a binary one: given an image, if the synset "Living Thing" is present on the path tracing from the leaf label of the image to the root, assign label 1 to this image; otherwise, assign 0. This problem almost evenly splits the training and validation sets of ImageNet21k: 5,448,549:7,294,772 for training, 43,745:58,655 for validation.

**Network choice and pretraining pipeline**. We experiment with the ViT-B/16 model Dosovitskiy et al. (2021). The pretraining pipeline of this model follows the one in Dosovitskiy et al. (2021) exactly: we train the model for 90 epochs using the Adam optimizer, with $\beta_1 = 0.9, \beta_2 = 0.999$, weight decay coefficient equal to 0.03 and a batch size of 4096; we let the dropout rate be 0.1; the output dense layer's bias is initialized to $-10.0$ to prevent huge loss value coming from the off-diagonal classes near the beginning of training Cui et al. (2019b); for learning rate, we perform linear warmup for 10,000 steps until the learning rate reaches $10^{-3}$, then it is linearly decayed to $10^{-5}$. The data augmentations are the common ones in ImageNet-type training Dosovitskiy et al. (2021); He et al. (2016): random cropping and horizontal flipping. Note that we use the sigmoid cross-entropy for training since the dataset has multi-label samples.

Each training instance (90 epochs) is run on 64 TPU v4 chips, taking approximately 1.5 to 2 days.

**Evaluation on the binary problem**. After the 90-epoch pretraining on the manual hierarchies, we evaluate the model on the binary problem. We report the best accuracies on each hierarchy level in Table 4. To get a sense of how the relevant hyperparameters influence final accuracy of the model, we try out the following finetuning/linear probing strategies on the backbone trained on the *leaf labels* and the target *binary problem* of the dataset, and report the results in Table 5 (similar to our experiments on iNaturalist, we include the backbone trained on the binary problem in these ablation studies to ensure that our comparisons against the baseline are fair) :

1. 90-epochs finetuning in the same fashion as the pretraining stage, but with a small grid search over

$$(\text{batch size, base learning rate}) = \{(4096, 0.001), (4096/4 = 1024, 0.001/4 = 0.00025),$$
$$(4096/8 = 512, 0.001/8 = 0.000125)\}.$$

2. Linear probing with 20 epochs training length, using exactly the same training pipeline as in pretraining. We ran a small grid search over (batch size, base learning rate) = $\{(4096, 0.001), (4096/8 = 512, 0.001/8 = 0.000125)\}$.

3. 10-epochs finetuning, no linear warmup, 3 epochs of constant learning rate in the beginning followed by 7 epochs of linear decay, with a small grid search over (batch size, base learning rate) = $\{(4096, 0.001), (4096/8 = 512, 0.001/8 = 0.000125)\}$.

Table 5 helps us decide the best accuracies to report. First, as expected the linear probing results are much worse than the finetuning ones. Second, the "retraining" accuracy of 92.102 is the best baseline we can report (the same thing happened in the iNaturalist case) — if we only train the model for 90 epochs (the naive one-pass training) on the binary problem, then the model's final validation accuracy is 91.746%, which is lower than 92.102% by a nontrivial margin. In contrast, the short 10-epoch finetuning strategy

Table 5: **In-dataset transfer, ImageNet21k**. ViT-B/16 validation accuracy on the binary problem "Is the object a Living Thing" on ImageNet21k. Ablation study on the exact finetuning/linear probing strategy.

| Eval strategy | | 90-epoch finetune | | | Linear probe | | 10-epoch finetune | |
|---|---|---|---|---|---|---|---|---|
| Leaf-pretrained | (Batch size, base lr) | (4096,1e-3) | (1024,2.5e-4) | (512,1.25e-4) | (4096, 1e-3) | (512, 1.25e-4) | (4096,1e-3) | (512,1.25e-4) |
| | Validation error | 92.782 | 93.177 | 93.295 | 87.497 | 87.493 | 92.294 | **93.439** |
| Baseline | (Batch size, base lr) | (4096,1e-3) | (1024,2.5e-4) | (512,1.25e-4) | (4096, 1e-3) | (512, 1.25e-4) | (4096,1e-3) | (512,1.25e-4) |
| | Validation error | 92.102 | 91.971 | 91.939 | 91.703 | 91.719 | 92.002 | 91.856 |

Table 6: **In-dataset transfer, ImageNet1k**. ResNet50 finetuned average validation error and standard deviation on the vanilla 1000 classes, pretrained on label hierarchies with different label granularity.

| ResNet50 CLIP+kMeans | $\mathcal{G}(\mathcal{Y}^{\mathrm{src}})$ | 2000 | 4000 | 8000 |
|---|---|---|---|---|
| per-class | Validation error | 23.4±0.13 | 23.48±0.098 | 23.49±0.204 |
| ViT-L/14 CLIP+kMeans | $\mathcal{G}(\mathcal{Y}^{\mathrm{src}})$ | 2000 | 4000 | 8000 |
| per-class | Validation error | 23.4±0.127 | 23.47±0.074 | 23.78±0.048 |
| Random ID | $\mathcal{G}(\mathcal{Y}^{\mathrm{src}})$ | 2000 | 4000 | 8000 |
| per-class | Validation error | 23.4±0.068 | 23.4±0.070 | 23.65±0.071 |

works best for the backbone trained on the leaf labels, therefore, we also use this strategy to evaluate the backbones trained on all the other manual hierarchies. A peculiar observation we made was that, finetuning the leaf-labels-pretrained backbone for extended period of time on the binary problem caused it to overfit severely: for batch size and base learning rate in the set $\{(4096, 0.001), (1024, 0.00025), (512, 0.000125)\}$, throughout the 90 epochs of finetuning, although its training loss exhibits the normal behavior of staying mostly monotonically decreasing, its validation accuracy actually reached its peak during the linear warmup period!

### A.1.3 ImageNet1k

Our ImageNet1k in-dataset transfer experiments are done in a very similar fashion to the iNaturalist ones. In particular, the pretraining and finetuning pipeline for ResNet50 is exactly the same as the one in the iNaturalist case, so we do not repeat it here.

Due to a lack of more fine-grained manual label on this dataset, we generate fine-grained labels by performing kMeans on the ViT-L/14 CLIP embedding of the dataset separately for each class; the exact procedure is also identical to the iNaturalist case. The CLIP backbones we use here are the ResNet50 version and the ViT-L/14 version. We report the average accuracies and their standard deviation in Table 6. All results are obtained from at least one random seed during pretraining and 3 random seeds during finetuning.

The best baseline we report is the one using retraining: if we adopt the pretrain-then-finetune procedure but with $\mathcal{D}_{\mathrm{train}}^{\mathrm{tgt}}$ (i.e. the vanilla 1000-class labels) set as the pretraining dataset, then we obtain an average validation error of 23.28% with standard deviation of 0.103, averaged over results of 3 random seeds. In comparison, if we only perform the naive one-pass 90-epoch training, we obtain average valiation error 24.04%, with standard deviation 0.057.

From Table 6, we see that there is virtually no difference between the baseline and the best errors obtained by the models trained on the custom hierarchies: they are almost equally bad. Noting that the sample size of each class in ImageNet1k is only around $10^3$, and the fact that ImageNet1k classification is a "hard problem" — it is a problem of high sample complexity — further decomposing the classes causes each fine-grained class to have too few samples, leading to the above negative results. This reflects the intuition that higher label granularity does not necessarily mean better model generalization, since the sample size per class might become too small.

Table 7: **Cross-dataset transfer**. ViT-B/16 average *finetuning* validation accuracy on ImageNet1k along with standard deviation, pretrained on various hierarchy levels of ImageNet21k, and a small grid search over the base learning rate.

| Pretrained on / Base lr | $3 \times 10^{-3}$ | $3 \times 10^{-2}$ | $6 \times 10^{-2}$ | $3 \times 10^{-1}$ |
|---|---|---|---|---|
| ImageNet21k, Hier. lv. 0 | 80.87±0.012 | 82.48±0.005 | **82.51±0.042** | 81.40±0.041 |
| ImageNet21k, Hier. lv. 1 | 77.38±0.037 | 81.03±0.054 | **81.28±0.045** | 80.40±0.087 |
| ImageNet21k, Hier. lv. 2 | 74.91±0.012 | 79.76±0.021 | **80.26±0.05** | 79.7±0.019 |
| ImageNet21k, Hier. lv. 4 | 63.65±0.052 | 76.43±0.033 | **77.32±0.088** | 77.53±0.078 |
| ImageNet21k, Hier. lv. 6 | 62.17±0.012 | 73.65±0.033 | 73.92±0.073 | **75.53±0.024** |
| ImageNet21k, Hier. lv. 9 | 53.68±0.034 | 69.33±0.045 | 71.08±0.068 | **72.75±0.071** |

Table 8: **Cross-dataset transfer**. ViT-B/16 average *linear-probing* validation accuracy on ImageNet1k along with standard deviation, pretrained on various hierarchy levels of ImageNet21k.

| Pretrained on | Hier. lv | $\mathcal{G}(\mathcal{Y}^{\mathrm{src}})$ | Validation acc. |
|---|---|---|---|
| IM21k | 0 (leaf) | 21843 | **81.45±0.021** |
| | 1 | 5995 | 78.33±0.018 |
| | 2 | 2281 | 75.66±0.005 |
| | 4 | 519 | 68.95±0.051 |
| | 6 | 160 | 63.65±0.035 |
| | 9 | 38 | 57.35±0.016 |

## A.2 Cross-dataset transfer, ImageNet21k→ImageNet1k

In this subsection, we report the average validation accuracy and standard deviation of the cross-dataset transfer experiment from ImageNet21k to ImageNet1k, as discussed in Figure 2 and Section 1 in the main text.

*Network choice.* We use the same architecture ViT-B/16 as the one in the in-dataset ImageNet21k transfer experiment and follow the same training procedure, which we repeat here for the reader's convenience. The pretraining pipeline of this model follows the one in Dosovitskiy et al. (2021): we train the model for 90 epochs using the Adam optimizer, with $\beta_1 = 0.9, \beta_2 = 0.999$, weight decay coefficient equal to 0.03 and a batch size of 4096; we let the dropout rate be 0.1; the output dense layer's bias is initialized to $-10.0$ to prevent huge loss value coming from the off-diagonal classes near the beginning of training Cui et al. (2019b); for learning rate, we perform linear warmup for 10,000 steps until the learning rate reaches $10^{-3}$, then it is linearly decayed to $10^{-5}$. The data augmentations are the common ones in ImageNet-type training Dosovitskiy et al. (2021); He et al. (2016): random cropping and horizontal flipping. Note that we use the sigmoid cross-entropy for training since the dataset has multi-label samples.

Additionally, each training instance (90 epochs) is run on 64 TPU v4 chips, taking approximately 1.5 to 2 days.

*Finetuning.* For finetuning on ImageNet1k, our procedure is very similar to the one in the original ViT paper Dosovitskiy et al. (2021), described in its Appendix B.1.1. We optimize the network for 8 epochs using SGD with momentum factor set to 0.9, zero weight decay, and batch size of 512. The dropout rate, unlike in pretraining, is set to 0. Gradient clipping at 1.0 is applied. Unlike Dosovitskiy et al. (2021), we still finetune at the resolution of 224×224. For learning rate, we apply linear warmup for 500 epochs until it reaches the base learning rate, then cosine annealing is applied; we perform a small grid search of base learning rate = $\{3 \times 10^{-3}, 3 \times 10^{-2}, 6 \times 10^{-2}, 3 \times 10^{-1}\}$. Every one of these grid search is repeated over

3 random seeds. We report the ImageNet1k validation accuracies and their standard deviations in Table 7. In the main text, we report the best accuracy for each hierarchy level.

*Linear probing.* For linear probing, we use the following procedure. We optimize the linear classifier for 40 epochs (similar to Lee et al. (2021)) using SGD with Nesterov momentum factor set to 0.9, a small weight decay coefficient $10^{-6}$, and batch size 512. We start with a base learning rate of 0.9, and multiply it by 0.97 per 0.5 epoch. In terms of data augmentation, we adopt the standard ones like before: horizontal flipping and random cropping of size 224×224. We repeat this linear probing procedure over 3 random seeds given the pretrained backbone, and report the average validation accuracy and standard deviation in Table 8.

*Baseline.* The baseline accuracy on ImageNet1k is directly taken from the ViT paper Dosovitskiy et al. (2021) (see Table 5 in it), in which the ViT-B/16 model is trained for 300 epochs on ImageNet1k.

# B Theory, Problem Setup

## B.1 Data Properties

1. Coarse classification: a binary task, $+1$ vs. $-1$.

2. An input sample $\boldsymbol{X} \in \mathbb{R}^{d \times P}$ consists of $P$ patches, each with dimension $d$. In this work, always assume $d$ is sufficiently large[1];

3. Assume there exists $k_+$ subclasses of the superclass "+", and $k_-$ subclasses of the superclass "−". Let $k_+ = k_-$.

4. Assume orthonormal dictionary $\mathcal{V} = \{\boldsymbol{v}_1, ..., \boldsymbol{v}_d\} \subset \mathbb{R}^d$, which forms an orthonormal basis of $\mathbb{R}^d$. Define $\boldsymbol{v}_+ \in \mathcal{V}$ to be the common feature of class "+". For each subclass $(+, c)$ (where $c \in [k_+]$), denote the subclass feature of it as $\boldsymbol{v}_{+,c} \in \mathcal{V}$. Similar for the "−" class.

5. For an easy sample $\boldsymbol{X}$ belonging to the $(+, c)$ class (for $c \in [k_+]$), we sample its patches as follows:

   **Definition**: we define the function $\mathcal{P} : \mathbb{R}^{d \times P} \times \mathcal{V} \to [P]$ (so $(\boldsymbol{X}; \boldsymbol{v}) \mapsto I \subseteq [P]$) to extract, from sample $\boldsymbol{X}$, the indices of the patches on which the dictionary word $\boldsymbol{v} \in \mathcal{D}$ dominates.

   (a) (Common-feature patches) With probability $\frac{s^*}{P}$, a patch $\boldsymbol{x}_p$ in $\boldsymbol{X}$ is a common-feature patch, on which $\boldsymbol{x}_p = \alpha_p \boldsymbol{v}_+ + \boldsymbol{\zeta}_p$ for some (random) $\alpha_p \in \left[\sqrt{1 - \iota}, \sqrt{1 + \iota}\right]$;

   (b) (Subclass-feature patches) With probability $\frac{s^*}{P - |\mathcal{P}(\boldsymbol{X}; \boldsymbol{v}_+)|}$, a patch with index $p \in ([P] - \mathcal{P}(\boldsymbol{X}; \boldsymbol{v}_+))$ is a subclass-feature patch, on which $\boldsymbol{x}_p = \alpha_p \boldsymbol{v}_{+,c} + \boldsymbol{\zeta}_p$, for random $\alpha_p \in \left[\sqrt{1 - \iota}, \sqrt{1 + \iota}\right]$;

   (c) (Noise patches) For the remaining $P - |\mathcal{P}(\boldsymbol{X}; \boldsymbol{v}_+)| - |\mathcal{P}(\boldsymbol{X}; \boldsymbol{v}_{+,c})|$ patches, $\boldsymbol{x}_p = \boldsymbol{\zeta}_p$.

6. A hard sample $\boldsymbol{X}_{\text{hard}}$ for class $(+, c)$ is exactly the same as an easy one except:

   (a) Its common-feature patches are replaced by noise patches;

   (b) (Feature noise patches) With probability $\frac{s^\dagger}{P - |\mathcal{P}(\boldsymbol{X}; \boldsymbol{v}_{+,c})|}$, a patch with index $p \in ([P] - \mathcal{P}(\boldsymbol{X}; \boldsymbol{v}_{+,c}))$ is a feature-noise patch, on which $\boldsymbol{x}_p = \alpha_p^\dagger \boldsymbol{v}_- + \boldsymbol{\zeta}_p$ for some (random) $\alpha_p \in \left[\iota_{lower}^\dagger, \iota_{upper}^\dagger\right]$;

   (c) Set one of the noise patches to $\boldsymbol{\zeta}^* \sim \mathcal{N}(\boldsymbol{0}, \sigma_{\zeta^*}^2 \boldsymbol{I}_d)$.

7. A sample $\boldsymbol{X}$ belongs to the "+" superclass if $|\mathcal{P}(\boldsymbol{X}; \boldsymbol{v}_+)| > 0$ or $|\mathcal{P}(\boldsymbol{X}; \boldsymbol{v}_{+,c})| > 0$ for any $c$ (excluding feature-noise patches).

8. The above sample definitions also apply to the "−" classes by switching the class signs.

9. A training batch of samples contains exactly $N/2k_+$ samples for each $(+, c)$ and $(-, c)$ subclass. This also means that each training batch contains exactly $N/2$ samples belonging to the $+1$ superclass, and $N/2$ samples for the $-1$ superclass.

10. As discussed in the main text, for both coarse-grained (baseline) and fine-grained training, we only train on *easy* samples.

## B.2 Learner Assumptions and Training Algorithm

Assume the learner is a two-layer convolutional ReLU network:

$$F_c(\boldsymbol{X}) = \sum_{r=1}^{m} a_{c,r} \sum_{p=1}^{P} \sigma(\langle \boldsymbol{w}_{c,r}, \boldsymbol{x}_p \rangle + b_{c,r}) \tag{14}$$

To simplify analysis and only focus on the learning of the feature extractor, we freeze $a_{c,r} = 1$ throughout training. The nonlinear activation $\sigma(\cdot) = \max(0, \cdot)$ is ReLU. Note that the convolution kernels have dimension $d$ and stride $d$.

---

[1]Consider each $d$-dimensional patch of the input as an embedding of the input image generated by, for instance, an intermediate layer of a DNN.

*Remark.* One difference between this architecture and a CNN used in practice is that we do not allow feature sharing across classes: for each class $c$, we are assigning a disjoint group of neurons $\boldsymbol{w}_{c,r}$ to it. Separating neurons for each class is a somewhat common trick to lower the complexity of analysis in DNN theory literature Allen-Zhu & Li (2023b); Karp et al. (2021); Cao et al. (2022), as it reduces complex coupling between neurons *across* classes which is not the central focus of our study in this paper.

Now we discuss the **training algorithm**.

**Initialization**.

Sample $\boldsymbol{w}_{c,r}^{(0)} \sim \mathcal{N}(\boldsymbol{0}, \sigma_0^2 \boldsymbol{I}_d)$, and set $b_{c,r}^{(0)} = -\sigma_0 c_b \sqrt{\log(d)}$.

**Training**.

We adopt the standard cross-entropy training:

$$\mathcal{L}(F) = \sum_{n=1}^{N} L(F; \boldsymbol{X}_n, y_n) = -\sum_{n=1}^{N} \log \left( \frac{\exp(F_{y_n}(\boldsymbol{X}_n))}{\sum_{c=1}^{C} \exp(F_c(\boldsymbol{X}_n))} \right) \tag{15}$$

This induces the stochastic gradient descent update for each hidden neuron ($c \in [k], r \in [m]$) per minibatch of $N$ iid samples:

$$\boldsymbol{w}_{c,r}^{(t+1)} = \boldsymbol{w}_{c,r}^{(t)} + \eta \frac{1}{NP} \sum_{n=1}^{N} \Bigg( \mathbb{1}\{y_n = c\}[1 - \text{logit}_c^{(t)}(\boldsymbol{X}_n^{(t)})] \sum_{p \in [P]} \sigma'(\langle \boldsymbol{w}_{c,r}^{(t)}, \boldsymbol{x}_{n,p}^{(t)} \rangle + b_{c,r}^{(t)}) \boldsymbol{x}_{n,p}^{(t)} +$$

$$\mathbb{1}\{y_n \neq c\}[-\text{logit}_c^{(t)}(\boldsymbol{X}_n^{(t)})] \sum_{p \in [P]} \sigma'(\langle \boldsymbol{w}_{c,r}^{(t)}, \boldsymbol{x}_{n,p}^{(t)} \rangle + b_{c,r}^{(t)}) \boldsymbol{x}_{n,p}^{(t)} \Bigg) \tag{16}$$

where

$$\text{logit}_c^{(t)}(\boldsymbol{X}) = \frac{\exp(F_c(\boldsymbol{X}))}{\sum_{y=1}^{C} \exp(F_y(\boldsymbol{X}))} \tag{17}$$

As for the bias,

$$b_{c,r}^{(t+1)} = b_{c,r}^{(t)} - \frac{\|\boldsymbol{w}_{c,r}^{(t+1)} - \boldsymbol{w}_{c,r}^{(t)}\|_2}{\log^5(d)} \tag{18}$$

*Remark.* 1. The initialization strategy is similar to the one in Allen-Zhu & Li (2022).

2. Since the only difference between the training samples of coarse and fine-grained pretraining is the label space, the form of SGD update is identical. The only difference is the number of output nodes of the network: for coarse training, the output nodes are just $F_+$ and $F_-$ (binary classification), while for fine-grained training, the output nodes are $F_{+,1}, F_{+,2}, ..., F_{+,k_+}, F_{-,1}, F_{-,2}, ..., F_{-,k_-}$, a total of $k_+ + k_-$ nodes.

3. The bias is for thresholding out the neuron's noisy activations that grow slower than $1/\log^5(d)$ times the activations on the features which the neuron detects. This way, the bias does not really influence updates to the neuron's response to the (common and/or fine-grained) features which it activates strongly on, since $1 - \frac{1}{\log^5(d)} \approx 1$, while it removes useless low-magnitude noisy activations. This in fact creates a (generalization) gap between the nonlinear model that we are studying and linear models. Due to our parameter choices (as discussed below), if the model has no nonlinearity (remove the ReLU activations), then even if the model can be written as $F_+(\boldsymbol{X}) = \sum_{p \in [P]} c_+ \langle \boldsymbol{v}_+, \boldsymbol{x}_p \rangle + c_{+,1} \langle \boldsymbol{v}_{+,1}, \boldsymbol{x}_p \rangle +$ $... + c_{+,k_+} \langle \boldsymbol{v}_{+,k_+}, \boldsymbol{x}_p \rangle$ and $F_-(\boldsymbol{X}) = \sum_{p \in [P]} c_- \langle \boldsymbol{v}_-, \boldsymbol{x}_p \rangle + c_{-,1} \langle \boldsymbol{v}_{-,1}, \boldsymbol{x}_p \rangle + ... + c_{-,k_-} \langle \boldsymbol{v}_{-,k_-}, \boldsymbol{x}_p \rangle$ for any sequence of nonnegative real numbers $c_+, c_-, \{c_{+,j}\}_{j=1}^{k_+}, \{c_{-,j}\}_{j=1}^{k_-}$ (which is the ideal situation since the true features are not corrupted by anything), it is impossible for the model to reach $o(1)$ error on the input samples, because the number of noise patches will accumulate to a variance of $(P - O(s^*))\sigma_\zeta \gg O(s^*)$, which significantly overwhelms the signal from the true features. On the

other hand, each noise patch is sufficiently small in magnitude with high probability (their strength is $o(1/\log^5(d))$), so a slightly negative bias, as described above, can threshold out these noise-based signals and prevent them from accumulating across the patches.

An important difference between our bias update rule and the one in Allen-Zhu & Li (2022) is that, our rule depends on the $\ell_2$ norm of the neuron's update, while the one in Allen-Zhu & Li (2022) is hard-coded and not dependent on the neuron weights. The reason that we should not hard code the bias update rate is that, the neurons that are responsible for detecting the common features will grow more quickly in norm than those responsible for detecting the fine-grained features, therefore, to ensure fairness between the different groups of neurons (i.e. only using the bias to remove useless activations on the noise patches while creating minimal disturbance to the neurons' activation on feature-dominated patches), we rely on our neuron-dependent bias update rule.

### B.3 Parameter Choices

The following are fixed choices of parameters for the sake of simplicity in our proofs.

1. Always assume $d$ is sufficiently large. All of our asymptotic results are presented with respect to $d$;

2. $\mathrm{poly}(d)$ denotes the asymptotic order "polynomial in $d$";

3. $\mathrm{polylog}(d)$ aymptotic order "polylogarithmic in $d$";

4. $\mathrm{polylog}(d) \leq k_+ = k_- \leq d^{0.4}$ and $s^* \log^5(d) \leq k_+$ (i.e. $k_+$ lower bounded by polynomial of $\log(d)$ of sufficiently high degree);

5. Small positive constant $c_0 \in (0, 0.1)$;

6. For coarse-grained (baseline) training, set $c_b = \sqrt{4 + 2c_0}$, and for fine-grained training, set $c_b = \sqrt{2 + 2c_0}$;

7. $0 \leq \iota \leq \frac{1}{\mathrm{polylog}(d)}$;

8. $\iota^\dagger_{lower} \geq \frac{1}{\log^4(d)}$, and $s^\dagger \iota^\dagger_{upper} \leq O\left(\frac{1}{\log(d)}\right)$;

9. $s^\dagger \geq 1$;

10. $s^* \in \mathrm{polylog}(d)$ with a degree $> 15$;

11. $\sigma_\zeta = \frac{1}{\log^{10}(d)\sqrt{d}}$;

12. $\sigma_{\zeta^*} \in \left[\omega\left(\frac{\mathrm{polylog}(d)}{\sqrt{d}}\right), O\left(\frac{1}{\mathrm{polylog}(d)}\right)\right]$;

13. $P\sigma_\zeta \geq \omega(\mathrm{polylog}(d))$, and $P \leq \mathrm{poly}(d)$;

14. $\sigma_0 \leq O\left(\frac{1}{d^3 s^* \log(d)}\right)$, and set $\eta = \Theta(\sigma_0)$ for simplicity;

15. Batch of samples $\mathcal{B}^{(t)}$ at every iteration has a deterministic size of $N \in (\Omega(\mathrm{polylog}(d)k_+ d), \mathrm{poly}(d))$.

16. Note: we sometimes abuse the notation $x = a \pm b$ as an abbreviation for $x \in [a - b, a + b]$.

*Remark.* We believe the range of parameter choice can be (asymptotically) wider than what is considered here, but for the purpose of illustrating the main messages of the paper, we do not consider a more general set of parameter choice necessary because having a wider range of it can significantly complicate and obscure the already lengthy proofs without adding to the core messages.

### B.4 Plan of presentation and central ideas

We shall devote the majority of our effort to proving results for the coarse-label learning dynamics, starting with appendix section C and ending on E, and only devote section G to the fine-grained-label learning dynamics, since the analysis of fine-grained training overlaps significantly with the coarse-grained one.

One technical difficulty in making the above ideas rigorous lies in the ReLU activation (with time-dependent bias): due to randomness in the gradient updates and the initialization, it is possible for individual hidden neurons that activate on $v$-dominated patches at one time iterate to no longer do so at the next iterate, and the opposite can happen. This can be problematic: for instance, it is possible that certain "lucky" neurons for $v_+$ at one iterate become dead on $v_+$-dominated patches at the next iterate, while some "unlucky" neurons that were dead on $v_{+,c}$-dominated patches before start activating on these patches at the current iterate. In our proof, we show that this kind of situation does not happen too frequently nor do they contribute too much to the overall behavior of the neural network, by carefully keeping track of each hidden neuron's response to feature vectors and noise vectors throughout training.

## C   Coarse-grained training, Initialization Geometry

**For coarse-grained training, assume $m = \Theta(d^{2+2c_0})$.**

**Definition C.1.** Define the following sets of interest of the hidden neurons:

1. $\mathcal{U}_{+,r}^{(0)} = \{v \in \mathcal{V} : \langle w_{+,r}^{(0)}, v \rangle \geq \sigma_0\sqrt{4+2c_0}\sqrt{\log(d) - \frac{1}{\log^5(d)}}\}$

2. Given $v \in \mathcal{V}$, $S_+^{*(0)}(v) \subseteq + \times [m]$ satisfies:

    (a) $\langle w_{+,r}^{(0)}, v \rangle \geq \sigma_0\sqrt{4+2c_0}\sqrt{\log(d) + \frac{1}{\log^5(d)}}$

    (b) $\forall v' \in \mathcal{V}$ s.t. $v' \perp v$, $\langle w_{+,r}^{(0)}, v' \rangle < \sigma_0\sqrt{4+2c_0}\sqrt{\log(d) - \frac{1}{\log^5(d)}}$

3. Given $v \in \mathcal{D}$, $S_+^{(0)}(v) \subseteq + \times [m]$ satisfies:

    (a) $\langle w_{+,r}^{(0)}, v \rangle \geq \sigma_0\sqrt{4+2c_0}\sqrt{\log(d) - \frac{1}{\log^5(d)}}$

4. For any $(+,r) \in S_{+,reg}^{*(0)} \subseteq + \times [m]$:

    (a) $\langle w_{+,r}^{(0)}, v \rangle \leq \sigma_0\sqrt{10}\sqrt{\log(d)} \; \forall v \in \mathcal{V}$

    (b) $\left| \mathcal{U}_{+,r}^{(0)} \right| \leq O(1)$

**Proposition 1.** *Assume $m = \Theta(d^{2+2c_0})$, i.e. the number of neurons assigned to the $+$ and $-$ class are equal and set to $\Theta(d^{2+2c_0})$.*

*At $t = 0$, for all $v \in \mathcal{V}$, the following properties are true with probability at least $1 - d^{-2}$ over the randomness of the initialized kernels:*

1. *$|S_+^{*(0)}(v)|, |S_+^{(0)}(v)| = \Theta\left(\frac{1}{\sqrt{\log(d)}}\right) d^{c_0}$*

2. *In particular, for any $v, v' \in \mathcal{V}$, $\left| \frac{|S_+^{*(0)}(v)|}{|S_+^{*(0)}(v')|} - 1 \right|, \left| \frac{|S_+^{*(0)}(v)|}{|S_+^{(0)}(v')|} - 1 \right| \leq O\left(\frac{1}{\log^5(d)}\right)$*

3. *$S_{+,reg}^{(0)} = [m]$*

*Proof.* Recall the tail bound of $g \sim \mathcal{N}(0,1)$ for every $\epsilon > 0$:

$$\frac{1}{2}\frac{1}{\sqrt{2\pi}}\frac{\epsilon}{\epsilon^2 + 1}e^{-\epsilon^2/2} \leq \mathbb{P}[g \geq \epsilon] \leq \frac{1}{2}\frac{1}{\sqrt{2\pi}}\frac{1}{\epsilon}e^{-\epsilon^2/2} \tag{19}$$

First note that for any $r \in [m]$, $\{\langle w_{+,r}^{(0)}, v \rangle\}_{v \in \mathcal{V}}$ is a sequence of iid random variables with distribution $\mathcal{N}(0, \sigma_0^2)$.

The proof of the first point proceeds in two steps.

1. The following properties hold at $t = 0$:

$$p_1 := \mathbb{P}\left[\langle \boldsymbol{w}_{+,r}^{(0)}, \boldsymbol{v}\rangle \geq \sigma_0\sqrt{4 + 2c_0}\sqrt{\log(d) + \frac{1}{\log^5(d)}}\right]$$

$$\in \frac{1}{\sqrt{8\pi}}d^{-2-c_0}e^{(-2-c_0)/\log^5(d)}$$

$$\times \left[\frac{\sqrt{(4 + 2c_0)\left(\log(d) + \frac{1}{\log^5(d)}\right)}}{(4 + 2c_0)\left(\log(d) + \frac{1}{\log^5(d)}\right) + 1}, \frac{1}{\sqrt{(4 + 2c_0)\left(\log(d) + \frac{1}{\log^5(d)}\right)}}\right] \qquad (20)$$

$$= \Theta\left(\frac{1}{\sqrt{\log(d)}}\right)d^{-2-c_0}$$

and

$$p_2 := \mathbb{P}\left[\langle \boldsymbol{w}_{+,r}^{(0)}, \boldsymbol{v}\rangle \geq \sigma_0\sqrt{4 + 2c_0}\sqrt{\log(d) - \frac{1}{\log^5(d)}}\right]$$

$$\in \frac{1}{\sqrt{8\pi}}d^{-2-c_0}e^{-(-2-c_0)/\log^5(d)}$$

$$\times \left[\frac{\sqrt{(4 + 2c_0)\left(\log(d) - \frac{1}{\log^5(d)}\right)}}{(4 + 2c_0)\left(\log(d) - \frac{1}{\log^5(d)}\right) + 1}, \frac{1}{\sqrt{(4 + 2c_0)\left(\log(d) - \frac{1}{\log^5(d)}\right)}}\right] \qquad (21)$$

$$= \Theta\left(\frac{1}{\sqrt{\log(d)}}\right)d^{-2-c_0}$$

Therefore, for any $r \in [m]$, the random event described in $S_+^{*(0)}$ holds with probability

$$p_1 \times (1 - p_2)^{d-1} = \Theta\left(\frac{1}{\sqrt{\log(d)}}\right)d^{-2-c_0} \times \left(1 - \Theta\left(\frac{1}{\sqrt{\log(d)}}\right)d^{-2-c_0}\right)^{d-1}$$

$$= \Theta\left(\frac{1}{\sqrt{\log(d)}}\right)d^{-2-c_0}. \qquad (22)$$

The last equality holds because defining $f(d) = d^{-2-c_0}$ and $d$ being sufficiently large,

$$g(d) := |(d-1)\log(1 - f(d))| \leq (d-1) \times (f(d) + O(f(d)^2)) \leq O(d^{-1}) \qquad (23)$$

which means

$$(1 - f(d))^{d-1} = e^{-g(d)} \in (1 - O(d^{-1}), 1) \qquad (24)$$

2. Given $\boldsymbol{v} \in \mathcal{V}$, $|S_+^{*(0)}(\boldsymbol{v})|$ is a binomial random variable, with each Bernoulli trial (ranging over $r \in [m]$) having success probability $p_1(1 - p_2)^{d-1}$. Therefore, $\mathbb{E}\left[|S_+^{*(0)}(\boldsymbol{v})|\right] = mp_1(1 - p_2)^{d-1} = \Theta\left(\frac{1}{\sqrt{\log(d)}}\right)d^{c_0}$.

   Now recall the Chernoff bound of binomial random variables. Let $\{X_n\}_{n=1}^m$ be an iid sequence of Bernoulli random variable with success rate $p$, and $S_n = \sum_{n=1}^m X_n$. Then for any $\delta \in (0, 1)$,

$$\mathbb{P}[S_n \geq (1 + \delta)mp] \leq \exp\left(-\frac{\delta^2 mp}{3}\right)$$

$$\mathbb{P}[S_n \leq (1 - \delta)mp] \leq \exp\left(-\frac{\delta^2 mp}{2}\right) \qquad (25)$$

It follows that, for each $\boldsymbol{v} \in \mathcal{V}$, $|S_+^{*(0)}(\boldsymbol{v})| = \Theta\left(\frac{1}{\sqrt{\log(d)}}\right) d^{c_0}$ with probability at least $1 - \exp(-\Omega(\log^{-1/2}(d))d^{c_0})$. Taking union bound over all possible $\boldsymbol{v} \in \mathcal{D}$, the random event still holds with probability at least $1 - \exp(-\Omega(\log^{-1/2}(d))d^{c_0} + \mathcal{O}(\log(d))) \geq 1 - \exp(-\Omega(d^{0.5c_0}))$ (in sufficiently high dimension).

The proof for $S_+^{(0)}(\boldsymbol{v})$ proceeds in virtually the same way, so we omit the calculations here.

To show the second point, in particular $\left|\frac{|S_+^{*(0)}(\boldsymbol{v})|}{|S_+^{(0)}(\boldsymbol{v}')|} - 1\right| \leq O\left(\frac{1}{\log^5(d)}\right)$, we need to be a bit more careful in our bounds of the relevant sets. In particular, we need to directly use the CDF of gaussian random variables:

$$
\begin{aligned}
&\left|\mathbb{P}\left[\langle \boldsymbol{w}_{+,r}^{(0)}, \boldsymbol{v}\rangle \geq \sigma_0\sqrt{4 + 2c_0}\sqrt{\log(d) + \frac{1}{\log^5(d)}}\right](1 \pm O(d^{-1}))\right.\\
&\left.\quad - \mathbb{P}\left[\langle \boldsymbol{w}_{+,r}^{(0)}, \boldsymbol{v}'\rangle \geq \sigma_0\sqrt{4 + c_0}\sqrt{\log(d) - \frac{1}{\log^5(d)}}\right]\right|\\
&\leq \frac{1}{2\sqrt{2\pi}}\int_{\sqrt{4+2c_0}\sqrt{\log(d)-\frac{1}{\log^5(d)}}}^{\sqrt{4+2c_0}\sqrt{\log(d)+\frac{1}{\log^5(d)}}} e^{-\epsilon^2/2}d\epsilon + O\left(\frac{1}{d^{3+c_0}\sqrt{\log(d)}}\right)\\
&\leq \frac{1}{2\sqrt{2\pi}}d^{-2-c_0}e^{(2+c_0)/\log^5(d)}\sqrt{4 + 2c_0}\left(\sqrt{\log(d) + \frac{1}{\log^5(d)}} - \sqrt{\log(d) - \frac{1}{\log^5(d)}}\right)\\
&\quad + O\left(\frac{1}{d^{3+c_0}\sqrt{\log(d)}}\right)\\
&= \frac{1}{2\sqrt{2\pi}}d^{-2-c_0}e^{(2+c_0)/\log^5(d)}\sqrt{4 + 2c_0}\frac{\frac{2}{\log^5(d)}}{\sqrt{\log(d) + \frac{1}{\log^5(d)}} + \sqrt{\log(d) - \frac{1}{\log^5(d)}}} + O\left(\frac{1}{d^{3+c_0}\sqrt{\log(d)}}\right)
\end{aligned}
\tag{26}
$$

The expected difference in number between the two sets is just the above expression multiplied by $m = \Theta(d^{2+2c_0})$, and with probability at least $1 - \exp(-\Omega(d^{-c_0/4}))$, the difference term satisfies

$$
\begin{aligned}
&\frac{1}{2\sqrt{2\pi}}(1 \pm d^{-c_0/2})\Theta(d^{c_0})e^{(2+c_0)/\log^5(d)}\sqrt{4 + 2c_0}\frac{\frac{2}{\log^5(d)}}{\sqrt{\log(d) + \frac{1}{\log^5(d)}} + \sqrt{\log(d) - \frac{1}{\log^5(d)}}}\\
&\pm O\left(\frac{d^{2+2c_0}}{d^{3+c_0}\sqrt{\log(d)}}\right)\\
&\in \Theta\left(\frac{1}{\sqrt{\log(d)}}\right)d^{c_0} \times \frac{1}{\log^5(d)}
\end{aligned}
\tag{27}
$$

By further noting from before that $|S_+^{(0)}(\boldsymbol{v})| = \Theta\left(\frac{1}{\sqrt{\log(d)}}\right)d^{c_0}$, $\left|\frac{|S_+^{*(0)}(\boldsymbol{v})|}{|S_+^{(0)}(\boldsymbol{v}')|} - 1\right| \leq O\left(\frac{1}{\log^5(d)}\right)$ follows. The proof of $\left|\frac{|S_+^{*(0)}(\boldsymbol{v})|}{|S_+^{*(0)}(\boldsymbol{v}')|} - 1\right| \leq O\left(\frac{1}{\log^5(d)}\right)$ follows a very similar argument, so we omit the calculations here.

Now, as for the set $S_{reg}^{(0)}$, we know for any $r \in [m]$ and $\boldsymbol{v}_i \in \mathcal{D}$,

$$
\mathbb{P}\left[\langle \boldsymbol{w}_{+,r}^{(0)}, \boldsymbol{v}_i\rangle \geq \sigma_0\sqrt{10}\sqrt{\log(d)}\right] \leq O\left(\frac{1}{\sqrt{\log(d)}}\right)d^{-5}.
\tag{28}
$$

Taking the union bound over $r$ and $i$ yields

$$
\mathbb{P}\left[\exists r \text{ and } i \text{ s.t.} \langle \boldsymbol{w}_{+,r}^{(0)}, \boldsymbol{v}_i\rangle \geq \sigma_0\sqrt{10}\sqrt{\log(d)}\right] \leq mdO\left(\frac{1}{\sqrt{\log(d)}}\right)d^{-5} < d^{-2}.
\tag{29}
$$

Finally, to show $\left|\mathcal{U}_{+,r}^{(0)}\right| \leq O(1)$ holds for every $(+,r)$, we just need to note that for any arbitrary $(+,r)$ neuron, the probability of $\left|\mathcal{U}_{+,r}^{(0)}\right| > 4$ is no greater than

$$p_2^4 \binom{d}{4} \leq O\left(\frac{1}{\log^2 d}\right) d^{-8-4c_0} \times d^4 \leq O\left(\frac{1}{\log^2 d}\right) d^{-4-4c_0} \tag{30}$$

Taking union bound over all $m \leq O\left(d^{2+2c_0}\right)$ neurons yields the desired result.

$\square$

# D  Coarse-grained SGD Phase I: (Almost) Constant Loss, Neurons Diversify

**Definition D.1.** We define $T_0$ to be the first time which there exists some sample $n$ such that

$$F_c^{(T_0)}(\boldsymbol{X}_n^{(T_0)}) \geq d^{-1} \tag{31}$$

Without loss of generality assume $c = +$. Define phase I to be the time $t \in [0, T_0)$.

## D.1  Main results

**Theorem D.1** (Phase 1 SGD update properties)**.** *The following properties hold with probability at least* $1 - O\left(\frac{mNPk_+t}{poly(d)}\right) - O(e^{-\Omega(\log^2(d))})$ *for every* $t \in [0, T_0)$.

1. *(On-diagonal common-feature neuron growth) For every* $(+,r),(+,r') \in S_+^{*(0)}(\boldsymbol{v}_+)$,

$$\boldsymbol{w}_{+,r}^{(t)} - \boldsymbol{w}_{+,r}^{(0)} = \boldsymbol{w}_{+,r'}^{(t)} - \boldsymbol{w}_{+,r'}^{(0)} \tag{32}$$

   *Moreover,*

$$\Delta\boldsymbol{w}_{+,r}^{(t)} = \eta\left(\left(\frac{1}{2} \pm \psi_1\right)\sqrt{1 \pm \iota}\left(1 \pm s^{*-1/3}\right) \pm O\left(\frac{1}{\log^{10}(d)}\right)\right)\frac{s^*}{2P}\boldsymbol{v}_+ + \Delta\boldsymbol{\zeta}_{+,r}^{(t)} \tag{33}$$

   *where* $\Delta\boldsymbol{\zeta}_{+,r}^{(t)} \sim \mathcal{N}(\boldsymbol{0}, \sigma_{\Delta\zeta_{+,r}}^{(t)2}\boldsymbol{I})$, $\sigma_{\Delta\zeta_{+,r}}^{(t)} = \eta\sigma_\zeta\left(\left(\frac{1}{2} \pm \psi_1\right)\sqrt{1 \pm s^{*-1/3}}\right)\frac{\sqrt{s^*}}{P\sqrt{2N}}$, *and* $|\psi_1| \leq d^{-1}$.

   *Furthermore, every* $(+,r) \in S_+^{*(0)}(\boldsymbol{v}_+)$ *activates on* $\boldsymbol{v}_+$-*dominated patches at time* $t$.

2. *(On-diagonal finegrained-feature neuron growth) For every possible choice of* $c$ *and every* $(+,r),(+,r') \in S_+^{*(0)}(\boldsymbol{v}_{+,c})$,

$$\boldsymbol{w}_{+,r}^{(t)} - \boldsymbol{w}_{+,r}^{(0)} = \boldsymbol{w}_{+,r'}^{(t)} - \boldsymbol{w}_{+,r'}^{(0)} \tag{34}$$

   *Moreover,*

$$\Delta\boldsymbol{w}_{+,r}^{(t)} = \eta\left(\left(\frac{1}{2} \pm \psi_1\right)\sqrt{1 \pm \iota}\left(1 \pm s^{*-1/3}\right) \pm O\left(\frac{1}{\log^{10}(d)}\right)\right)\frac{s^*}{2k_+P}\boldsymbol{v}_{+,c} + \Delta\boldsymbol{\zeta}_{+,r}^{(t)} \tag{35}$$

   *where* $\boldsymbol{\zeta}_{+,r}^{(t)} \sim \mathcal{N}(\boldsymbol{0}, \sigma_{\Delta\zeta_{+,r}}^{(t)2}\boldsymbol{I})$, *and* $\sigma_{\Delta\zeta_{+,r}}^{(t)} = \eta\sigma_\zeta\left(\left(\frac{1}{2} \pm \psi_1\right)\sqrt{1 \pm s^{*-1/3}}\right)\frac{\sqrt{s^*}}{P\sqrt{2Nk_+}}$.

   *Furthermore, every* $(+,r) \in S_+^{*(0)}(\boldsymbol{v}_{+,c})$ *activates on* $\boldsymbol{v}_+$-*dominated patches at time* $t$.

3. *The above results also hold with the "+" and "−" signs flipped.*

*Proof.* The SGD update rule produces the following update:

$$\boldsymbol{w}_{+,r}^{(t+1)} = \boldsymbol{w}_{+,r}^{(t)} + \eta\frac{1}{NP} \times \tag{36}$$

$$\sum_{n=1}^{N}\left(\mathbb{1}\{y_n = +\}[1 - \text{logit}_+^{(t)}(\boldsymbol{X}_n^{(t)})]\sum_{p\in[P]}\sigma'(\langle\boldsymbol{w}_{+,r}^{(t)}, \boldsymbol{x}_{n,p}^{(t)}\rangle + b_{+,r}^{(t)})\boldsymbol{x}_{n,p}^{(t)}\right. \tag{37}$$

$$\left. + \mathbb{1}\{y_n = -\}[-\text{logit}_+^{(t)}(\boldsymbol{X}_n^{(t)})]\sum_{p\in[P]}\sigma'(\langle\boldsymbol{w}_{+,r}^{(t)}, \boldsymbol{x}_{n,p}^{(t)}\rangle + b_{+,r}^{(t)})\boldsymbol{x}_{n,p}^{(t)}\right) \tag{38}$$

In particular,

$$
\begin{aligned}
equation\ 37 =& \sum_{n=1}^{N} \mathbb{1}\{y_n = +\} \left(\frac{1}{2} \pm \psi_1\right) \times \\
& \left\{ \mathbb{1}\{|\mathcal{P}(\boldsymbol{X}_n^{(t)}; \boldsymbol{v}_+)| > 0\} \left[ \sum_{p \in \mathcal{P}(\boldsymbol{X}_n^{(t)}; \boldsymbol{v}_+)} \sigma'(\langle \boldsymbol{w}_{+,r}^{(t)}, \alpha_{n,p}^{(t)} \boldsymbol{v}_+ + \boldsymbol{\zeta}_{n,p}^{(t)}\rangle + b_{+,r}^{(t)}) \left(\alpha_{n,p}^{(t)} \boldsymbol{v}_+ + \boldsymbol{\zeta}_{n,p}^{(t)}\right) \right.\right. \\
& + \left. \sum_{p \notin \mathcal{P}(\boldsymbol{X}_n^{(t)}; \boldsymbol{v}_+)} \sigma'(\langle \boldsymbol{w}_{+,r}^{(t)}, \boldsymbol{x}_{n,p}^{(t)}\rangle + b_{+,r}^{(t)})\boldsymbol{x}_{n,p}^{(t)} \right] \\
& + \left. \mathbb{1}\{|\mathcal{P}(\boldsymbol{X}_n^{(t)}; \boldsymbol{v}_+)| = 0\} \sum_{p \in [P]} \sigma'(\langle \boldsymbol{w}_{+,r}^{(t)}, \boldsymbol{x}_{n,p}^{(t)}\rangle + b_{+,r}^{(t)})\boldsymbol{x}_{n,p}^{(t)} \right\} \\
=& \sum_{n=1}^{N} \mathbb{1}\{y_n = +\} \left(\frac{1}{2} \pm \psi_1\right) \times \\
& \left\{ \mathbb{1}\{|\mathcal{P}(\boldsymbol{X}_n^{(t)}; \boldsymbol{v}_+)| > 0\} \left[ \sum_{p \in \mathcal{P}(\boldsymbol{X}_n^{(t)}; \boldsymbol{v}_+)} \mathbb{1}\left\{\langle \boldsymbol{w}_{+,r}^{(t)}, \alpha_{n,p}^{(t)} \boldsymbol{v}_+ + \boldsymbol{\zeta}_{n,p}^{(t)}\rangle \geq b_{+,r}^{(t)}\right\} \left(\alpha_{n,p}^{(t)} \boldsymbol{v}_+ + \boldsymbol{\zeta}_{n,p}^{(t)}\right) \right.\right. \\
& + \left. \sum_{p \notin \mathcal{P}(\boldsymbol{X}_n^{(t)}; \boldsymbol{v}_+)} \mathbb{1}\left\{\langle \boldsymbol{w}_{+,r}^{(t)}, \boldsymbol{x}_{n,p}^{(t)}\rangle \geq b_{+,r}^{(t)}\right\} \boldsymbol{x}_{n,p}^{(t)} \right] \\
& + \left. \mathbb{1}\{|\mathcal{P}(\boldsymbol{X}_n^{(t)}; \boldsymbol{v}_+)| = 0\} \sum_{p \in [P]} \mathbb{1}\left\{\langle \boldsymbol{w}_{+,r}^{(t)}, \boldsymbol{x}_{n,p}^{(t)}\rangle \geq b_{+,r}^{(t)}\right\} \boldsymbol{x}_{n,p}^{(t)} \right\}
\end{aligned}
\tag{39}
$$

The rest of the proof proceeds by induction (in Phase 1).

First, recall that we set $b_{c,r}^{(0)} = -\sqrt{4+2c_0}\sqrt{\log(d)}$, and $\Delta b_{c,r}^{(t)} = -\frac{\|\Delta \boldsymbol{w}_{c,r}^{(t)}\|_2}{\log^5(d)}$ for all $t$ in phase 1, and for any $+$-class sample $\boldsymbol{X}_n$ with $p \in \mathcal{P}(\boldsymbol{X}_n^{(t)}; \boldsymbol{v}_+)$, $\alpha_{n,p}^{(t)} \in \sqrt{1 \pm \iota}$ by our data assumption.

**Base case** $t = 0$.

*1. (On-diagonal common-feature neuron growth)*

The base case for the neuron expression of point 1. is trivially true.

We show that the neurons $(+, r) \in S_+^{*(0)}(\boldsymbol{v}_+)$ only activate on $\boldsymbol{v}_+$-dominated patches at time $t = 0$.

With probability at least $1 - O\left(\frac{mNP}{\text{poly}(d)}\right)$, by Lemma H.3, we have for all possible choices of $r, n, p$:

$$
\left|\langle \boldsymbol{w}_{+,r}^{(0)}, \boldsymbol{\zeta}_{n,p}^{(0)}\rangle\right| \leq O(\sigma_0 \sigma_\zeta \sqrt{d \log(d)}) \leq O\left(\frac{\sigma_0}{\log^9(d)}\right)
\tag{40}
$$

It follows that

$$
\begin{aligned}
& \langle \boldsymbol{w}_{+,r}^{(0)}, \alpha_{n,p}^{(0)} \boldsymbol{v}_+ + \boldsymbol{\zeta}_{n,p}^{(0)}\rangle \\
=& \sigma_0 \left\{ \sqrt{1 \pm \iota} \times \left(\sqrt{4+2c_0}\sqrt{\log(d) + 1/\log^5(d)}, \sqrt{10}\sqrt{\log(d)}\right) \pm \frac{1}{\log^9(d)} \right\} \\
=& \sigma_0 \left\{ \left(\sqrt{1-\iota}\sqrt{4+2c_0}\sqrt{\log(d) + 1/\log^5(d)}, \sqrt{1+\iota}\sqrt{10}\sqrt{\log(d)}\right) \pm \frac{1}{\log^9(d)} \right\}
\end{aligned}
\tag{41}
$$

Employing the basic identity $a - b = \frac{a^2 - b^2}{a+b}$, we have the lower bound

$$\sigma_0^{-1} \left( \langle \boldsymbol{w}_{+,r}^{(0)}, \alpha_{n,p}^{(0)} \boldsymbol{v}_+ + \boldsymbol{\zeta}_{n,p}^{(0)} \rangle + b_{+,r}^{(0)} \right)$$

$$\geq \sqrt{(1-\iota)(4+2c_0)(\log(d) + 1/\log^5(d))} - \sqrt{(4+2c_0)\log(d)} - O\left( \frac{1}{\log^9(d)} \right)$$

$$= \frac{(1-\iota)(4+2c_0)(\log(d) + 1/\log^5(d)) - (4+2c_0)\log(d)}{\sqrt{(1-\iota)(4+2c_0)(\log(d) + 1/\log^5(d))} + \sqrt{(4+2c_0)\log(d)}} - O\left( \frac{1}{\log^9(d)} \right) \tag{42}$$

$$= \frac{(4+2c_0)(-\iota\log(d) + (1-\iota)/\log^5(d))}{\sqrt{(1-\iota)(4+2c_0)(\log(d) + 1/\log^5(d))} + \sqrt{(4+2c_0)\log(d)}} - O\left( \frac{1}{\log^9(d)} \right)$$

$$> 0$$

The last inequality holds since $\iota \leq \frac{1}{\mathrm{polylog}(d)}$ and $d$ is sufficiently large such that $\frac{1}{\log^9(d)}$ does not drive the positive term down past 0.

Therefore, the neurons in $S_+^{*(0)}(\boldsymbol{v}_+)$ indeed activate on the $\boldsymbol{v}_+$-dominated patches at $t = 0$.

The rest of the patches $\boldsymbol{x}_{n,p}^{(0)}$ is either a feature patch (not dominated by $\boldsymbol{v}_+$) or a noise patch. By definition, $(+,r) \in S_+^{*(0)}(\boldsymbol{v}_+) \implies (+,r) \in S_+^{(0)}(\boldsymbol{v}_+)$. Therefore, by Theorem F.1, with probability at least $1 - O\left( \frac{mk_+ NP}{\mathrm{poly}(d)} \right)$, at time $t = 0$, the $(+,r) \in S_+^{*(0)}(\boldsymbol{v}_+)$ neurons we are considering cannot activate on any feature patch dominated by $\boldsymbol{v} \perp \boldsymbol{v}_+$, nor on any noise patches.

It follows that the expression equation 37 at time $t = 0$ is as follows:

$$\text{equation } 37 = \sum_{n=1}^{N} \mathbb{1}\{y_n = +\} \left( \frac{1}{2} \pm \psi_1 \right) \times$$

$$\left\{ \mathbb{1}\{|\mathcal{P}(\boldsymbol{X}_n^{(0)}; \boldsymbol{v}_+)| > 0\} \left[ \sum_{p \in \mathcal{P}(\boldsymbol{X}_n^{(0)}; \boldsymbol{v}_+)} \left( \sqrt{1 \pm \iota} \boldsymbol{v}_+ + \boldsymbol{\zeta}_{n,p}^{(0)} \right) + \sum_{p \notin \mathcal{P}(\boldsymbol{X}_n^{(0)}; \boldsymbol{v}_+)} 0 \right] \right.$$

$$\left. + \mathbb{1}\{|\mathcal{P}(\boldsymbol{X}_n^{(0)}; \boldsymbol{v}_+)| = 0\} \sum_{p \in [P]} 0 \right\}$$

$$= \left( \frac{1}{2} \pm \psi_1 \right) \sum_{n=1}^{N} \mathbb{1}\{y_n = +, |\mathcal{P}(\boldsymbol{X}_n^{(0)}; \boldsymbol{v}_+)| > 0\} \sum_{p \in \mathcal{P}(\boldsymbol{X}_n^{(0)}; \boldsymbol{v}_+)} \left( \sqrt{1 \pm \iota} \boldsymbol{v}_+ + \boldsymbol{\zeta}_{n,p}^{(0)} \right) \tag{43}$$

$$= \left( \frac{1}{2} \pm \psi_1 \right) \times$$

$$\left| \left\{ (n,p) \in [N] \times [P] : y_n = +, |\mathcal{P}(\boldsymbol{X}_n^{(0)}; \boldsymbol{v}_+)| > 0, p \in \mathcal{P}(\boldsymbol{X}_n^{(0)}; \boldsymbol{v}_+) \right\} \right| \left( \sqrt{1 \pm \iota} \boldsymbol{v}_+ \right)$$

$$+ \sum_{n=1}^{N} \sum_{p \in \mathcal{P}(\boldsymbol{X}_n^{(0)}; \boldsymbol{v}_+)} \{y_n = +\} \left( \frac{1}{2} \pm \psi_1 \right) \boldsymbol{\zeta}_{n,p}^{(0)}$$

On average,

$$\mathbb{E}\left[ \left| \left\{ (n,p) \in [N] \times [P] : y_n = +, |\mathcal{P}(\boldsymbol{X}_n^{(0)}; \boldsymbol{v}_+)| > 0, p \in \mathcal{P}(\boldsymbol{X}_n^{(0)}; \boldsymbol{v}_+) \right\} \right| \right]$$

$$= \frac{s^*}{P} \times P \times \frac{N}{2} = \frac{s^* N}{2} \tag{44}$$

Furthermore, with our parameter choices, and by concentration of binomial random variables, with probability at least $1 - e^{-\Omega(\mathrm{polylog}(d))}$,

$$\left| \left\{ (n,p) \in [N] \times [P] : y_n = +, |\mathcal{P}(\boldsymbol{X}_n^{(0)}; \boldsymbol{v}_+)| > 0, p \in \mathcal{P}(\boldsymbol{X}_n^{(0)}; \boldsymbol{v}_+) \right\} \right| = \frac{s^* N}{2} \left( 1 \pm s^{*-1/3} \right) \tag{45}$$

must be true.

It follows that

$$
\begin{aligned}
\text{equation } 37 = &\left(\frac{1}{2} \pm \psi_1\right) \times \frac{s^* N}{2}\left(1 \pm s^{*-1/2}\right) \times \left(\sqrt{1 \pm \iota} \boldsymbol{v}_+\right) \\
&+ \sum_{n=1}^{N} \sum_{p \in \mathcal{P}(\boldsymbol{X}_n^{(0)}; \boldsymbol{v}_+)} \{y_n = +\} \left(\frac{1}{2} \pm \psi_1\right) \boldsymbol{\zeta}_{n,p}^{(0)}
\end{aligned}
\tag{46}
$$

The other component expression equation 38 is zero with probability at least $1 - O\left(\frac{mk_+ NP}{\text{poly}(d)}\right)$ by Theorem F.1.

By noting that

$$
\begin{aligned}
\text{Var}\left(\Delta \boldsymbol{\zeta}_{+,r}^{(0)}\right) =& \text{Var}\left(\frac{\eta}{NP} \sum_{n=1}^{N} \sum_{p \in \mathcal{P}(\boldsymbol{X}_n^{(0)}; \boldsymbol{v}_+)} \{y_n = +\} \left(\frac{1}{2} \pm \psi_1\right) \boldsymbol{\zeta}_{n,p}^{(0)}\right) \\
=& \eta^2 \left(\frac{1}{2} \pm \psi_1\right)^2 \frac{s^*}{2NP^2}\left(1 \pm s^{*-1/3}\right) \sigma_\zeta^2,
\end{aligned}
\tag{47}
$$

and

$$
\mathbb{E}\left[\Delta \boldsymbol{\zeta}_{+,r}^{(0)}\right] = \mathbb{E}\left[\frac{\eta}{NP} \sum_{n=1}^{N} \sum_{p \in \mathcal{P}(\boldsymbol{X}_n^{(0)}; \boldsymbol{v}_+)} \{y_n = +\} \left(\frac{1}{2} \pm \psi_1\right) \boldsymbol{\zeta}_{n,p}^{(0)}\right] = \boldsymbol{0},
\tag{48}
$$

we finish the proof of the base case for point 1.

*2. (On-diagonal finegrained-feature neuron growth)*

The proof of the base case of point 2. is virtually identical to point 1, so we omit the computations here.

**Inductive step**: We condition on the high probability events of the induction hypothesis for $t \in [0, T]$ (with $T < T_0$ of course), and prove the statements for $t = T + 1$.

*1. (On-diagonal common-feature neuron growth)*

By the induction hypothesis, up to time $t = T$, with probability at least $1 - O\left(\frac{mk_+ NPT}{\text{poly}(d)}\right)$, for all $(+, r) \in S_+^{*(T)}(\boldsymbol{v}_+)$,

$$
\Delta \boldsymbol{w}_{+,r}^{(t)} = \eta \left(\left(\frac{1}{2} \pm \psi_1\right) \sqrt{1 \pm \iota}\left(1 \pm s^{*-1/3}\right)\right) \frac{s^*}{2P} \boldsymbol{v}_+ + \Delta \boldsymbol{\zeta}_{+,r}^{(t)}
\tag{49}
$$

where $\Delta \boldsymbol{\zeta}_{+,r}^{(t)} \sim \mathcal{N}(\boldsymbol{0}, \sigma_{\Delta\zeta}^{(t)2} \boldsymbol{I})$, $\sigma_{\Delta\zeta}^{(t)} = \eta \sigma_\zeta \left(\left(\frac{1}{2} \pm \psi_1\right) \sqrt{1 \pm s^{*-1/3}}\right) \frac{\sqrt{s^*}}{P\sqrt{2N}}$.

**Expression of $\boldsymbol{w}_{+,r}^{(T+1)}$.**

Conditioning on the high-probability event of the induction hypothesis, at time $t = T + 1$,

$$
\begin{aligned}
\boldsymbol{w}_{+,r}^{(T+1)} =& \boldsymbol{w}_{+,r}^{(0)} + \sum_{\tau=0}^{T} \Delta \boldsymbol{w}_{+,r}^{(\tau)} \\
=& \eta T \left(\left(\frac{1}{2} \pm \psi_1\right) \sqrt{1 \pm \iota}\left(1 \pm s^{*-1/3}\right)\right) \frac{s^*}{2P} \boldsymbol{v}_+ + \boldsymbol{\zeta}_{+,r}^{(t)}
\end{aligned}
\tag{50}
$$

where $\boldsymbol{\zeta}_{+,r}^{(t)} \sim \mathcal{N}(\boldsymbol{0}, \sigma_\zeta^{(t)2} \boldsymbol{I})$, $\sigma_\zeta^{(t)} = \eta \sigma_\zeta \sqrt{T} \left(\left(\frac{1}{2} \pm \psi_1\right) \sqrt{1 \pm s^{*-1/3}}\right) \frac{\sqrt{s^*}}{P\sqrt{2N}}$.

Let us compute $\Delta \boldsymbol{w}_{+,r}^{(T+1)}$.

We first want to show that $\boldsymbol{w}_{+,r}^{(T+1)}$ activates on $\boldsymbol{v}_+$-dominated patches $\boldsymbol{x}_{n,p}^{(T+1)} = \sqrt{1 \pm \iota}\boldsymbol{v}_+ + \boldsymbol{\zeta}_{n,p}^{(T+1)}$. We need to show that the following expression is above 0:

$$
\begin{aligned}
&\langle \boldsymbol{w}_{+,r}^{(T+1)}, \boldsymbol{x}_{n,p}^{(T+1)}\rangle + b_{+,r}^{(T+1)} \\
=&\langle \boldsymbol{w}_{+,r}^{(0)}, \sqrt{1 \pm \iota}\boldsymbol{v}_+ + \boldsymbol{\zeta}_{n,p}^{(T+1)}\rangle + b_{+,r}^{(0)} \\
&+ \left\langle \eta T\left( \left(\frac{1}{2} \pm \psi_1\right)\sqrt{1 \pm \iota}\left(1 \pm s^{*-1/3}\right) \pm O\left(\frac{1}{\log^{10}(d)}\right)\right)\frac{s^*}{2P}\boldsymbol{v}_+ + \boldsymbol{\zeta}_{+,r}^{(T+1)}, \sqrt{1 \pm \iota}\boldsymbol{v}_+ + \boldsymbol{\zeta}_{n,p}^{(T+1)}\right\rangle \quad (51) \\
&+ \sum_{\tau=0}^{T}\Delta b_{+,r}^{(\tau)}
\end{aligned}
$$

Let us treat the three terms (on three lines) separately.

First, following virtually the same argument as in the base case, the following lower bound holds with probability at least $1 - O\left(\frac{mNP}{\mathrm{poly}(d)}\right)$ for all $n, p$ and $(+, r) \in S_+^{*(T)}(\boldsymbol{v}_+)$:

$$
\begin{aligned}
&\langle \boldsymbol{w}_{+,r}^{(0)}, \sqrt{1 \pm \iota}\boldsymbol{v}_+ + \boldsymbol{\zeta}_{n,p}^{(T+1)}\rangle + b_{+,r}^{(0)} \\
\geq&\sigma_0\left\{\sqrt{(1-\iota)(4+2c_0)(\log(d) + 1/\log^5(d))} - \sqrt{(4+2c_0)\log(d)} - O\left(\frac{1}{\log^9(d)}\right)\right\} \quad (52) \\
>&0
\end{aligned}
$$

Now consider the second term.

We know, with probability at least $1 - e^{-\Omega(d)}$, for all $n$ and $p$,

$$
\left|\langle \boldsymbol{\zeta}_{n,p}^{(T+1)}, \boldsymbol{v}_+\rangle\right| \leq O\left(\frac{1}{\log^{10}(d)}\right), \quad (53)
$$

therefore,

$$
\begin{aligned}
&\left|\langle \eta T\left( \left(\frac{1}{2} \pm \psi_1\right)\sqrt{1 \pm \iota}\left(1 \pm s^{*-1/3}\right) \pm O\left(\frac{1}{\log^{10}(d)}\right)\right)\frac{s^*}{2P}\boldsymbol{v}_+, \boldsymbol{\zeta}_{n,p}^{(T+1)}\rangle\right| \\
\leq&\eta T\frac{s^*}{2P}O\left(\frac{1}{\log^{10}(d)}\right). 
\end{aligned} \quad (54)
$$

Moreover, with probability at least $1 - e^{-\Omega(d)}$,

$$
\left|\langle \boldsymbol{\zeta}_{+,r}^{(T+1)}, \boldsymbol{v}_+\rangle\right| \leq \eta\sqrt{T}\frac{\sqrt{s^*}}{P\sqrt{2N}} \times O\left(\frac{1}{\log^{10}(d)}\right) \quad (55)
$$

and with probability at least $1 - e^{-\Omega(d)}$,

$$
\left|\langle \boldsymbol{\zeta}_{+,r}^{(T)}, \boldsymbol{\zeta}_{n,p}^{(T+1)}\rangle\right| \leq O\left(\sigma_\zeta\sigma_\zeta^{(T)}d\right) \leq O\left(\eta\sqrt{T}\frac{\sqrt{s^*}}{P\sqrt{2N}}\frac{1}{\log^{20}(d)d}d\right) \leq \eta\sqrt{T}\frac{\sqrt{s^*}}{P\sqrt{2N}}\frac{1}{\log^{19}(d)} \quad (56)
$$

therefore

$$
\langle \eta T\boldsymbol{\zeta}_{+,r}^{(T+1)}, \sqrt{1 \pm \iota}\boldsymbol{v}_+ + \boldsymbol{\zeta}_{n,p}^{(T+1)}\rangle \leq \eta\sqrt{T}\frac{\sqrt{s^*}}{P\sqrt{2N}}O\left(\frac{1}{\log^{10}(d)}\right). \quad (57)
$$

It follows that with probability at least $1 - O(e^{-\Omega(d)})$,

$$
\begin{aligned}
&\left\langle \eta T\left( \left(\frac{1}{2} \pm \psi_1\right) \sqrt{1 \pm \iota}\left(1 \pm s^{*-1/3}\right) \right) \frac{s^*}{2P}\boldsymbol{v}_+ + \boldsymbol{\zeta}_{+,r}^{(T+1)}, \sqrt{1 \pm \iota}\boldsymbol{v}_+ + \boldsymbol{\zeta}_{n,p}^{(T+1)} \right\rangle \\
&= \langle \eta T\left( \left(\frac{1}{2} \pm \psi_1\right) \sqrt{1 \pm \iota}\left(1 \pm s^{*-1/3}\right) \right) \frac{s^*}{2P}\boldsymbol{v}_+, \sqrt{1 \pm \iota}\boldsymbol{v}_+ \rangle \\
&\quad + \langle \eta T\left( \left(\frac{1}{2} \pm \psi_1\right) \sqrt{1 \pm \iota}\left(1 \pm s^{*-1/3}\right) \right) \frac{s^*}{2P}\boldsymbol{v}_+, \boldsymbol{\zeta}_{n,p}^{(T+1)} \rangle \\
&\quad + \langle \eta\boldsymbol{\zeta}_{+,r}^{(T+1)}, \sqrt{1 \pm \iota}\boldsymbol{v}_+ + \boldsymbol{\zeta}_{n,p}^{(T+1)} \rangle \\
&\geq \eta T\left(\frac{1}{2} - \psi_1^{(T+1)}\right)(1 - \iota)\left(1 - s^{*-1/3}\right)\frac{s^*}{2P} - \eta\sqrt{T}\frac{\sqrt{s^*}}{P\sqrt{2N}}O\left(\frac{1}{\log^{10}(d)}\right).
\end{aligned}
$$
(58)

Now we compute the third term. By the induction hypothesis,

$$
\begin{aligned}
&\sum_{t=0}^{T} \Delta b_{+,r}^{(t)} \\
&= \sum_{t=0}^{T} \frac{\|\Delta\boldsymbol{w}_{+,r}^{(t)}\|_2}{\log^5(d)} \\
&= \sum_{t=0}^{T} \frac{1}{\log^5(d)} \left\| \eta\left(\frac{1}{2} \pm \psi_1\right)\sqrt{1 \pm \iota}\left(1 \pm s^{*-1/3}\right)\frac{s^*}{2P}\boldsymbol{v}_+ + \Delta\boldsymbol{\zeta}_{+,r}^{(t)} \right\|_2 \\
&\leq \sum_{t=0}^{T} \frac{1}{\log^5(d)}\eta\left(\frac{1}{2} + \psi_1\right)\sqrt{1 + \iota}\left(1 + s^{*-1/3}\right)\frac{s^*}{2P}\|\boldsymbol{v}_+\|_2 + \sum_{t=0}^{T} \frac{1}{\log^5(d)}\left\|\Delta\boldsymbol{\zeta}_{+,r}^{(t)}\right\|_2 \\
&= \frac{1}{\log^5(d)}\eta T\left(\frac{1}{2} + \psi_1\right)\sqrt{1 + \iota}\left(1 + s^{*-1/3}\right)\frac{s^*}{2P} + \sum_{t=0}^{T} \frac{1}{\log^5(d)}\left\|\Delta\boldsymbol{\zeta}_{+,r}^{(t)}\right\|_2
\end{aligned}
$$
(59)

With probability at least $1 - O\left(\frac{mT}{\text{poly}(d)}\right)$, for all $t \in [0, T]$ and $r$ in consideration,

$$
\left\|\Delta\boldsymbol{\zeta}_{+,r}^{(t)}\right\|_2 \leq \eta\frac{\sqrt{s^*}}{P\sqrt{2N}}O\left(\frac{1}{\log^{10}(d)}\right)
$$
(60)

Therefore,

$$
\begin{aligned}
&\sum_{t=0}^{T} \Delta b_{+,r}^{(t)} \\
&\leq \frac{1}{\log^5(d)}\left(\eta T\left(\frac{1}{2} + \psi_1\right)\sqrt{1 + \iota}\left(1 + s^{*-1/3}\right)\frac{s^*}{2P} + \eta T\frac{\sqrt{s^*}}{P\sqrt{2N}}O\left(\frac{1}{\log^{10}(d)}\right)\right)
\end{aligned}
$$
(61)

Combining our calculations of the three terms from above, we find the following estimate:

$$
\langle \boldsymbol{w}_{+,r}^{(T+1)}, \boldsymbol{x}_{n,p}^{(T+1)} \rangle + b_{+,r}^{(T+1)}
$$

$$
> 0
$$

$$
+ \eta T \left( \frac{1}{2} - \psi_1 \right)(1-\iota)\left(1 - s^{*-1/3}\right)\frac{s^*}{2P} - \eta\sqrt{T}\frac{\sqrt{s^*}}{P\sqrt{2N}}O\left(\frac{1}{\log^{10}(d)}\right)
$$

$$
- \frac{1}{\log^5(d)}\left(\eta T\left(\frac{1}{2}+\psi_1\right)\sqrt{1+\iota}\left(1+s^{*-1/3}\right)\frac{s^*}{2P} + \eta T\frac{\sqrt{s^*}}{P\sqrt{2N}}O\left(\frac{1}{\log^{10}(d)}\right)\right) \tag{62}
$$

$$
> \eta T\left(\left(\frac{1}{2}-\psi_1\right)(1-\iota)\left(1-s^{*-1/3}\right) - O\left(\frac{1}{\log^4(d)}\right)\right)\frac{s^*}{2P}
$$

$$
> 0
$$

On the other hand, by Theorem F.1, with probability at least $1 - O\left(\frac{mk_+NPT}{\text{poly}(d)}\right)$, none of the $(+,r) \in S_+^{*(T)}(\boldsymbol{v}_+)$ can activate on $\boldsymbol{x}_{n,p}^{(T+1)}$ that are feature-patches dominated by $\boldsymbol{v} \perp \boldsymbol{v}_+$ or noise patches.

Combining the above observations, with probability at least $1 - O\left(\frac{mk_+NP(T+1)}{\text{poly}(d)}\right)$, the update expressions up to time $t = T+1$ can be written as follows:

$$
\Delta \boldsymbol{w}_{+,r}^{(t)} = \left(\frac{1}{2} \pm \psi_1\right)
$$

$$
\times \left\{ \left| \left\{(n,p) \in [N] \times [P] : y_n = +, |\mathcal{P}(\boldsymbol{X}_n^{(t)}; \boldsymbol{v}_+)| > 0, p \in \mathcal{P}(\boldsymbol{X}_n^{(t)}; \boldsymbol{v}_+)\right\} \right| \left(\sqrt{1 \pm \iota}\boldsymbol{v}_+\right) \right. \tag{63}
$$

$$
\left. + \sum_{n=1}^{N} \sum_{p \in \mathcal{P}(\boldsymbol{X}_n^{(0)}; \boldsymbol{v}_+)} \{y_n = +\}\left(\frac{1}{2} \pm \psi_1\right)\zeta_{n,p}^{(t)} \right\}
$$

The rest of the derivations proceeds virtually the same as in the base case; we just need to rely on the concentration of binomial random variables to calculate

$$
\left| \left\{(n,p) \in [N] \times [P] : y_n = +, |\mathcal{P}(\boldsymbol{X}_n^{(0)}; \boldsymbol{v}_+)| > 0, p \in \mathcal{P}(\boldsymbol{X}_n^{(0)}; \boldsymbol{v}_+)\right\} \right| = \frac{s^*N}{2}\left(1 \pm s^{*-1/3}\right) \tag{64}
$$

which completes the proof of the expression of $\Delta \boldsymbol{w}_{+,r}^{(t)}$.

Additionally, to show

$$
\boldsymbol{w}_{+,r}^{(T+1)} - \boldsymbol{w}_{+,r}^{(0)} = \boldsymbol{w}_{+,r'}^{(T+1)} - \boldsymbol{w}_{+,r'}^{(0)} \tag{65}
$$

we just need to note that, by the above sequence of derivations, for every $(+,r) \in S_+^{*(0)}(\boldsymbol{v}_+)$, these neurons receive exactly the same update at time $t = T+1$

$$
\sum_{n=1}^{N} \mathbb{1}\{y_n = +\}\mathbb{1}\{|\mathcal{P}(\boldsymbol{X}_n^{(T+1)}; \boldsymbol{v}_+)| > 0\}[1 - \text{logit}_+^{(T+1)}(\boldsymbol{X}_n^{(T+1)})] \sum_{p \in \mathcal{P}(\boldsymbol{X}_n^{(T+1)}; \boldsymbol{v}_+)} \left(\alpha_{n,p}^{(T+1)}\boldsymbol{v}_+ + \zeta_{n,p}^{(T+1)}\right). \tag{66}
$$

*2. (On-diagonal finegrained-feature neuron growth)*

For point 2, the proof strategy is almost identical, the only difference is that at every iteration, the expected number of patches in which subclass features appear in is

$$
\left| \left\{(n,p) \in [N] \times ([P] - \mathcal{P}(\boldsymbol{X}_n^{(T)}); \boldsymbol{v}_{+,c}) : y_n = +, |\mathcal{P}(\boldsymbol{X}_n^{(T)}; \boldsymbol{v}_{+,c})| > 0, p \in \mathcal{P}(\boldsymbol{X}_n^{(T)}; \boldsymbol{v}_{+,c})\right\} \right|
$$

$$
= \frac{s^*N}{2k_+}\left(1 \pm s^{*-1/3}\right) \tag{67}
$$

which holds with probability at least $1 - e^{-\Omega(\log^2(d))}$ for the relevant neurons. $\qquad \square$

**Corollary D.1.1.** $T_0 < O\left(\left(\eta\frac{s^*}{P}\right)^{-1}\right) \in poly(d)$.

*Proof.* Follows from Theorem D.1. □

### D.2 Lemmas

**Lemma D.2.** *During the time $t \in [0, T_0)$, for any $\boldsymbol{X}_n^{(t)}$,*

$$1 - logit_+^{(t)}(\boldsymbol{X}_n^{(t)}) = \frac{1}{2} \pm O(d^{-1}) \tag{68}$$

*The same holds for $1 - logit_-^{(t)}(\boldsymbol{X}_n^{(t)})$.*

*Therefore, $|\psi_1| \leq O(d^{-1})$ for $t \in [0, T_0)$.*

*Proof.* By definition of $T_0$, for any $t \in [0, T_0]$, we have $F_c^{(t)}(\boldsymbol{X}_n^{(t)}) < d^{-1} + O(\eta)$ for all $n$, therefore, using Taylor approximation,

$$1 - logit_+^{(t)}(\boldsymbol{X}_n^{(t)}) = \frac{\exp(F_-^{(t)}(\boldsymbol{X}_n^{(t)}))}{\exp(F_+^{(t)}(\boldsymbol{X}_n^{(t)})) + \exp(F_-^{(t)}(\boldsymbol{X}_n^{(t)}))} < \frac{\exp(d^{-1})}{1+1} \leq \frac{1}{2} + O(d^{-1}) \tag{69}$$

The lower bound can be proven due to convexity of the exponential:

$$\frac{\exp(F_-^{(t)}(\boldsymbol{X}_n^{(t)}))}{\exp(F_+^{(t)}(\boldsymbol{X}_n^{(t)})) + \exp(F_-^{(t)}(\boldsymbol{X}_n^{(t)}))} > \frac{1}{2}\exp(-d^{-1}) \geq \frac{1}{2} - \frac{1}{2d} \tag{70}$$

□

# E    Coarse-grained SGD Phase II: Loss Convergence, Large Neuron Movement

Recall that the desired probability events in Phase I happens with probability at least $1 - o(1)$.

In phase II, common-feature neurons start gaining large movement and drive the training loss down to $o(1)$. We show that the desired probability events occur with probability at least $1 - o(1)$.

We study the case of $T_1 \leq \text{poly}(d)$, where $T_1$ denotes the time step at the end of training.

## E.1    Main results

**Theorem E.1.** *With probability at least $1 - O\left(\frac{mk_+ NPT_1}{poly(d)}\right)$, the following events take place:*

1. *There exists time $T^* \in poly(d)$ such that for any $t \in [T^*, poly(d)]$, for any $n \in [N]$, the training loss $L(F; \boldsymbol{X}_n^{(t)}, y_n) \in o(1)$.*

2. *(Easy sample test accuracy is nearly perfect) Given an easy test sample $(\boldsymbol{X}_{easy}, y)$, for $y' \in \{+1, -1\} - \{y\}$, for $t \in [T^*, poly(d)]$,*

$$\mathbb{P}\left[F_y^{(t)}(\boldsymbol{X}_{easy}) \leq F_{y'}^{(t)}(\boldsymbol{X}_{easy})\right] \leq o(1). \tag{71}$$

3. *(Hard sample test accuracy is bad) However, for all $t \in [0, poly(d)]$, given a hard test sample $(\boldsymbol{X}_{hard}, y)$,*

$$\mathbb{P}\left[F_y^{(t)}(\boldsymbol{X}_{hard}) \leq F_{y'}^{(t)}(\boldsymbol{X}_{hard})\right] \geq \Omega(1). \tag{72}$$

*Proof.* The training loss property follows from Lemma E.3 and Lemma E.4. We can set $T^* = T_{1,1}$ or any time beyond it (and upper bounded by $\text{poly}(d)$).

The test accuracy properties follow from Lemma E.8 and Lemma E.9.

$\square$

## E.2    Lemmas

**Lemma E.2** (Phase II, Update Expressions). *For any $T_1 \in poly(d)$, with probability at least $1 - O\left(\frac{mNPk_+ t}{poly(d)}\right)$, during $t \in [T_0, T_1]$, for any $(+, r) \in S_+^{*(0)}(\boldsymbol{v}_+)$,*

$$\begin{aligned}
&\Delta \boldsymbol{w}_{+,r}^{(t)} \\
&= \eta \sum_{n=1}^{N} \mathbb{1}\{y_n = +\} \exp\left\{-F_+^{(t)}(\boldsymbol{X}_n^{(t)})\right\} \\
&\quad \times \frac{\exp(F_-^{(t)}(\boldsymbol{X}_n^{(t)}))}{\exp\left(F_-^{(t)}(\boldsymbol{X}_n^{(t)}) - F_+^{(t)}(\boldsymbol{X}_n^{(t)})\right) + 1} (1 \pm s^{*-1/3}) \frac{s^*}{NP}\left(\sqrt{1 \pm \iota} \boldsymbol{v}_+ + \boldsymbol{\zeta}_{n,p}^{(t)}\right),
\end{aligned} \tag{73}$$

*(where $c_n^t$ denotes the subclass index of sample $\boldsymbol{X}_n^{(t)}$) and for any $(+, r) \in S_+^{*(0)}(\boldsymbol{v}_{+,c})$,*

$$\begin{aligned}
&\Delta \boldsymbol{w}_{+,r}^{(t)} \\
&= \eta \exp\left\{-(1 \pm s^{*-1/3})\sqrt{1 \pm \iota}\left(1 \pm O\left(\frac{1}{\log^5(d)}\right)\right) s^*\left(A_{+,r^*}^{*(t)} \left|S_+^{*(0)}(\boldsymbol{v}_+)\right| + A_{+,c,r^*}^{*(t)} \left|S_+^{*(0)}(\boldsymbol{v}_{+,c})\right|\right)\right\} \\
&\quad \times \sum_{n=1}^{N} \mathbb{1}\{y_n = (+,c)\} \frac{\exp(F_-^{(t)}(\boldsymbol{X}_n^{(t)}))}{\exp\left(F_-^{(t)}(\boldsymbol{X}_n^{(t)}) - F_+^{(t)}(\boldsymbol{X}_n^{(t)})\right) + 1}(1 \pm s^{*-1/3}) \frac{s^*}{NP}\left(\sqrt{1 \pm \iota} \boldsymbol{v}_{+,c} + \boldsymbol{\zeta}_{n,p}^{(t)}\right),
\end{aligned} \tag{74}$$

*In fact, for any $\boldsymbol{v} \in \{\boldsymbol{v}_+\} \cup \{\boldsymbol{v}_{+,c}\}_{c=1}^{k_+}$, every neuron in $S_+^{*(0)}(\boldsymbol{v})$ remain activated (on $\boldsymbol{v}$-dominated patches) and receive exactly the same updates at every iteration as shown above.*

*For simpler exposition, for any $(+, r^*) \in S_+^{*(0)}(\boldsymbol{v}_+)$, we write $A_{+,r^*}^{*(t)} := \langle \boldsymbol{w}_{+,r^*}^{(t)}, \boldsymbol{v}_+ \rangle$; similarly for $A_{+,c,r^*}^{*(t)} := \langle \boldsymbol{w}_{+,r^*}, \boldsymbol{v}_{+,c} \rangle$ for neurons $(+, r^*) \in S_+^{*(0)}(\boldsymbol{v}_{+,c})$.*

*Moreover, on "$+$"-class samples, the neural network response satisfies the estimate for every $(+, r^*) \in S_+^{*(0)}(\boldsymbol{v}_+)$:*

$$
\begin{aligned}
&F_+^{(t)}(\boldsymbol{X}_n^{(t)}) \\
&= (1 \pm s^{*-1/3})\sqrt{1 \pm \iota}\left(1 \pm O\left(\frac{1}{\log^5(d)}\right)\right) \times s^*\left(A_{+,r^*}^{*(t)}\left|S_+^{*(0)}(\boldsymbol{v}_+)\right| + A_{+,c_n^t,r^*}^{*(t)}\left|S_+^{*(0)}(\boldsymbol{v}_{+,c_n^t})\right|\right),
\end{aligned}
\tag{75}
$$

*The same claims hold for the "$-$" class neurons (with the class signs flipped).*

*Proof.* In this proof we focus on the neurons in $S_+^{*(0)}(\boldsymbol{v}_+)$; the proof for the update expressions for those in $S_+^{*(0)}(\boldsymbol{v}_{+,c})$ are proven in virtually the same way.

**Base case**, $t = T_0$.

First define $A_{+,r^*}^{*(t)} := \langle \boldsymbol{w}_{+,r^*}, \boldsymbol{v}_+ \rangle$, $(+, r^*) \in S_+^{*(0)}(\boldsymbol{v}_+)$; similarly for $A_{+,c,r^*}^{*(t)} := \langle \boldsymbol{w}_{+,r^*}, \boldsymbol{v}_{+,c} \rangle$. Note that the choice of $r^*$ does not really matter, since we know from phase I that every neuron in $S_+^{*(0)}(\boldsymbol{v}_+)$ evolve at exactly the same rate, so by the end of phase I, $\|\boldsymbol{w}_{+,r}^{(T_0)} - \boldsymbol{w}_{+,r'}^{(T_0)}\|_2 \le O(\sigma_0 \log(d)) \ll \|\boldsymbol{w}_{+,r}^{(T_0)}\|_2$ for any $(+, r), (+, r') \in S_+^{*(0)}(\boldsymbol{v}_+)$.

Let $(+, r) \in S_+^{*(0)}(\boldsymbol{v}_+)$. Similar to phase I, consider the update equation

$$
\boldsymbol{w}_{+,r}^{(t+1)} = \boldsymbol{w}_{+,r}^{(t)} + \eta \frac{1}{NP} \times
\tag{76}
$$

$$
\sum_{n=1}^{N}\left(\mathbb{1}\{y_n = +\}[1 - \mathrm{logit}_+^{(t)}(\boldsymbol{X}_n^{(t)})]\sum_{p\in[P]}\sigma'(\langle \boldsymbol{w}_{+,r}^{(t)}, \boldsymbol{x}_{n,p}^{(t)}\rangle + b_{+,r}^{(t)})\boldsymbol{x}_{n,p}^{(t)}\right.
\tag{77}
$$

$$
\left. + \mathbb{1}\{y_n = -\}[-\mathrm{logit}_+^{(t)}(\boldsymbol{X}_n^{(t)})]\sum_{p\in[P]}\sigma'(\langle \boldsymbol{w}_{+,r}^{(t)}, \boldsymbol{x}_{n,p}^{(t)}\rangle + b_{+,r}^{(t)})\boldsymbol{x}_{n,p}^{(t)}\right)
\tag{78}
$$

For the on-diagonal update expression, we have

$$
\sum_{n=1}^{N}\mathbb{1}\{y_n = +\}[1 - \mathrm{logit}_+^{(t)}(\boldsymbol{X}_n^{(t)})]\sum_{p\in[P]}\sigma'(\langle \boldsymbol{w}_{+,r}^{(t)}, \boldsymbol{x}_{n,p}^{(t)}\rangle + b_{+,r}^{(t)})\boldsymbol{x}_{n,p}^{(t)}
$$

$$
= \sum_{n=1}^{N}\mathbb{1}\{y_n = +\}[1 - \mathrm{logit}_+^{(t)}(\boldsymbol{X}_n^{(t)})]
$$

$$
\left\{\mathbb{1}\{|\mathcal{P}(\boldsymbol{X}_n^{(t)}; \boldsymbol{v}_+)| > 0\}\left[\sum_{p\in\mathcal{P}(\boldsymbol{X}_n^{(t)};\boldsymbol{v}_+)}\mathbb{1}\left\{\langle \boldsymbol{w}_{+,r}^{(t)}, \alpha_{n,p}^{(t)}\boldsymbol{v}_+ + \boldsymbol{\zeta}_{n,p}^{(t)}\rangle \ge b_{+,r}^{(t)}\right\}\left(\alpha_{n,p}^{(t)}\boldsymbol{v}_+ + \boldsymbol{\zeta}_{n,p}^{(t)}\right)\right.\right.
\tag{79}
$$

$$
\left. + \sum_{p\notin\mathcal{P}(\boldsymbol{X}_n^{(t)};\boldsymbol{v}_+)}\mathbb{1}\left\{\langle \boldsymbol{w}_{+,r}^{(t)}, \boldsymbol{x}_{n,p}^{(t)}\rangle \ge b_{+,r}^{(t)}\right\}\boldsymbol{x}_{n,p}^{(t)}\right]
$$

$$
\left. + \mathbb{1}\{|\mathcal{P}(\boldsymbol{X}_n^{(t)}; \boldsymbol{v}_+)| = 0\}\sum_{p\in[P]}\mathbb{1}\left\{\langle \boldsymbol{w}_{+,r}^{(t)}, \boldsymbol{x}_{n,p}^{(t)}\rangle \ge b_{+,r}^{(t)}\right\}\boldsymbol{x}_{n,p}^{(t)}\right\}
$$

Following from Theorem D.1 and F.1, the neurons' non-activation on the patches that do not contain $\boldsymbol{v}_+$, and activation on the $\boldsymbol{v}_+$-dominated patches hold with probability at least $1 - O\left(\frac{mNPk_+}{\text{poly}(d)}\right)$ at time $T_0$. Therefore, the above update expression reduces to

$$\sum_{n=1}^{N} \mathbb{1}\{y_n = +, |\mathcal{P}(\boldsymbol{X}_n^{(t)}; \boldsymbol{v}_+)| > 0\}[1 - \text{logit}_+^{(t)}(\boldsymbol{X}_n^{(t)})] \sum_{p \in \mathcal{P}(\boldsymbol{X}_n^{(t)}; \boldsymbol{v}_+)} \left(\alpha_{n,p}^{(t)} \boldsymbol{v}_+ + \boldsymbol{\zeta}_{n,p}^{(t)}\right) \tag{80}$$

Note that for samples $\boldsymbol{X}_n^{(t)}$ with $y_n = +$,

$$[1 - \text{logit}_+^{(t)}(\boldsymbol{X}_n^{(t)})] = \frac{\exp(F_-^{(t)}(\boldsymbol{X}_n^{(t)}))}{\exp(F_-^{(t)}(\boldsymbol{X}_n^{(t)})) + \exp(F_+^{(t)}(\boldsymbol{X}_n^{(t)}))} \tag{81}$$

Now we need to estimate the network response $F_+^{(t)}(\boldsymbol{X}_n^{(t)})$. With probability at least $1 - \exp(-\Omega(s^{*1/3}))$, we have the upper bound (let $(+, c_n^t)$ denote the subclass which sample $\boldsymbol{X}_n^{(t)}$ belongs to):

$$\begin{aligned}
&F_+^{(t)}(\boldsymbol{X}_n^{(t)}) \\
&\leq \sum_{p \in \mathcal{P}(\boldsymbol{X}^{(t)}; \boldsymbol{v}_+)} \sum_{(+,r) \in S_+^{(0)}(\boldsymbol{v}_+)} \langle \boldsymbol{w}_{+,r}^{(t)}, \boldsymbol{v}_+ + \boldsymbol{\zeta}_{n,p}^{(t)} \rangle + b_{+,r}^{(t)} \\
&\quad + \sum_{p \in \mathcal{P}(\boldsymbol{X}^{(t)}; \boldsymbol{v}_{+,c_n^t})} \sum_{(+,r) \in S_+^{(0)}(\boldsymbol{v}_{+,c_n^t})} \langle \boldsymbol{w}_{+,r}^{(t)}, \boldsymbol{v}_{+,c_n^t} + \boldsymbol{\zeta}_{n,p}^{(t)} \rangle + b_{+,r}^{(t)} \\
&\leq (1 + s^{*-1/3})\sqrt{1 + \iota}s^* \left(1 + O\left(\frac{1}{\log^9(d)}\right)\right) \left(A_{+,r^*}^{*(t)} \left|S_+^{(0)}(\boldsymbol{v}_+)\right| + A_{+,c_n^t,r^*}^{*(t)} \left|S_+^{(0)}(\boldsymbol{v}_{+,c_n^t})\right|\right)
\end{aligned} \tag{82}$$

The second inequality is true since $\max_r \langle \boldsymbol{w}_{+,r}^{(t)}, \boldsymbol{v}_+ \rangle \leq A_{+,r^*}^{*(t)} + O(\sigma_0 \log(d))$, and for any $(+, r) \in S_+^{(0)}(\boldsymbol{v}_+)$, $|\langle \boldsymbol{w}_{+,r}^{(t)}, \boldsymbol{\zeta}_{n,p}^{(t)} \rangle| \leq O(1/\log^9(d))A_{+,r^*}^{*(t)}$. The bias value is negative (and so less than 0).

To further refine the bound, we recall $\left|S_+^{*(0)}(\boldsymbol{v})\right| / \left|S_+^{*(0)}(\boldsymbol{v}')\right|, \left|S_+^{*(0)}(\boldsymbol{v})\right| / \left|S_+^{(0)}(\boldsymbol{v}')\right| = 1 \pm O(1/\log^5(d))$. Therefore, we obtain the bound

$$\begin{aligned}
F_+^{(t)}(\boldsymbol{X}_n^{(t)}) \leq &(1 + s^{*-1/3})\sqrt{1 + \iota}s^* \left(1 + O\left(\frac{1}{\log^5(d)}\right)\right) \left(1 + O\left(\frac{1}{\log^5(d)}\right)\right) \\
&\times \left(A_{+,r^*}^{*(t)} \left|S_+^{*(0)}(\boldsymbol{v}_+)\right| + A_{+,c_n^t,r^*}^{*(t)} \left|S_+^{*(0)}(\boldsymbol{v}_{+,c_n^t})\right|\right)
\end{aligned} \tag{83}$$

Following a similar argument, we also have the lower bound

$$\begin{aligned}
&F_+^{(t)}(\boldsymbol{X}_n^{(t)}) \\
&\geq \sum_{p \in \mathcal{P}(\boldsymbol{X}^{(t)}; \boldsymbol{v}_+)} \sum_{(+,r) \in S_+^{*(0)}(\boldsymbol{v}_+)} \sigma\left(\langle \boldsymbol{w}_{+,r}^{(t)}, \boldsymbol{v}_+ + \boldsymbol{\zeta}_{n,p}^{(t)} \rangle + b_{+,r}^{(t)}\right) \\
&\quad + \sum_{p \in \mathcal{P}(\boldsymbol{X}^{(t)}; \boldsymbol{v}_{+,c_n^t})} \sum_{(+,r) \in S_+^{*(0)}(\boldsymbol{v}_{+,c_n^t})} \sigma\left(\langle \boldsymbol{w}_{+,r}^{(t)}, \boldsymbol{v}_{+,c_n^t} + \boldsymbol{\zeta}_{n,p}^{(t)} \rangle + b_{+,r}^{(t)}\right) \\
&\geq (1 - s^{*-1/3})\sqrt{1 - \iota}s^* \left(1 - O\left(\frac{1}{\log^5(d)}\right)\right) \left(1 - O\left(\frac{1}{\log^5(d)}\right)\right) \\
&\quad \times \left(A_{+,r^*}^{*(t)} \left|S_+^{*(0)}(\boldsymbol{v}_+)\right| + A_{+,c_n^t,r^*}^{*(t)} \left|S_+^{*(0)}(\boldsymbol{v}_{+,c_n^t})\right|\right)
\end{aligned} \tag{84}$$

The neurons in $S_+^{*(0)}(\boldsymbol{v}_+)$ have to activate, therefore they serve a key role in the lower bound, the bias bound for them is simply $-A_{+,r^*}^{*(t)}\Theta(1/\log^5(d))$; the neurons in $S_+^{(0)}(\boldsymbol{v}_{+,c})$ contribute at least 0 due to the ReLU activation; the rest of the neurons do not activate. The same reasoning holds for the $S_+^{*(0)}(\boldsymbol{v}_{+,c})$.

Knowing that neurons in $S_+^{*(0)}(\boldsymbol{v}_+)$ cannot activate on the patches in samples belonging to the "$-$" class, now we may write the update expression for every $(+,r) \in S_+^{*(t)}(\boldsymbol{v}_+)$ as (their updates are identical, same as in phase I):

$$
\begin{aligned}
&\Delta\boldsymbol{w}_{+,r}^{(t)}\\
&=\frac{\eta}{NP}\sum_{n=1}^{N}\mathbb{1}\{y_n=+\}[1-\mathrm{logit}_+^{(t)}(\boldsymbol{X}_n^{(t)})]\sum_{p\in[P]}\sigma'(\langle\boldsymbol{w}_{+,r}^{(t)},\boldsymbol{x}_{n,p}^{(t)}\rangle+b_{+,r}^{(t)})\boldsymbol{x}_{n,p}^{(t)}\\
&=\frac{\eta}{NP}\sum_{n=1}^{N}\mathbb{1}\{y_n=+,|\mathcal{P}(\boldsymbol{X}_n^{(t)};\boldsymbol{v}_+)|>0\}\exp(-F_+^{(t)}(\boldsymbol{X}_n^{(t)}))\\
&\quad\times\frac{\exp(F_-^{(t)}(\boldsymbol{X}_n^{(t)}))}{\exp\left(F_-^{(t)}(\boldsymbol{X}_n^{(t)})-\exp(F_+^{(t)}(\boldsymbol{X}_n^{(t)}))\right)+1}\sum_{p\in\mathcal{P}(\boldsymbol{X}_n^{(t)};\boldsymbol{v}_+)}\left(\alpha_{n,p}^{(t)}\boldsymbol{v}_++\boldsymbol{\zeta}_{n,p}^{(t)}\right)\\
&=\eta\sum_{n=1}^{N}\mathbb{1}\{y_n=+\}\exp\left\{-(1+s^{*-1/3})\sqrt{1+\iota}s^*\left(1+O\left(\frac{1}{\log^5(d)}\right)\right)\right.\\
&\quad\times\left.\left(A_{+,r^*}^{*(t)}\left|S_+^{*(0)}(\boldsymbol{v}_+)\right|+A_{+,c_n^t,r^*}^{*(t)}\left|S_+^{*(0)}(\boldsymbol{v}_{+,c_n^t})\right|\right)\right\}\\
&\quad\times\frac{\exp(F_-^{(t)}(\boldsymbol{X}_n^{(t)}))}{\exp\left(F_-^{(t)}(\boldsymbol{X}_n^{(t)})-F_+^{(t)}(\boldsymbol{X}_n^{(t)})\right)+1}(1\pm s^{*-1/3})\frac{s^*}{NP}\left(\sqrt{1\pm\iota}\boldsymbol{v}_++\boldsymbol{\zeta}_{n,p}^{(t)}\right)
\end{aligned}
\tag{85}
$$

This concludes the proof of the base case.

**Induction step**. Assume the statements hold for time period $[T_0,t]$, prove for time $t+1$.

At step $t+1$, based on the induction hypothesis, we know that with probability at least $1-O\left(\frac{mNPk_+t}{\mathrm{poly}(d)}\right)$, during time $\tau\in[T_0,t]$, for any $(+,r)\in S_+^{*(0)}(\boldsymbol{v}_+)$,

$$
\begin{aligned}
&\Delta\boldsymbol{w}_{+,r}^{(\tau)}\\
&=\eta\sum_{n=1}^{N}\mathbb{1}\{y_n=+\}\exp\left\{-(1+s^{*-1/3})\sqrt{1+\iota}s^*\left(1+O\left(\frac{1}{\log^5(d)}\right)\right)\right.\\
&\quad\times\left.\left(A_{+,r^*}^{*(\tau)}\left|S_+^{*(0)}(\boldsymbol{v}_+)\right|+A_{+,c_n^t,r^*}^{*(\tau)}\left|S_+^{*(0)}(\boldsymbol{v}_{+,c_n^t})\right|\right)\right\}\\
&\quad\times\frac{\exp(F_-^{(\tau)}(\boldsymbol{X}_n^{(\tau)}))}{\exp\left(F_-^{(\tau)}(\boldsymbol{X}_n^{(\tau)})-\exp(F_+^{(\tau)}(\boldsymbol{X}_n^{(\tau)}))\right)+1}(1\pm s^{*-1/3})\frac{s^*}{NP}\left(\sqrt{1\pm\iota}\boldsymbol{v}_++\boldsymbol{\zeta}_{n,p}^{(\tau)}\right)
\end{aligned}
\tag{86}
$$

and for the bias,

$$
\begin{aligned}
&\Delta b_{+,r}^{(\tau)} \\
&\leq -\eta \frac{1}{\log^5(d)} \sum_{n=1}^{N} \mathbb{1}\{y_n = +\} \exp\left\{ -(1+s^{*-1/3})\sqrt{1+\iota}s^* \left(1+O\left(\frac{1}{\log^5(d)}\right)\right) \right. \\
&\qquad \left. \times \left( A_{+,r^*}^{*(\tau)} \left| S_+^{*(0)}(\boldsymbol{v}_+) \right| + A_{+,c_n^t,r^*}^{*(\tau)} \left| S_+^{*(0)}(\boldsymbol{v}_{+,c_n^t}) \right| \right) \right\} \\
&\qquad \times (1-s^{*-1/3}) \frac{s^*}{NP}\left( \sqrt{1-\iota} - \frac{1}{\log^{10}(d)}\right) \frac{\exp(F_-^{(\tau)}(\boldsymbol{X}_n^{(\tau)}))}{\exp\left(F_-^{(\tau)}(\boldsymbol{X}_n^{(\tau)}) - \exp(F_+^{(\tau)}(\boldsymbol{X}_n^{(\tau)}))\right) + 1}
\end{aligned}
\tag{87}
$$

Conditioning on the high-probability events of the induction hypothesis,

$$
\begin{aligned}
&\boldsymbol{w}_{+,r}^{(t+1)} \\
&= \boldsymbol{w}_{+,r}^{(T_0)} \\
&\quad + \eta \sum_{\tau=T_0}^{t} \sum_{n=1}^{N} \mathbb{1}\{y_n = +\} \exp\left\{ -(1+s^{*-1/3})\sqrt{1+\iota}s^* \left(1+O\left(\frac{1}{\log^5(d)}\right)\right) \right. \\
&\qquad \left. \times \left( A_{+,r^*}^{*(\tau)} \left| S_+^{*(0)}(\boldsymbol{v}_+) \right| + A_{+,c_n^t,r^*}^{*(\tau)} \left| S_+^{*(0)}(\boldsymbol{v}_{+,c_n^t}) \right| \right) \right\} \\
&\qquad \times \frac{\exp(F_-^{(\tau)}(\boldsymbol{X}_n^{(\tau)}))}{\exp\left(F_-^{(\tau)}(\boldsymbol{X}_n^{(\tau)}) - \exp(F_+^{(\tau)}(\boldsymbol{X}_n^{(\tau)}))\right) + 1}(1 \pm s^{*-1/3}) \frac{s^*}{NP}\left( \sqrt{1 \pm \iota}\,\boldsymbol{v}_+ + \boldsymbol{\zeta}_{n,p}^{(\tau)}\right)
\end{aligned}
\tag{88}
$$

It follows that, with probability at least $1 - O\left(\frac{mNP}{\text{poly}(d)}\right)$, for all $\boldsymbol{v}_+$-dominated patch $\boldsymbol{x}_{n,p}^{(t+1)}$,

$$
\langle \boldsymbol{w}_{+,r}^{(t+1)}, \boldsymbol{x}_{n,p}^{(t+1)} \rangle + b_{+,r}^{(t+1)}
$$

$$
= \langle \boldsymbol{w}_{+,r}^{(T_0)}, \sqrt{1 \pm \iota} \boldsymbol{v}_+ + \boldsymbol{\zeta}_{n,p}^{(t+1)} \rangle + b_{+,r}^{(T_0)}
$$

$$
+ \eta \sum_{\tau=T_0}^{t} \sum_{n=1}^{N} \mathbb{1}\{y_n = +\} \exp\left\{ -(1+s^{*-1/3})\sqrt{1+\iota}s^* \left(1 + O\left(\frac{1}{\log^5(d)}\right)\right) \right.
$$

$$
\times \left( A_{+,r^*}^{*(\tau)} \left| S_+^{*(0)}(\boldsymbol{v}_+) \right| + A_{+,c_n^t,r^*}^{*(\tau)} \left| S_+^{*(0)}(\boldsymbol{v}_{+,c_n^t}) \right| \right) \right\}
$$

$$
\times \frac{\exp(F_-^{(\tau)}(\boldsymbol{X}_n^{(\tau)}))}{\exp\left(F_-^{(\tau)}(\boldsymbol{X}_n^{(\tau)}) - F_+^{(\tau)}(\boldsymbol{X}_n^{(\tau)})\right) + 1} (1 \pm s^{*-1/3}) \frac{s^*}{NP}
$$

$$
\times \langle \sqrt{1 \pm \iota} \boldsymbol{v}_+ + \boldsymbol{\zeta}_{n,p}^{(\tau)}, \sqrt{1 \pm \iota} \boldsymbol{v}_+ + \boldsymbol{\zeta}_{n,p}^{(t+1)} \rangle + \Delta b_{+,r}^{(\tau)}
$$

$$
\geq 0 \tag{89}
$$

$$
+ \eta \sum_{\tau=T_0}^{t} \sum_{n=1}^{N} \mathbb{1}\{y_n = +\} \exp\left\{ -(1+s^{*-1/3})\sqrt{1+\iota}s^* \left(1 + O\left(\frac{1}{\log^5(d)}\right)\right) \right.
$$

$$
\times \left( A_{+,r^*}^{*(\tau)} \left| S_+^{*(0)}(\boldsymbol{v}_+) \right| + A_{+,c_n^t,r^*}^{*(\tau)} \left| S_+^{*(0)}(\boldsymbol{v}_{+,c_n^t}) \right| \right) \right\}
$$

$$
\times \frac{\exp(F_-^{(\tau)}(\boldsymbol{X}_n^{(\tau)}))}{\exp\left(F_-^{(\tau)}(\boldsymbol{X}_n^{(\tau)}) - F_+^{(\tau)}(\boldsymbol{X}_n^{(\tau)})\right) + 1} (1 \pm s^{*-1/3}) \frac{s^*}{NP}
$$

$$
\times \left(1 - \iota - O\left(\frac{1}{\log^5(d)}\right)\right)
$$

$$
> 0
$$

Therefore the neurons $(+, r) \in S_+^{*(0)}(\boldsymbol{v}_+)$ activate on the $\boldsymbol{v}_+$-dominated patches $\boldsymbol{x}_{n,p}^{(t+1)}$. We also know that they cannot activate on patches that are not dominated by $\boldsymbol{v}_+$ by Theorem F.1. Following a similar derivation to the base case, we arrive at the result that, conditioning on the events of the induction hypothesis, with probability at least $1 - O\left(\frac{mNPk_+}{\text{poly}(d)}\right)$, for all $(+, r) \in S_+^{*(0)}(\boldsymbol{v}_+)$,

$$
\Delta \boldsymbol{w}_{+,r}^{(t+1)}
$$

$$
= \eta \sum_{n=1}^{N} \mathbb{1}\{y_n = +\} \exp\left\{ -(1+s^{*-1/3})\sqrt{1+\iota}s^* \left(1 + O\left(\frac{1}{\log^5(d)}\right)\right) \right.
$$

$$
\times \left( A_{+,r^*}^{*(t+1)} \left| S_+^{*(0)}(\boldsymbol{v}_+) \right| + A_{+,c_n^t,r^*}^{*(t+1)} \left| S_+^{*(0)}(\boldsymbol{v}_{+,c_n^t}) \right| \right) \right\} \tag{90}
$$

$$
\times (1 \pm s^{*-1/3}) \frac{s^*}{NP} \frac{\exp(F_-^{(t+1)}(\boldsymbol{X}_n^{(t+1)}))}{\exp\left(F_-^{(t+1)}(\boldsymbol{X}_n^{(t+1)}) - F_+^{(t+1)}(\boldsymbol{X}_n^{(t+1)})\right) + 1} \left(\sqrt{1 \pm \iota} \boldsymbol{v}_+ + \boldsymbol{\zeta}_{n,p}^{(t+1)}\right)
$$

Consequently, with probability at least $1 - O\left(\frac{mNP}{\text{poly}(d)}\right)$,

$$
\begin{aligned}
\Delta b_{+,r}^{(t+1)} \\
\leq & -\frac{1}{\log^5(d)} \sum_{n=1}^{N} \mathbb{1}\{y_n = +\}\eta \exp\Bigg\{ -(1 + s^{*-1/3})\sqrt{1 + \iota} s^* \left(1 + O\left(\frac{1}{\log^5(d)}\right)\right) \\
& \times \left(A_{+,r^*}^{*(t+1)} \left|S_+^{*(0)}(\boldsymbol{v}_+)\right| + A_{+,c_n^t,r^*}^{*(t+1)} \left|S_+^{*(0)}(\boldsymbol{v}_{+,c_n^t})\right|\right) \Bigg\} \\
& \times \frac{\exp(F_-^{(t+1)}(\boldsymbol{X}_n^{(t+1)}))}{\exp\left(F_-^{(t+1)}(\boldsymbol{X}_n^{(t+1)}) - F_+^{(t+1)}(\boldsymbol{X}_n^{(t+1)})\right) + 1}(1 - s^{*-1/3})\frac{s^*}{NP} \\
& \times \left(1 - \iota - O\left(\frac{1}{\log^9(d)}\right)\right)
\end{aligned}
\tag{91}
$$

Utilizing the definition of conditional probability, we conclude that the expressions for $\Delta w_{+,r}^{(\tau)}$ and $\Delta b_{+,r}^{(t+1)}$ are indeed as described in the theorem during time $\tau \in [T_0, t+1]$ with probability at least $\left(1 - O\left(\frac{mNPk_+t}{\text{poly}(d)}\right)\right) \times \left(1 - O\left(\frac{mNPk_+}{\text{poly}(d)}\right)\right) \geq 1 - O\left(\frac{mNPk_+(t+1)}{\text{poly}(d)}\right)$.

Moreover, based on the expression of $\Delta w_{+,r}^{(\tau)}$ and $\Delta b_{+,r}^{(t+1)}$, following virtually the same argument as in the base case, we can estimate the network output for any $(\boldsymbol{X}_n^{(t+1)}, y_n = +)$:

$$
\begin{aligned}
F_+^{(t+1)}(\boldsymbol{X}_n^{(t+1)}) = & (1 \pm s^{*-1/3})\sqrt{1 \pm \iota}\left(1 \pm O\left(\frac{1}{\log^5(d)}\right)\right) s^* \\
& \times \left(A_{+,r^*}^{*(t)} \left|S_+^{*(0)}(\boldsymbol{v}_+)\right| + A_{+,c_n^t,r^*}^{*(t)} \left|S_+^{*(0)}(\boldsymbol{v}_{+,c_n^t})\right|\right)
\end{aligned}
\tag{92}
$$

$\square$

**Lemma E.3.** *Define time $T_{1,1}$ to be the first point in time which the following identity holds on all $\boldsymbol{X}_n^{(t)}$ belonging to the "+" class:*

$$
\frac{\exp(F_-^{(t)}(\boldsymbol{X}_n^{(t)}))}{\exp(F_-^{(t)}(\boldsymbol{X}_n^{(t)}) - F_+^{(t)}(\boldsymbol{X}_n^{(t)})) + 1} \geq 1 - O\left(\frac{1}{\log^5(d)}\right)
\tag{93}
$$

*Then $T_{1,1} \leq poly(d)$, and for all $t \in [T_{1,1}, T_1]$, the above holds. The following also holds for this time period:*

$$
[1 - logit_+^{(t)}(\boldsymbol{X}_n^{(t)})] \leq O\left(\frac{1}{\log^5(d)}\right)
\tag{94}
$$

*The same results also hold with the class signs flipped.*

*Proof.* We first note that, the training loss $[1 - logit_+^{(t)}(\boldsymbol{X}_n^{(t)})]$ on samples belonging to the "+" class at any time during $t \in [T_0, T_1]$ is, asymptotically speaking, monotonically decreasing from $\frac{1}{2} - O(d^{-1})$. This can be easily proven by observing the way $s^* \left(A_{+,r^*}^{*(t)} \left|S_+^{*(0)}(\boldsymbol{v}_+)\right| + A_{+,c,r^*}^{*(t)} \left|S_+^{*(0)}(\boldsymbol{v}_{+,c})\right|\right)$ monotonically increases from the proof of Lemma E.2: before $F_+^{(t)}(\boldsymbol{X}_n^{(t)}) \geq \log\log^5(d)$ on all $\boldsymbol{X}_n^{(t)}$ belonging to the "+" class, there

must be some samples $\boldsymbol{X}_n^{(t)}$ on which

$$
\begin{aligned}
[1 - \text{logit}_+^{(t)}(\boldsymbol{X}_n^{(t)})] = & \frac{\exp(F_-^{(t)}(\boldsymbol{X}_n^{(t)}))}{\exp(F_-^{(t)}(\boldsymbol{X}_n^{(t)})) + \exp(F_+^{(t)}(\boldsymbol{X}_n^{(t)}))} \\
\geq & \frac{1 - O(\sigma_0 \log(d) s^* d^{c_0})}{1 + O(\sigma_0 \log(d) s^* d^{c_0}) + \log^5(d)} \\
\geq & \Omega\left(\frac{1}{\log^5(d)}\right).
\end{aligned}
\tag{95}
$$

Therefore, by the update expressions in the proof of Lemma E.2, $F_+^{(t)}(\boldsymbol{X}_n^{(t)})$ can reach $\log \log^5(d)$ in time at most $O\left(\frac{NP \log^5(d)}{\eta s^*}\right) \in \text{poly}(d)$ (in the worst case scenario). At time $T_{1,1}$ and beyond,

$$
\begin{aligned}
1 - \frac{\exp(F_-^{(t)}(\boldsymbol{X}_n^{(t)}))}{\exp(F_-^{(t)}(\boldsymbol{X}_n^{(t)}) - F_+^{(t)}(\boldsymbol{X}_n^{(t)})) + 1} \leq & 1 - \frac{\exp(1 - O(\sigma_0 d^{c_0} s^*))}{\exp(1 + O(\sigma_0 d^{c_0} s^*))\frac{1}{\log^5(d)} + 1} \\
\leq & O\left(\frac{1}{\log^5(d)}\right).
\end{aligned}
\tag{96}
$$

$\square$

**Lemma E.4.** *Denote $C = \eta \frac{s^*}{2k_+ P}$, and write (for any $c \in [k_+]$)*

$$
A_c(t) = s^* \left( A_{+,r^*}^{*(t)} \left| S_+^{*(0)}(\boldsymbol{v}_+) \right| + A_{+,c,r^*}^{*(t)} \left| S_+^{*(0)}(\boldsymbol{v}_{+,c}) \right| \right)
\tag{97}
$$

*(see Lemma E.2 for definition of $A_\cdot^{*(t)}$). Define $t_{c,0} = \exp(A_c(T_{1,1}))$. We write $A(t)$ and $t_0$ below for cleaner notations.*

*Then with probability at least $1 - o(1)$, during $t \in [T_{1,1}, T_1]$,*

$$
A(t) = \log(C(t - T_{1,1}) + t_0) + E(t)
\tag{98}
$$

*where $|E(t)| \leq O\left(\frac{1}{\log^4(d)}\right) \sum_{\tau = C^{-1} t_0}^{t - T_{1,1} + C^{-1} t_0} \frac{1}{\tau} \leq O\left(\frac{\log(t) - \log(C^{-1} t_0)}{\log^4(d)}\right).$*

*The same results also hold with the class signs flipped.*

*Proof.* **Sidenote**: To make the writing a bit cleaner, we assume in the proof below that $C^{-1} t_0$ is an integer. The general case is easy to extend to by observing that $\left| \frac{1}{t - T_{1,1} + \lceil C^{-1} t_0 \rceil} - \frac{1}{t - T_{1,1} + C^{-1} t_0} \right| \leq \frac{1}{(t - T_{1,1} + \lceil C^{-1} t_0 \rceil)(t - T_{1,1} + C^{-1} t_0)}$, which can be absorbed into the error term at every iteration since $\frac{1}{t - T_{1,1} + \lceil C^{-1} t_0 \rceil} \ll \frac{1}{\log^4(d)}$ due to $C^{-1} t_0 \geq \Omega(\sigma_0^{-1}/(\text{polylog}(d) d^{c_0})) \gg d \gg \log^4(d)$.

Based on result from Lemmas E.2 and E.3, as long as $A(t) \leq O(\log(d))$, we know during time $t \in [T_{1,1}, T_1]$ the update rule for $A(t)$ is as follows:

$$
\begin{aligned}
A(t+1) - A(t) = & C \exp\left\{ -(1 \pm s^{*-1/3})\sqrt{1 \pm \iota}\left(1 \pm O\left(\frac{1}{\log^5(d)}\right)\right) A(t) \right\} \\
& \times \left(1 \pm O\left(\frac{1}{\log^5(d)}\right)\right)(1 \pm s^{*-1/3})\left(\sqrt{1 \pm \iota} \pm \frac{1}{\log^{10}(d)}\right) \\
= & C \exp\{-A(t)\} \exp\left\{ \pm O\left(\frac{1}{\log^4(d)}\right) \right\}\left(1 \pm O\left(\frac{1}{\log^5(d)}\right)\right) \\
= & C \exp\{-A(t)\}\left(1 \pm \frac{C_1}{\log^4(d)}\right)
\end{aligned}
\tag{99}
$$

where we write $C_1$ in place of $O(\cdot)$ for a more concrete update expression.

The base case $t = T_{1,1}$ is trivially true.

We proceed with the induction step. Assume the hypothesis true for $t \in [T_{1,1}, T]$, prove for $t + 1 = T + 1$.

Note that by Lemma E.10,

$$
\begin{aligned}
A(t+1) =& \log(C(t - T_{1,1}) + t_0) + E(t) \\
&+ C \exp\left\{-\log(C(t - T_{1,1}) + t_0) - E(t))\right\} \left(1 \pm \frac{C_1}{\log^4(d)}\right) \\
=& \log(C) + \log(t - T_{1,1} + C^{-1}t_0) + E(t) \\
&+ C \frac{1}{C(t - T_{1,1}) + t_0} \left(1 - E(t) \pm O(E(t)^2)\right) \left(1 \pm \frac{C_1}{\log^4(d)}\right) \\
=& \log(C) + \sum_{\tau=1}^{t - T_{1,1} + C^{-1}t_0 - 1} \frac{1}{\tau} + \frac{1}{2}\frac{1}{t - T_{1,1} + C^{-1}t_0} + \left[0, \frac{1}{8}\frac{1}{(t - T_{1,1} + C^{-1}t_0)^2}\right] \\
&+ \frac{1}{t - T_{1,1} + C^{-1}t_0} \pm \frac{C_1}{\log^4(d)}\frac{1}{t - T_{1,1} + C^{-1}t_0} \\
&+ E(t) + \frac{1}{t - T_{1,1} + C^{-1}t_0} \left(-E(t) \pm O(E(t)^2)\right) \left(1 \pm \frac{C_1}{\log^4(d)}\right) \\
=& \log(C) + \sum_{\tau=1}^{t - T_{1,1} + C^{-1}t_0} \frac{1}{\tau} + \frac{1}{2}\frac{1}{t - T_{1,1} + C^{-1}t_0} + \left[0, \frac{1}{8}\frac{1}{(t - T_{1,1} + C^{-1}t_0)^2}\right] \\
&\pm \frac{C_1}{\log^4(d)}\frac{1}{t - T_{1,1} + C^{-1}t_0} \\
&+ E(t) + \frac{1}{t - T_{1,1} + C^{-1}t_0} \left(-E(t) \pm O(E(t)^2)\right) \left(1 \pm \frac{C_1}{\log^4(d)}\right)
\end{aligned}
\tag{100}
$$

Invoking Lemma E.10 again,

$$
\begin{aligned}
A(t+1) =& \log(C) + \log(t + 1 - T_{1,1} + C^{-1}t_0) \\
&- \frac{1}{2}\frac{1}{t + 1 - T_{1,1} + C^{-1}t_0} + \frac{1}{2}\frac{1}{t - T_{1,1} + C^{-1}t_0} \\
&+ \left[-\frac{1}{8}\frac{1}{(t + 1 - T_{1,1} + C^{-1}t_0)^2}, 0\right] + \left[0, \frac{1}{8}\frac{1}{(t - T_{1,1} + C^{-1}t_0)^2}\right] \\
&\pm \frac{C_1}{\log^4(d)}\frac{1}{t - T_{1,1} + C^{-1}t_0} \\
&+ E(t) + \frac{1}{t - T_{1,1} + C^{-1}t_0} \left(-E(t) \pm O(E(t)^2)\right) \left(1 \pm \frac{C_1}{\log^4(d)}\right) \\
=& \log(C(t + 1 - T_{1,1}) + t_0) \\
&+ \frac{1}{2}\frac{1}{(t + 1 - T_{1,1} + C^{-1}t_0)(t - T_{1,1} + C^{-1}t_0)} \pm O\left(\frac{1}{(t + 1 - T_{1,1} + C^{-1}t_0)^2}\right) \\
&\pm \frac{C_1}{\log^4(d)}\frac{1}{t - T_{1,1} + C^{-1}t_0} \\
&+ E(t) + \frac{1}{t - T_{1,1} + C^{-1}t_0} \left(-E(t) \pm O(E(t)^2)\right) \left(1 \pm \frac{C_1}{\log^4(d)}\right)
\end{aligned}
\tag{101}
$$

To further refine the expression, first note that the error passed down from the previous step $t$ does not grow in this step (in fact it slightly decreases):

$$\left| E(t) + \frac{1}{t - T_{1,1} + C^{-1}t_0} \left( -E(t) \pm O(E(t)^2) \right) \left( 1 \pm \frac{C_1}{\log^4(d)} \right) \right|$$

$$< |E(t)| \tag{102}$$

$$\leq O\left( \frac{1}{\log^4(d)} \right)^{t - T_{1,1} + C^{-1}t_0} \sum_{\tau = C^{-1}t_0}^{} \frac{1}{\tau}.$$

Moreover, notice that at step $t+1$, since $\frac{1}{t+1-T_{1,1}+C^{-1}t_0} \ll \frac{1}{\log^4(d)}$, the error term $|E(t+1)| = |A(t+1) - \log(C(t+1-T_{1,1}) + t_0)| \leq O\left( \frac{1}{\log^4(d)} \right) \sum_{\tau=C^{-1}t_0}^{t+1-T_{1,1}+C^{-1}t_0} \frac{1}{\tau}$, which finishes the inductive step.

$\square$

**Lemma E.5.** *With probability at least* $1 - O\left( \frac{mNPk_+T_1}{poly(d)} \right)$, *for all* $t \in [0, T_1]$, *all* $c \in [k_+]$,

$$\frac{\Delta A_{+,c,r^*}^{*(t)}}{\Delta A_{+,r^*}^{*(t)}} = \Theta\left( \frac{1}{k_+} \right),$$

$$\frac{A_{+,c,r^*}^{*(t)}}{A_{+,r^*}^{*(t)}} = \Theta\left( \frac{1}{k_+} \right). \tag{103}$$

*The same identity holds for the "$-$"-classes.*

*Proof.* The statements in the lemma follow trivially from Theorem D.1 for time period $[0, T_0]$. Let us focus on the phase $[T_0, T_1]$.

In this proof, we condition on the high-probability events of Lemma E.4 and Lemma E.2.

First of all, based on Lemma E.4, we know that $s^* A_{+,r^*}^{*(t)} \left| S_+^{*(0)}(\boldsymbol{v}_+) \right| \leq O(\log(d))$. We will make use of this fact later.

**Base case**, $t = T_0$.

The base case directly follows from our Theorem D.1.

**Induction step**, assume statement holds for $\tau \in [T_0, t]$, prove statement for $t+1$.

By Lemma E.2, we know that

$$\Delta A_{+,r^*}^{*(t)}$$

$$= \eta \sum_{c=1}^{k_+} \exp\left\{ -(1 \pm s^{*-1/3})\sqrt{1 \pm \iota}s^* \left( 1 \pm O\left( \frac{1}{\log^5(d)} \right) \right) \times \left( A_{+,r^*}^{*(t)} \left| S_+^{*(0)}(\boldsymbol{v}_+) \right| + A_{+,c,r^*}^{*(t)} \left| S_+^{*(0)}(\boldsymbol{v}_{+,c}) \right| \right) \right\}$$

$$\times [1/3, 1](1 \pm s^{*-1/3}) \frac{s^*}{2k_+P} \left( \sqrt{1 \pm \iota} \pm O\left( \frac{1}{\log^9(d)} \right) \right), \tag{104}$$

and for any $c \in [k_+]$,

$$\Delta A_{+,c,r^*}^{*(t)}$$

$$= \eta \exp\left\{ -(1 \pm s^{*-1/3})\sqrt{1 \pm \iota}s^* \left( 1 \pm O\left( \frac{1}{\log^5(d)} \right) \right) \times \left( A_{+,r^*}^{*(t)} \left| S_+^{*(0)}(\boldsymbol{v}_+) \right| + A_{+,c,r^*}^{*(t)} \left| S_+^{*(0)}(\boldsymbol{v}_{+,c}) \right| \right) \right\}$$

$$\times [1/3, 1](1 \pm s^{*-1/3}) \frac{s^*}{2k_+P} \left( \sqrt{1 \pm \iota} \pm O\left( \frac{1}{\log^9(d)} \right) \right), \tag{105}$$

Relying on the induction hypothesis, we can reduce the above expressions to

$$
\begin{aligned}
&\Delta A_{+,r^*}^{*(t)} \\
&= \eta \sum_{c=1}^{k_+} \exp\left\{ -(1 \pm s^{*-1/3})\sqrt{1 \pm \iota}\left(1 \pm O\left(\frac{1}{\log^5(d)}\right)\right)\left(1 \pm O\left(\frac{1}{k_+}\right)\right) s^* A_{+,r^*}^{*(t)} \left|S_+^{*(0)}(\boldsymbol{v}_+)\right| \right\} \\
&\quad \times [1/3, 1](1 \pm s^{*-1/3})\frac{s^*}{2k_+ P}\left(\sqrt{1 \pm \iota} \pm O\left(\frac{1}{\log^9(d)}\right)\right) \\
&= \eta \exp\left\{ -(1 \pm s^{*-1/3})\sqrt{1 \pm \iota}\left(1 \pm O\left(\frac{1}{\log^5(d)}\right)\right)\left(1 \pm O\left(\frac{1}{k_+}\right)\right) s^* A_{+,r^*}^{*(t)} \left|S_+^{*(0)}(\boldsymbol{v}_+)\right| \right\} \\
&\quad \times \Theta(1) \times \frac{s^*}{2P},
\end{aligned}
\tag{106}
$$

and for any $c \in [k_+]$,

$$
\begin{aligned}
&\Delta A_{+,c,r^*}^{*(t)} \\
&= \eta \exp\left\{ -(1 \pm s^{*-1/3})\sqrt{1 \pm \iota}\left(1 \pm O\left(\frac{1}{\log^5(d)}\right)\right)\left(1 \pm O\left(\frac{1}{k_+}\right)\right) s^* A_{+,r^*}^{*(t)} \left|S_+^{*(0)}(\boldsymbol{v}_+)\right| \right\} \\
&\quad \times \Theta(1) \times \frac{s^*}{2k_+ P}.
\end{aligned}
\tag{107}
$$

By invoking the property that $s^* A_{+,r^*}^{*(t)} \left|S_+^{*(0)}(\boldsymbol{v}_+)\right| \le O(\log(d))$, we find that for all $c \in [k_+]$,

$$
\begin{aligned}
\frac{\Delta A_{+,c,r^*}^{*(t)}}{\Delta A_{+,r^*}^{*(t)}} &= \exp\left\{ \pm O\left(\frac{1}{\log^5(d)}\right) s^* A_{+,r^*}^{*(t)} \left|S_+^{*(0)}(\boldsymbol{v}_+)\right| \right\} \times \Theta\left(\frac{1}{k_+}\right) \\
&= \left(1 \pm O\left(\frac{1}{\log^4(d)}\right)\right) \times \Theta\left(\frac{1}{k_+}\right) \\
&= \Theta\left(\frac{1}{k_+}\right).
\end{aligned}
\tag{108}
$$

Therefore, we can finish our induction step:

$$
\frac{A_{+,c,r^*}^{*(t+1)}}{A_{+,r^*}^{*(t+1)}} = \frac{A_{+,c,r^*}^{*(t)} + \Delta A_{+,c,r^*}^{*(t)}}{A_{+,r^*}^{*(t)} + \Delta A_{+,r^*}^{*(t)}} = \frac{A_{+,c,r^*}^{*(t)} + \Delta A_{+,c,r^*}^{*(t)}}{\Theta(k_+) \times \left(A_{+,c,r^*}^{*(t)} + \Delta A_{+,c,r^*}^{*(t)}\right)} = \Theta\left(\frac{1}{k_+}\right).
\tag{109}
$$

$\square$

**Lemma E.6.** *Let $T_{\Omega(1)}$ be the first point in time such that either $s^* A_{+,r^*}^{*(t)} \left|S_+^{*(0)}(\boldsymbol{v}_+)\right| \ge \Omega(1)$ or $s^* A_{-,r^*}^{*(t)} \left|S_-^{*(0)}(\boldsymbol{v}_-)\right| \ge \Omega(1)$. Then for any $t < T_{\Omega(1)}$,*

$$
\frac{A_{-,r^*}^{*(t)}}{A_{+,r^*}^{*(t)}} = \Theta(1)
\tag{110}
$$

*and for any $t \in [T_{\Omega(1)}, T_1]$,*

$$
\frac{A_{-,r^*}^{*(t)}}{A_{+,r^*}^{*(t)}}, \frac{A_{+,r^*}^{*(t)}}{A_{-,r^*}^{*(t)}} \ge \Omega\left(\frac{1}{\log(d)}\right).
\tag{111}
$$

*Proof.* This lemma is a consequence of Theorem D.1, Lemma E.2 and Lemma E.4.

Due to Theorem D.1, we already know that $\frac{A^{*(t)}_{-,r^*}}{A^{*(t)}_{+,r^*}} = \Theta(1)$ up to time $T_0$. In addition, with Lemma E.2 we know that before $s^* A^{*(t)}_{+,r^*} \left| S^{*(0)}_+(\boldsymbol{v}_+) \right| \geq \Omega(1)$, the loss term (on a +-class sample) $1 - \text{logit}^{(t)}_+(\boldsymbol{X}^{(t)}_n) = \Theta(1)$ (the same holds with the class signs flipped), in which case it is also easy to derive $\frac{A^{*(t)}_{-,r^*}}{A^{*(t)}_{+,r^*}} = \Theta(1)$ by noting that the update expressions $\Delta A^{*(t)}_{-,r^*}/\Delta A^{*(t)}_{+,r^*} = \Theta(1)$.

Beyond time $T_{\Omega(1)}$, by Lemma E.4, we know that $s^* A^{*(t)}_{+,r^*} \left| S^{*(0)}_+(\boldsymbol{v}_+) \right|, s^* A^{*(t)}_{-,r^*} \left| S^{*(0)}_-(\boldsymbol{v}_-) \right| \leq O(\log(d))$. With the understanding that $s^* A^{*(t)}_{+,r^*} \left| S^{*(0)}_+(\boldsymbol{v}_+) \right|, s^* A^{*(t)}_{-,r^*} \left| S^{*(0)}_-(\boldsymbol{v}_-) \right| \geq \Omega(1)$ beyond $T_{\Omega(1)}$ due to the monotonicity of these functions, and the property $\left| \frac{|S^{*(0)}_-(\boldsymbol{v}_-)|}{|S^{*(0)}_+(\boldsymbol{v}_+)|} - 1 \right| \leq O\left( \frac{1}{\log^5(d)} \right)$ from Proposition 1, the rest of the lemma follows. $\qquad\square$

**Lemma E.7.** *With probability at least $1 - O\left( \frac{mNPk_+t}{poly(d)} \right)$, for all $t \in [0, T_1]$ and all $(+, r) \in S^{*(0)}_+(\boldsymbol{v}_+)$,*

$$\frac{\Delta b^{(t)}_{+,r}}{\Delta A^{(t)}_{+,r}} = -\Theta\left( \frac{1}{\log^5(d)} \right). \tag{112}$$

*The same holds with the +-class signs replaced by the −-class signs.*

*Proof.* Choose any $(+, r) \in S^{*(0)}_+(\boldsymbol{v}_+)$.

The statement in this lemma for time period $t \in [0, T_0]$ follows easily from Theorem D.1 and its proof. Let us examine the period $t \in [T_0, T_1]$.

Based on Lemma E.2 and its proof and Lemma E.5, we know that for $t \in [T_0, T_1]$, with probability at least $1 - O\left( \frac{mNPk_+t}{poly(d)} \right)$,

$$\begin{aligned}
&\Delta A^{(t)}_{+,r} \\
&= \eta \exp\left\{ -(1 \pm s^{*-1/3})\sqrt{1 \pm \iota} \left( 1 \pm O\left( \frac{1}{\log^5(d)} \right) \right) \left( 1 \pm O\left( \frac{1}{k_+} \right) \right) s^* A^{*(t)}_{+,r^*} \left| S^{*(0)}_+(\boldsymbol{v}_+) \right| \right\} \\
&\quad \times (1 \pm s^{*-1/3}) \frac{s^*}{NP} \left( \sqrt{1 \pm \iota} \pm O\left( \frac{1}{\log^9(d)} \right) \right) \sum_{n=1}^{N} \mathbb{1}\{y_n = +\} \frac{\exp(F^{(t)}_-(\boldsymbol{X}^{(t)}_n))}{\exp\left( F^{(t)}_-(\boldsymbol{X}^{(t)}_n) - F^{(t)}_+(\boldsymbol{X}^{(t)}_n) \right) + 1}
\end{aligned} \tag{113}$$

Furthermore,

$$\begin{aligned}
&\Delta b^{(t)}_{+,r} \\
&= -\frac{\|\Delta \boldsymbol{w}^{(t)}_{+,r}\|_2}{\log^5(d)} \\
&= -\eta \frac{1}{\log^5(d)} \exp\left\{ -(1 \pm s^{*-1/3})\sqrt{1 \pm \iota} \left( 1 \pm O\left( \frac{1}{\log^5(d)} \right) \right) \left( 1 \pm O\left( \frac{1}{k_+} \right) \right) s^* A^{*(t)}_{+,r^*} \left| S^{*(0)}_+(\boldsymbol{v}_+) \right| \right\} \\
&\quad \times (1 \pm s^{*-1/3}) \frac{s^*}{NP} \left( 1 \pm \iota \pm \frac{1}{\log^9(d)} \right) \sum_{n=1}^{N} \mathbb{1}\{y_n = +\} \frac{\exp(F^{(t)}_-(\boldsymbol{X}^{(t)}_n))}{\exp\left( F^{(t)}_-(\boldsymbol{X}^{(t)}_n) - F^{(t)}_+(\boldsymbol{X}^{(t)}_n) \right) + 1}
\end{aligned} \tag{114}$$

With the understanding that $s^* A^{*(t)}_{+,r^*} \left| S^{*(0)}_+(\boldsymbol{v}_+) \right| \leq O(\log(d))$ from Lemma E.4 and the fact that $\frac{\exp(F^{(t)}_-(\boldsymbol{X}^{(t)}_n))}{\exp\left( F^{(t)}_-(\boldsymbol{X}^{(t)}_n) - F^{(t)}_+(\boldsymbol{X}^{(t)}_n) \right) + 1} = \Theta(1)$, we have

$$
\begin{aligned}
\frac{\Delta b_{+,r}^{(t)}}{\Delta A_{+,r}^{(t)}} &= -\Theta\left(\frac{1}{\log^5(d)}\right)\exp\left\{-\left(1\pm O\left(\frac{1}{\log^5(d)}\right)\right)s^* A_{+,r^*}^{*(t)}\left|S_+^{*(0)}(\boldsymbol{v}_+)\right|\right\} \\
&= -\Theta\left(\frac{1}{\log^5(d)}\right)\left(1\pm O\left(\frac{1}{\log^4(d)}\right)\right) \\
&= -\Theta\left(\frac{1}{\log^5(d)}\right).
\end{aligned}
\tag{115}
$$

$\qquad\qquad\square$

**Lemma E.8** (Probability of mistake on hard samples is high). *For all* $t \in [0, T_1]$, *given a hard test sample* $(\boldsymbol{X}_{hard}, y)$, $y' \neq y$,

$$
\mathbb{P}\left[F_y^{(T)}(\boldsymbol{X}_{hard}) \leq F_{y'}^{(T)}(\boldsymbol{X}_{hard})\right] \geq \Omega(1).
\tag{116}
$$

*Proof.* We first show that at time $t = 0$, the probability of the network making a mistake on hard test samples is $\Omega(1)$, then prove that for the rest of the time, i.e. $t \in (0, T_1]$, the model still makes mistake on hard test samples with probability $\Omega(1)$.

At time $t = 0$, by Lemma H.3, we know that for any $r \in [m]$, with probability $\Omega(1)$,

$$
\langle \boldsymbol{w}_{+,r}^{(0)}, \boldsymbol{\zeta}^* \rangle \geq \Omega(\sigma_0 \sigma_{\zeta^*} \sqrt{d}) \geq \Omega(\sigma_0 \text{polylog}(d)) \gg \Omega\left(\sigma_0 \sqrt{\log(d)}\right).
\tag{117}
$$

Relying on concentration of the binomial random variable, with probability at least $1 - e^{-\Omega(\text{polylog}(d))}$,

$$
\sum_{r=1}^m \sigma\left(\langle \boldsymbol{w}_{+,r}^{(0)}, \boldsymbol{\zeta}^* \rangle + b_{+,r}^{(0)}\right) \geq \Omega(m\sigma_0\sigma_{\zeta^*}\sqrt{d}),
\tag{118}
$$

which is asymptotically larger than the activation from the features, which, following from Proposition 1, is upper bounded by $O\left(\sigma_0\sqrt{\log(d)}s^* d^{c_0}\right)$. The same can be said for the "$-$" class. In other words,

$$
\begin{aligned}
&F_-^{(0)}(\boldsymbol{X}_{\text{hard}}) - F_+^{(0)}(\boldsymbol{X}_{\text{hard}}) > 0 \\
\iff &\left\{\sum_{r=1}^m \mathbb{1}\{\langle \boldsymbol{w}_{-,r}^{(0)}, \boldsymbol{\zeta}^* \rangle + b_{-,r}^{(0)} > 0\}\langle \boldsymbol{w}_{-,r}^{(0)}, \boldsymbol{\zeta}^* \rangle \right. \\
&\left. - \sum_{r=1}^m \mathbb{1}\{\langle \boldsymbol{w}_{+,r}^{(0)}, \boldsymbol{\zeta}^* \rangle + b_{+,r}^{(0)} > 0\}\langle \boldsymbol{w}_{+,r}^{(0)}, \boldsymbol{\zeta}^* \rangle\right\}(1 \pm o(1)) > 0
\end{aligned}
\tag{119}
$$

which clearly holds with probability $\Omega(1)$.

Now consider $t \in (0, T_1]$.

During this period of time, by Theorem D.1 and Lemma E.2, we note that for any $c \in [k_+]$ and $(+, r) \in S_+^{*(0)}(\boldsymbol{v}_{+,c})$, $\Delta\boldsymbol{\zeta}_{+,r}^{(t)} \sim \mathcal{N}(\boldsymbol{0}, \sigma_{\Delta\zeta_{+,r}}^{(t)2}\boldsymbol{I}_d)$, with $\sigma_{\Delta\zeta_{+,r}}^{(t)} = \Theta\left(\Delta A_{+,c,r}^{(t)}\sqrt{\frac{2k_+}{s^*N}}\sigma_\zeta\right)$. The same can be said for $(+, r) \in S_+^{*(0)}(\boldsymbol{v}_+)$, although with the $\Delta A_{+,c,r}^{(t)}\sqrt{\frac{2k_+}{s^*N}}$ factor replaced by $\Delta A_{+,r}^{(t)}\sqrt{\frac{2}{s^*N}}$. Also from the proofs of Theorem D.1 and Lemma E.2, and using the property $|\mathcal{U}_{+,r}^{(0)}| \leq O(1)$ from Proposition 1, we know that for all neurons, the updates to the neurons also take the feature-plus-Gaussian-noise form of $\sum_{\boldsymbol{v}' \in \mathcal{U}_{+,r}^{(0)}} c^{(t)}(\boldsymbol{v}')\boldsymbol{v}' + \Delta\boldsymbol{\zeta}_{+,r}^{(t)}$, with $c^{(t)}(\boldsymbol{v}') \leq \left(1 + O\left(\frac{1}{\log^5(d)}\right)\right)\Delta A_{+,c,r}^{(t)}$ if $\boldsymbol{v}' = \boldsymbol{v}_{+,c}$ for some $c \in [k_+]$, or $c^{(t)}(\boldsymbol{v}') \leq \left(1 + O\left(\frac{1}{\log^5(d)}\right)\right)\Delta A_{+,r}^{(t)}$ if $\boldsymbol{v}' = \boldsymbol{v}_+$ (because the $\boldsymbol{v}'$ component of a $\boldsymbol{v}'$-singleton neuron's update is already the maximum possible).

Moreover, if $\boldsymbol{v}_+ \in \mathcal{U}_{+,r}^{(0)}$, then $\sigma_{\Delta\zeta_{+,r}}^{(t)} \leq O\left(\Delta A_{+,r}^{(t)}\sqrt{\frac{2}{s^*N}}\sigma_\zeta\right) + O\left(\Delta A_{+,c,r}^{(t)}\sqrt{\frac{2k_+}{s^*N}}\sigma_\zeta\right) \leq O\left(\Delta A_{+,r}^{(t)}\sqrt{\frac{2}{s^*N}}\sigma_\zeta\right)$,

otherwise, if $\mathcal{U}_{+,r}^{(0)}$ only contains the fine-grained features, then $\sigma_{\Delta\zeta_{+,r}}^{(t)} \leq O\left(\Delta A_{+,c,r}^{(t)}\sqrt{\frac{2k_+}{s^*N}}\sigma_\zeta\right)$.

With the understanding that only neurons in $S_y^{(0)}(\boldsymbol{v}_y)$ and $S_y^{(0)}(\boldsymbol{v}_{y,c})$ can possibly activate on the feature patches of a sample when $t \leq T_1$ (coming from Theorem F.1), we have

$$
\begin{aligned}
F_+^{(t)}(\boldsymbol{X}_{\text{hard}}) \leq &\sum_{(+,r)\in S_+^{(0)}(\boldsymbol{v}_{+,c})} \sum_{p\in\mathcal{P}(\boldsymbol{X}_{\text{hard}};\boldsymbol{v}_{+,c})} \sigma\left(\langle \boldsymbol{w}_{+,r}^{(0)} + \sum_{\tau=0}^{t-1}\Delta\boldsymbol{w}_{+,r}^{(\tau)}, \sqrt{1\pm\iota}\boldsymbol{v}_{+,c} + \boldsymbol{\zeta}_p\rangle + b_{+,r}^{(t)}\right) \\
&+ \sum_{r\in[m]} \sigma\left(\langle \boldsymbol{w}_{+,r}^{(0)} + \sum_{\tau=0}^{t-1}\Delta\boldsymbol{w}_{+,r}^{(\tau)}, \boldsymbol{\zeta}^*\rangle + b_{+,r}^{(t)}\right) \\
&+ \sum_{(+,r)\in S_+^{(0)}(\boldsymbol{v}_-)} \sum_{p\in\mathcal{P}(\boldsymbol{X}_{\text{hard}};\boldsymbol{v}_-)} \sigma\left(\langle \boldsymbol{w}_{+,r}^{(0)} + \sum_{\tau=0}^{t-1}\Delta\boldsymbol{w}_{+,r}^{(\tau)}, \alpha_p^\dagger\boldsymbol{v}_- + \boldsymbol{\zeta}_p\rangle + b_{+,r}^{(t)}\right)
\end{aligned}
\tag{120}
$$

To further refine this upper bound, we first note that with probability at least $1 - O\left(\frac{mNPk_+t}{\text{poly}(d)}\right)$, the following holds with arbitrary choice of $(+,r^*) \in S_+^{(0)}(\boldsymbol{v}_{+,c})$:

$$
\sum_{(+,r)\in S_+^{(0)}(\boldsymbol{v}_{+,c})} \sum_{p\in\mathcal{P}(\boldsymbol{X}_{\text{hard}};\boldsymbol{v}_{+,c})} \langle\sum_{\tau=0}^{t-1}\Delta\boldsymbol{w}_{+,r}^{(\tau)}, \sqrt{1\pm\iota}\boldsymbol{v}_{+,c} + \boldsymbol{\zeta}_p\rangle \leq O\left(s^*\left|S_+^{(0)}(\boldsymbol{v}_{+,c})\right|\sum_{\tau=0}^{t-1}\Delta A_{+,c,r^*}^{(\tau)}\right)
\tag{121}
$$

Invoking Lemma E.5, we obtain (for arbitrary $(+,r^*) \in S_+^{(0)}(\boldsymbol{v}_+)$):

$$
\sum_{(+,r)\in S_+^{(0)}(\boldsymbol{v}_{+,c})} \sum_{p\in\mathcal{P}(\boldsymbol{X}_{\text{hard}};\boldsymbol{v}_{+,c})} \langle\sum_{\tau=0}^{t-1}\Delta\boldsymbol{w}_{+,r}^{(\tau)}, \sqrt{1\pm\iota}\boldsymbol{v}_{+,c} + \boldsymbol{\zeta}_p\rangle \leq O\left(\frac{1}{k_+}s^*\left|S_+^{(0)}(\boldsymbol{v}_{+,c})\right|\sum_{\tau=0}^{t-1}\Delta A_{+,r^*}^{(\tau)}\right)
\tag{122}
$$

Let us examine the term $\sum_{r\in[m]} \sigma\left(\langle \boldsymbol{w}_{+,r}^{(0)} + \sum_{\tau=0}^{t-1}\Delta\boldsymbol{w}_{+,r}^{(\tau)}, \boldsymbol{\zeta}^*\rangle + b_{+,r}^{(t)}\right)$ more carefully. First of all, denoting $S_+^{(0)} = \cup_{c=1}^{k_+}S_+^{(0)}(\boldsymbol{v}_{+,c})\cup\cup_{c=1}^{k_-}S_+^{(0)}(\boldsymbol{v}_{-,c})\cup S_+^{(0)}(\boldsymbol{v}_+)\cup S_+^{(0)}(\boldsymbol{v}_-)$, neurons $(+,r) \notin S_+^{(0)}$ cannot receive any update at all during training due to Theorem F.1. Therefore we can rewrite the term

$$
\begin{aligned}
&\sum_{r\in[m]} \sigma\left(\langle \boldsymbol{w}_{+,r}^{(0)} + \sum_{\tau=0}^{t-1}\Delta\boldsymbol{w}_{+,r}^{(\tau)}, \boldsymbol{\zeta}^*\rangle + b_{+,r}^{(t)}\right) \\
&= \sum_{(+,r)\in S_+^{(0)}} \sigma\left(\langle \boldsymbol{w}_{+,r}^{(0)} + \sum_{\tau=0}^{t-1}\Delta\boldsymbol{w}_{+,r}^{(\tau)}, \boldsymbol{\zeta}^*\rangle + b_{+,r}^{(t)}\right) + \sum_{(+,r)\notin S_+^{(0)}} \sigma\left(\langle \boldsymbol{w}_{+,r}^{(0)}, \boldsymbol{\zeta}^*\rangle + b_{+,r}^{(0)}\right)
\end{aligned}
\tag{123}
$$

Relying on Corollary F.1.1, we know

$$
\sum_{\tau=0}^{t-1}\Delta b_{+,r}^{(\tau)} < \sum_{\tau=0}^{t-1}-\Omega\left(\frac{\text{polylog}(d)}{\log^5(d)}\right)\left|\langle\Delta\boldsymbol{w}_{+,r}^{(\tau)}, \boldsymbol{\zeta}^*\rangle\right|.
\tag{124}
$$

Therefore, we know that for $r \in [m]$,

$$
\sum_{\tau=0}^{t-1}\langle\Delta\boldsymbol{w}_{+,r}^{(\tau)}, \boldsymbol{\zeta}^*\rangle + \Delta b_{+,r}^{(\tau)} \leq 0
\tag{125}
$$

As a consequence, we can write the naive upper bound

$$
\begin{aligned}
\sum_{r \in [m]} & \sigma \left( \langle \boldsymbol{w}_{+,r}^{(0)} + \sum_{\tau=0}^{t-1} \Delta \boldsymbol{w}_{+,r}^{(\tau)}, \boldsymbol{\zeta}^* \rangle + b_{+,r}^{(t)} \right) \\
& \leq \sum_{(+,r) \in S_+^{(0)}} \sigma \left( \langle \boldsymbol{w}_{+,r}^{(0)}, \boldsymbol{\zeta}^* \rangle + b_{+,r}^{(0)} \right) + \sum_{(+,r) \notin S_+^{(0)}} \sigma \left( \langle \boldsymbol{w}_{+,r}^{(0)}, \boldsymbol{\zeta}^* \rangle + b_{+,r}^{(0)} \right) \\
& = \sum_{r \in [m]} \sigma \left( \langle \boldsymbol{w}_{+,r}^{(0)}, \boldsymbol{\zeta}^* \rangle + b_{+,r}^{(0)} \right)
\end{aligned}
\tag{126}
$$

Additionally, due to Theorem F.1 (and its proof), we know that

$$
\begin{aligned}
\sum_{(+,r) \in S_+^{(0)}(\boldsymbol{v}_-)} \sum_{p \in \mathcal{P}(\boldsymbol{X}_{\text{hard}}; \boldsymbol{v}_-)} & \sigma \left( \langle \boldsymbol{w}_{+,r}^{(0)} + \sum_{\tau=0}^{t-1} \Delta \boldsymbol{w}_{+,r}^{(\tau)}, \alpha_p^\dagger \boldsymbol{v}_- + \boldsymbol{\zeta}_p \rangle + b_{+,r}^{(t)} \right) \\
& \leq \sum_{(+,r) \in S_+^{(0)}(\boldsymbol{v}_-)} \sum_{p \in \mathcal{P}(\boldsymbol{X}_{\text{hard}}; \boldsymbol{v}_-)} \sigma \left( \langle \boldsymbol{w}_{+,r}^{(0)}, \alpha_p^\dagger \boldsymbol{v}_- + \boldsymbol{\zeta}_p \rangle + b_{+,r}^{(0)} \right)
\end{aligned}
\tag{127}
$$

It follows that

$$
\begin{aligned}
& F_+^{(t)}(\boldsymbol{X}_{\text{hard}}) \\
& \leq O \left( \frac{1}{k_+} s^* \left| S_+^{(0)}(\boldsymbol{v}_{+,c}) \right| \sum_{\tau=0}^{t-1} \Delta A_{+,r^*}^{(\tau)} \right) + \sum_{(+,r) \in S_+^{(0)}(\boldsymbol{v}_{+,c})} \sum_{p \in \mathcal{P}(\boldsymbol{X}_{\text{hard}}; \boldsymbol{v}_{+,c})} \left| \langle \boldsymbol{w}_{+,r}^{(0)}, \sqrt{1 \pm \iota} \boldsymbol{v}_{+,c} + \boldsymbol{\zeta}_p \rangle \right| \\
& \quad + \sum_{r \in [m]} \sigma \left( \langle \boldsymbol{w}_{+,r}^{(0)}, \boldsymbol{\zeta}^* \rangle + b_{+,r}^{(0)} \right) + \sum_{(+,r) \in S_+^{(0)}(\boldsymbol{v}_-)} \sum_{p \in \mathcal{P}(\boldsymbol{X}_{\text{hard}}; \boldsymbol{v}_-)} \sigma \left( \langle \boldsymbol{w}_{+,r}^{(0)}, \alpha_p^\dagger \boldsymbol{v}_- + \boldsymbol{\zeta}_p \rangle + b_{+,r}^{(0)} \right)
\end{aligned}
\tag{128}
$$

On the other hand, for the "$-$" neurons, denoting $S_-^{(0)} = \cup_{c=1}^{k_+} S_-^{(0)}(\boldsymbol{v}_{+,c}) \cup \cup_{c=1}^{k_-} S_-^{(0)}(\boldsymbol{v}_{-,c}) \cup S_-^{(0)}(\boldsymbol{v}_+) \cup S_-^{(0)}(\boldsymbol{v}_-)$,

$$
\begin{aligned}
F_-^{(t)}(\boldsymbol{X}_{\text{hard}}) \geq & \sum_{(+,r) \in S_-^{*(0)}(\boldsymbol{v}_-)} \sum_{p \in \mathcal{P}(\boldsymbol{X}_{\text{hard}}; \boldsymbol{v}_-)} \sigma \left( \langle \boldsymbol{w}_{-,r}^{(0)} + \sum_{\tau=0}^{t-1} \Delta \boldsymbol{w}_{-,r}^{(\tau)}, \alpha_p^\dagger \boldsymbol{v}_- + \boldsymbol{\zeta}_p \rangle + b_{+,r}^{(t)} \right) \\
& + \sum_{(+,r) \notin S_-^{(0)}} \sigma \left( \langle \boldsymbol{w}_{-,r}^{(0)}, \boldsymbol{\zeta}^* \rangle + b_{+,r}^{(0)} \right),
\end{aligned}
\tag{129}
$$

note that the last line is true because neurons outside the set $S_-^{(0)}$ cannot receive any update during training with probability at least $1 - O \left( \frac{mNPk_+ t}{\text{poly}(d)} \right)$ due to Theorem F.1. Estimating the activation value of the neurons from $S_-^{*(0)}(\boldsymbol{v}_-)$ on the feature noise patches requires some care. We define time $t_-$ to be the first point in time such that any $(-, r^*) \in S_-^{*(0)}(\boldsymbol{v}_-)$ satisfies $\sum_{\tau=0}^{t_-} \Delta A_{-,r^*}^{(\tau)} \geq \sigma_0 \log^5(d)$, and beyond this point in time, i.e. for $t \in [t_-, T_1]$, the neurons in $S_-^{*(0)}(\boldsymbol{v}_-)$ have to activate with high probability, since

$$
\langle \boldsymbol{w}_{-,r}^{(0)} + \sum_{\tau=0}^{t-1} \Delta \boldsymbol{w}_{-,r}^{(\tau)}, \alpha_p^\dagger \boldsymbol{v}_- + \boldsymbol{\zeta}_p \rangle + b_{+,r}^{(t)} \geq \left( 1 - O \left( \frac{1}{\log^5(d)} \right) \right) \sigma_0 \log^5(d) / \log^4(d) - O(\sigma_0 \sqrt{\log(d)})
\tag{130}
$$

$$
> 0.
$$

Now we can proceed to prove the lemma for $t \in (0, T_1]$ by combining the above estimates for $F_+^{(t)}(\boldsymbol{X}_{\text{hard}})$ and $F_-^{(t)}(\boldsymbol{X}_{\text{hard}})$.

For $t \in (0, t_-]$, relying argument similar to the situation of $t = 0$ and the fact that $m - |S_-^{(0)}| = (1 - o(1))m$,

$$
\begin{aligned}
&\Bigg\{ \sum_{(+,r) \notin S_-^{(0)}} \mathbb{1}\{\langle \boldsymbol{w}_{-,r}^{(0)}, \boldsymbol{\zeta}^* \rangle + b_{-,r}^{(0)} > 0\} \langle \boldsymbol{w}_{-,r}^{(0)}, \boldsymbol{\zeta}^* \rangle \\
&\quad - \sum_{r=1}^{m} \mathbb{1}\{\langle \boldsymbol{w}_{+,r}^{(0)}, \boldsymbol{\zeta}^* \rangle + b_{+,r}^{(0)} > 0\} \langle \boldsymbol{w}_{+,r}^{(0)}, \boldsymbol{\zeta}^* \rangle \Bigg\} (1 \pm o(1)) > 0 \\
&\implies F_-^{(t)}(\boldsymbol{X}_{\text{hard}}) - F_+^{(t)}(\boldsymbol{X}_{\text{hard}}) > 0
\end{aligned}
\tag{131}
$$

which has to be true with probability $\Omega(1)$.

On the other hand, with $t \in (t_-, T_1]$, we have

$$
\begin{aligned}
&F_-^{(t)}(\boldsymbol{X}_{\text{hard}}) - F_+^{(t)}(\boldsymbol{X}_{\text{hard}}) \\
&\geq \Bigg\{ \sum_{\tau=0}^{t-1} \left(1 - O\left(\frac{1}{\log^5(d)}\right)\right) s^\dagger |S_-^{*(0)}(\boldsymbol{v}_-)| \Delta A_{-,r^*}^{(\tau)} - O(\sigma_0\sqrt{\log(d)}) \\
&\quad - O\left(\frac{1}{k_+} s^* \left| S_+^{(0)}(\boldsymbol{v}_{+,c}) \right| \sum_{\tau=0}^{t-1} \Delta A_{+,r^*}^{(\tau)} \right) \Bigg\} \\
&\quad + \Bigg\{ \sum_{(+,r) \notin S_-^{(0)}} \sigma\left(\langle \boldsymbol{w}_{-,r}^{(0)}, \boldsymbol{\zeta}^* \rangle + b_{+,r}^{(0)}\right) - \sum_{(+,r) \in S_+^{(0)}(\boldsymbol{v}_{+,c})} \sum_{p \in \mathcal{P}(\boldsymbol{X}_{\text{hard}}; \boldsymbol{v}_{+,c})} \left| \langle \boldsymbol{w}_{+,r}^{(0)}, \sqrt{1 \pm \iota} \boldsymbol{v}_{+,c} + \boldsymbol{\zeta}_p \rangle \right| \\
&\quad - \sum_{r \in [m]} \sigma\left(\langle \boldsymbol{w}_{+,r}^{(0)}, \boldsymbol{\zeta}^* \rangle + b_{+,r}^{(0)}\right) - \sum_{(+,r) \in S_+^{(0)}(\boldsymbol{v}_-)} \sum_{p \in \mathcal{P}(\boldsymbol{X}_{\text{hard}}; \boldsymbol{v}_-)} \sigma\left(\langle \boldsymbol{w}_{+,r}^{(0)}, \alpha_p^\dagger \boldsymbol{v}_- + \boldsymbol{\zeta}_p \rangle + b_{+,r}^{(0)}\right) \Bigg\}
\end{aligned}
\tag{132}
$$

Let us begin analyzing the first $\{\cdot\}$ bracket.

By Proposition 1 we know that $\left| S_-^{*(0)}(\boldsymbol{v}_-) \right| = (1 \pm O(1/\log^5(d))) \left| S_+^{(0)}(\boldsymbol{v}_{+,c}) \right|$, and by Lemma E.5, we know that $\Delta A_{+,r^*}^{(\tau)} \leq O(\log(d)\Delta A_{-,r^*}^{(\tau)})$, therefore,

$$
\begin{aligned}
O\left(\frac{1}{k_+} s^* \left| S_+^{(0)}(\boldsymbol{v}_{+,c}) \right| \sum_{\tau=0}^{t-1} \Delta A_{+,r^*}^{(\tau)} \right) &\leq O\left(\frac{\log(d)}{k_+} s^* \left| S_-^{*(0)}(\boldsymbol{v}_-) \right| \sum_{\tau=0}^{t-1} \Delta A_{-,r^*}^{(\tau)} \right) \\
&\ll \sum_{\tau=0}^{t-1} \left(1 - O\left(\frac{1}{\log^5(d)}\right)\right) s^\dagger |S_-^{*(0)}(\boldsymbol{v}_-)| \Delta A_{-,r^*}^{(\tau)} - O(\sigma_0\sqrt{\log(d)})
\end{aligned}
\tag{133}
$$

Therefore, we obtained the simpler lower bound

$$
\begin{aligned}
&F_-^{(t)}(\boldsymbol{X}_{\text{hard}}) - F_+^{(t)}(\boldsymbol{X}_{\text{hard}}) \\
&\geq \Bigg\{ \sum_{(+,r) \notin S_-^{(0)}} \sigma\left(\langle \boldsymbol{w}_{-,r}^{(0)}, \boldsymbol{\zeta}^* \rangle + b_{+,r}^{(0)}\right) - \sum_{(+,r) \in S_+^{(0)}(\boldsymbol{v}_{+,c})} \sum_{p \in \mathcal{P}(\boldsymbol{X}_{\text{hard}}; \boldsymbol{v}_{+,c})} \left| \langle \boldsymbol{w}_{+,r}^{(0)}, \sqrt{1 \pm \iota} \boldsymbol{v}_{+,c} + \boldsymbol{\zeta}_p \rangle \right| \\
&\quad - \sum_{r \in [m]} \sigma\left(\langle \boldsymbol{w}_{+,r}^{(0)}, \boldsymbol{\zeta}^* \rangle + b_{+,r}^{(0)}\right) - \sum_{(+,r) \in S_+^{(0)}(\boldsymbol{v}_-)} \sum_{p \in \mathcal{P}(\boldsymbol{X}_{\text{hard}}; \boldsymbol{v}_-)} \sigma\left(\langle \boldsymbol{w}_{+,r}^{(0)}, \alpha_p^\dagger \boldsymbol{v}_- + \boldsymbol{\zeta}_p \rangle + b_{+,r}^{(0)}\right) \Bigg\}
\end{aligned}
\tag{134}
$$

which is greater than 0 with probability $\Omega(1)$ (by relying on an argument almost identical to the $t = 0$ case again, and noting that $m - |S_-^{(0)}| = (1 - o(1))m$). This concludes the proof.

$\square$

**Lemma E.9** (Probability of mistake on easy samples is low after training). *For $t \in [T_{1,1}, T_1]$, given an easy test sample $(\boldsymbol{X}_{easy}, y)$,*

$$\mathbb{P}\left[F_y^{(T)}(\boldsymbol{X}_{easy}) \leq F_{y'}^{(T)}(\boldsymbol{X}_{easy})\right] \leq o(1). \tag{135}$$

*Proof.* Without loss of generality, assume the true label of $\boldsymbol{X}_{\text{easy}}$ is $+1$. Assume $t \geq T_{1,1}$.

Firstly, conditioning on the events of Theorem F.1, the following upper bound on $F_{-}^{(t)}(\boldsymbol{X}_{\text{easy}})$ holds with probability at least $1 - O\left(\frac{m}{\text{poly}(d)}\right)$:

$$
\begin{aligned}
F_{-}^{(t)}\left(\boldsymbol{X}_{\text{easy}}\right) &= \sum_{(-,r)\in S_{-}^{(0)}(\boldsymbol{v}_+)} \sum_{p\in\mathcal{P}(\boldsymbol{X}_{\text{easy}};\boldsymbol{v}_+)} \sigma\left(\langle \boldsymbol{w}_{-,r}^{(t)}, \sqrt{1\pm\iota}\boldsymbol{v}_+ + \boldsymbol{\zeta}_p\rangle + b_{-,r}^{(t)}\right) \\
&\quad + \sum_{(-,r)\in S_{-}^{(0)}(\boldsymbol{v}_{+,c})} \sum_{p\in\mathcal{P}(\boldsymbol{X}_{\text{easy}};\boldsymbol{v}_{+,c})} \sigma\left(\langle \boldsymbol{w}_{-,r}^{(t)}, \sqrt{1\pm\iota}\boldsymbol{v}_{+,c} + \boldsymbol{\zeta}_p\rangle + b_{-,r}^{(t)}\right) \\
&\leq \sum_{(-,r)\in S_{-}^{(0)}(\boldsymbol{v}_+)} \sum_{p\in\mathcal{P}(\boldsymbol{X}_{\text{easy}};\boldsymbol{v}_+)} \sigma\left(\langle \boldsymbol{w}_{-,r}^{(0)}, \sqrt{1\pm\iota}\boldsymbol{v}_+ + \boldsymbol{\zeta}_p\rangle + b_{-,r}^{(0)}\right) \\
&\quad + \sum_{(-,r)\in S_{-}^{(0)}(\boldsymbol{v}_{+,c})} \sum_{p\in\mathcal{P}(\boldsymbol{X}_{\text{easy}};\boldsymbol{v}_{+,c})} \sigma\left(\langle \boldsymbol{w}_{-,r}^{(0)}, \sqrt{1\pm\iota}\boldsymbol{v}_{+,c} + \boldsymbol{\zeta}_p\rangle + b_{-,r}^{(0)}\right) \\
&< O\left(s^* d^{c_0}\sigma_0\right) \\
&\leq o(1),
\end{aligned}
\tag{136}
$$

and on the other hand,

$$
\begin{aligned}
F_{+}^{(t)}\left(\boldsymbol{X}_{\text{easy}}\right) &\geq \sum_{(+,r)\in S_{+}^{*(0)}(\boldsymbol{v}_+)} \sum_{p\in\mathcal{P}(\boldsymbol{X}_{\text{easy}};\boldsymbol{v}_+)} \sigma\left(\langle \boldsymbol{w}_{+,r}^{(t)}, \sqrt{1\pm\iota}\boldsymbol{v}_+ + \boldsymbol{\zeta}_p\rangle + b_{+,r}^{(t)}\right) \\
&\quad + \sum_{(+,r)\in S_{+}^{*(0)}(\boldsymbol{v}_{+,c})} \sum_{p\in\mathcal{P}(\boldsymbol{X}_{\text{easy}};\boldsymbol{v}_{+,c})} \sigma\left(\langle \boldsymbol{w}_{+,r}^{(t)}, \sqrt{1\pm\iota}\boldsymbol{v}_{+,c} + \boldsymbol{\zeta}_p\rangle + b_{+,r}^{(t)}\right) \\
&> \Omega(1).
\end{aligned}
\tag{137}
$$

Therefore, $F_{+}^{(t)}(\boldsymbol{X}_{\text{easy}}) \gg F_{-}^{(t)}(\boldsymbol{X}_{\text{easy}})$, which completes the proof. □

**Lemma E.10** (Jr. & John W. Wrench (1971)). *The partial sum of harmonic series satisfies the following identity:*

$$\sum_{k=1}^{n-1} \frac{1}{k} = \log(n) + \mathcal{E} - \frac{1}{2n} - \epsilon_n \tag{138}$$

*where $\mathcal{E}$ is the Euler–Mascheroni constant (approximately 0.58), and $\epsilon_n \in [0, 1/8n^2]$.*

# F   Coarse-grained SGD, Poly-time properties

In this section, set $T_e \in \mathrm{poly}(d)$.

Please note that we are performing stochastic gradient descent on easy samples only.

**Theorem F.1.** *Fix any $t \in [0, T_e]$.*

1. *(Non-activation invariance) For any $\tau \geq t$, with probability at least $1 - O\left(\frac{mk_+ NPt}{poly(d)}\right)$, any feature $v \in \{v_{+,c}\}_{c=1}^{k_+} \cup \{v_{-,c}\}_{c=1}^{k_-} \cup \{v_+, v_-\}$, any $t' \leq t$, $(+, r) \notin S_+^{(0)}(v)$ and $v$-dominated patch sample $x_{n,p}^{(\tau)} = \alpha_{n,p}^{(\tau)} v + \zeta_{n,p}^{(\tau)}$, the following holds:*

$$\sigma\left(\langle w_{+,r}^{(t')}, x_{n,p}^{(\tau)}\rangle + b_{+,r}^{(t')}\right) = 0 \tag{139}$$

2. *(Non-activation on noise patches) For any $\tau \geq t$, with probability at least $1 - O\left(\frac{mNPt}{poly(d)}\right)$, for every $t' \leq t$, $r \in [m]$ and noise patch $x_{n,p}^{(\tau)} = \zeta_{n,p}^{(\tau)}$, the following holds:*

$$\sigma\left(\langle w_{+,r}^{(t')}, x_{n,p}^{(\tau)}\rangle + b_{+,r}^{(t')}\right) = 0 \tag{140}$$

3. *(Off-diagonal nonpositive growth) For any $\tau \geq t$, with probability at least $1 - O\left(\frac{mk_+ NPt}{poly(d)}\right)$, for any $t' \leq t$, any feature $v \in \{v_{-,c}\}_{c=1}^{k_-} \cup \{v_-\}$, any $(+, r) \in S_+^{(0)}(v)$ and $v$-dominated patch $x_{n,p}^{(\tau)} = \alpha_{n,p}^{(\tau)} v + \zeta_{n,p}^{(\tau)}$, $\sigma\left(\langle w_{+,r}^{(t')}, x_{n,p}^{(\tau)}\rangle + b_{+,r}^{(t')}\right) \leq \sigma\left(\langle w_{+,r}^{(0)}, x_{n,p}^{(\tau)}\rangle + b_{+,r}^{(0)}\right).$*

*Proof.* **Base case $t = 0$.**

*1. (Nonactivation invariance)*

Choose any $\tau \geq 0$, $v^*$ from the set $\{v_{+,c}\}_{c=1}^{k_+} \cup \{v_{-,c}\}_{c=1}^{k_-} \cup \{v_+, v_-\}$. We will work with neuron sets in the "$+$" class in this proof; the "$-$"-class case can be handled in the same way.

First, we need to show that, for every $n$ such that $|\mathcal{P}(X_n^{(\tau)}; v^*)| > 0$ and $p \in \mathcal{P}(X_n^{(\tau)}; v^*)$, for every $(+, r)$ neuron index,

$$\langle w_{+,r}^{(0)}, v^*\rangle < \sigma_0\sqrt{4 + 2c_0}\sqrt{\log(d) - \frac{1}{\log^5(d)}} \implies \sigma\left(\langle w_{+,r}^{(0)}, x_{n,p}^{(\tau)}\rangle + b_{+,r}^{(0)}\right) = 0 \tag{141}$$

This is indeed true. The following holds with probability at least $1 - O\left(\frac{mNP}{poly(d)}\right)$ for all $(+, r) \notin S_+^{(0)}(v)$ and all such $x_{n,p}^{(\tau)}$:

$$
\begin{aligned}
\langle w_{+,r}^{(0)}, x_{n,p}^{(\tau)}\rangle + b_{+,r}^{(0)} \leq & \sigma_0\sqrt{1+\iota}\sqrt{(4 + 2c_0)(\log(d) - 1/\log^5(d))} + O\left(\frac{\sigma_0}{\log^9(d)}\right) - \sqrt{4 + 2c_0}\sqrt{\log(d)}\sigma_0 \\
= & \sigma_0\left(\frac{(4 + 2c_0)(1 + \iota)(\log(d) - 1/\log^5(d)) - (4 + 2c_0)\log(d)}{\sqrt{(4 + 2c_0)(\log(d) - 1/\log^5(d))} + \sqrt{4 + 2c_0}\sqrt{\log(d)}} + O\left(\frac{1}{\log^9(d)}\right)\right) \\
= & \sigma_0\left(\frac{(4 + 2c_0)\iota\log(d) - (1 + \iota)/\log^5(d)}{\sqrt{(4 + 2c_0)(\log(d) - 1/\log^5(d))} + \sqrt{4 + 2c_0}\sqrt{\log(d)}} + O\left(\frac{1}{\log^9(d)}\right)\right) \\
< & 0,
\end{aligned}
\tag{142}
$$

The first equality holds by utilizing the identity $a - b = \frac{a^2 - b^2}{a + b}$. As a consequence, $\sigma(\langle w_{+,r}^{(0)}, x_{n,p}^{(\tau)}\rangle + b_{+,r}^{(0)}) = 0$.

2. *(Non-activation on noise patches)* Invoking Lemma H.3, for any $\tau \geq 0$, with probability at least $1 - O\left(\frac{mNP}{\text{poly}(d)}\right)$, we have for all possible choices of $r \in [m]$ and the noise patches $\boldsymbol{x}_{n,p}^{(\tau)} = \boldsymbol{\zeta}_{n,p}^{(\tau)}$:

$$\left| \langle \boldsymbol{w}_{+,r}^{(0)}, \boldsymbol{\zeta}_{n,p}^{(\tau)} \rangle \right| \leq O(\sigma_0 \sigma_\zeta \sqrt{d \log(d)}) \leq O\left(\frac{\sigma_0}{\log^9(d)}\right) \ll b_{+,r}^{(0)}. \tag{143}$$

Therefore, no neuron can activate on the noise patches at time $t = 0$.

3. *(Off-diagonal nonpositive growth)* This point is trivially true at $t = 0$.

**Inductive step**: we assume the induction hypothesis for $t \in [0, T]$ (with $T < T_e$ of course), and prove the statements for $t = T + 1$.

*1. (Nonactivation invariance)*

Choose any $\boldsymbol{v}^*$ from the set $\{\boldsymbol{v}_{+,c}\}_{c=1}^{k_+} \cup \{\boldsymbol{v}_{-,c}\}_{c=1}^{k_-} \cup \{\boldsymbol{v}_+, \boldsymbol{v}_-\}$. We will work with neuron sets in the "+" class in this proof; the "−"-class case can be handled in the same way.

We need to prove that given $\tau \geq T + 1$, with probability at least $1 - O\left(\frac{mNP(T+1)}{\text{poly}(d)}\right)$, for every $t' \leq T + 1$, $(+, r)$ neuron index and $\boldsymbol{v}^*$-dominated patch $\boldsymbol{x}_{n,p}^{(\tau)}$,

$$(+, r) \notin S_+^{(0)}(\boldsymbol{v}^*) \implies \sigma\left(\langle \boldsymbol{w}_{+,r}^{(t')}, \boldsymbol{x}_{n,p}^{(\tau)} \rangle + b_{+,r}^{(t')}\right) = 0. \tag{144}$$

Conditioning on the (high-probability) event of the induction hypothesis of point 1., the following is already true on all the $\boldsymbol{v}^*$-dominated patches at time $t' \leq T$:

$$(+, r) \notin S_+^{(0)}(\boldsymbol{v}^*) \implies \sigma\left(\langle \boldsymbol{w}_{+,r}^{(t')}, \boldsymbol{x}_{n,p}^{(T)} \rangle + b_{+,r}^{(t')}\right) = 0. \tag{145}$$

In particular, $\sigma\left(\langle \boldsymbol{w}_{+,r}^{(T)}, \boldsymbol{x}_{n,p}^{(T)} \rangle + b_{+,r}^{(T)}\right) = 0$.

In other words, no $(+, r) \notin S_+^{(0)}(\boldsymbol{v}^*)$ can be updated on the $\boldsymbol{v}^*$-dominated patches at time $t = T$. Furthermore, the induction hypothesis of point 2. also states that the network cannot activate on any noise patch $\boldsymbol{x}_{n,p}^{(T)} = \boldsymbol{\zeta}_{n,p}^{(T)}$ with probability at least $1 - O\left(\frac{mNPT}{\text{poly}(d)}\right)$. Therefore, the neuron update for those $(+, r) \notin S_+^{(0)}(\boldsymbol{v}^*)$ takes the form

$$
\begin{aligned}
\Delta \boldsymbol{w}_{+,r}^{(T)} = &\frac{\eta}{NP} \sum_{\boldsymbol{v} \in \mathcal{C}(\boldsymbol{v}^*)} \sum_{n=1}^{N} \mathbb{1}\{|\mathcal{P}(\boldsymbol{X}_n^{(T)}; \boldsymbol{v})| > 0\} [\mathbb{1}\{y_n = +\} - \text{logit}_+^{(T)}(\boldsymbol{X}_n^{(T)})] \\
&\times \sum_{p \in \mathcal{P}(\boldsymbol{X}_n^{(T)}; \boldsymbol{v})} \mathbb{1}\{\langle \boldsymbol{w}_{+,r}^{(T)}, \alpha_{n,p}^{(T)} \boldsymbol{v} + \boldsymbol{\zeta}_{n,p}^{(T)} \rangle + b_{c,r}^{(T)} > 0\} \left(\alpha_{n,p}^{(T)} \boldsymbol{v} + \boldsymbol{\zeta}_{n,p}^{(T)}\right)
\end{aligned}
\tag{146}
$$

Now we can invoke Lemma F.2 and obtain that, with probability at least $1 - O\left(\frac{mNP}{\text{poly}(d)}\right)$, the following holds for all relevant neurons and $\boldsymbol{v}^*$-dominated patches:

$$\langle \Delta \boldsymbol{w}_{+,r}^{(T)}, \boldsymbol{x}_{n,p}^{(\tau)} \rangle + \Delta b_{+,r}^{(T)} < 0. \tag{147}$$

In conclusion, with $\tau \geq T + 1$, with probability at least $1 - O\left(\frac{mNP}{\text{poly}(d)}\right)$, for every $(+, r) \notin S_+^{(0)}(\boldsymbol{v}^*)$ and relevant $(n, p)$'s,

$$\langle \boldsymbol{w}_{+,r}^{(T)} + \Delta \boldsymbol{w}_{+,r}^{(T)}, \boldsymbol{x}_{n,p}^{(\tau)} \rangle + b_{+,r}^{(T)} + \Delta b_{+,r}^{(T)} = \langle \boldsymbol{w}_{+,r}^{(T+1)}, \boldsymbol{x}_{n,p}^{(\tau)} \rangle + b_{+,r}^{(T+1)} < 0, \tag{148}$$

which leads to $\langle \boldsymbol{w}_{+,r}^{(t')}, \boldsymbol{x}_{n,p}^{(\tau)}\rangle + b_{+,r}^{(t')} < 0$ for all $t' \leq T+1$ with probability at least $1 - O\left(\frac{mk_+ NP(T+1)}{\text{poly}(d)}\right)$ (also taking union bound over all the possible choices of $\boldsymbol{v}^*$). This finishes the inductive step for point 1.

*2. (Non-activation on noise patches)*

Relying on the event of the induction hypothesis, for any $\tau \geq T$, the following holds for every $r \in [m]$ and noise patch $\boldsymbol{x}_{n,p}^{(\tau)} = \boldsymbol{\zeta}_{n,p}^{(\tau)}$,

$$\langle \boldsymbol{w}_{+,r}^{(T)}, \boldsymbol{x}_{n,p}^{(\tau)}\rangle + b_{+,r}^{(T)} < 0. \tag{149}$$

Conditioning on this high-probability event, this means no neuron $\boldsymbol{w}_{+,r}^{(T)}$ can be updated on the noise patches. Denoting the set of features $\mathcal{M} = \{\boldsymbol{v}_{+,c}\}_{c=1}^{k_+} \cup \{\boldsymbol{v}_{-,c}\}_{c=1}^{k_-} \cup \{\boldsymbol{v}_+, \boldsymbol{v}_-\}$, for every $r \in [m]$, its update is reduced to

$$\begin{aligned}
\Delta \boldsymbol{w}_{+,r}^{(T)} =& \frac{\eta}{NP} \sum_{\boldsymbol{v} \in \mathcal{M}} \sum_{n=1}^{N} \mathbb{1}\{|\mathcal{P}(\boldsymbol{X}_n^{(T)}; \boldsymbol{v})| > 0\}[\mathbb{1}\{y_n = +\} - \text{logit}_+^{(T)}(\boldsymbol{X}_n^{(T)})] \\
&\times \sum_{p \in \mathcal{P}(\boldsymbol{X}_n^{(T)}; \boldsymbol{v})} \mathbb{1}\{\langle \boldsymbol{w}_{+,r}^{(T)}, \alpha_{n,p}^{(T)}\boldsymbol{v} + \boldsymbol{\zeta}_{n,p}^{(T)}\rangle + b_{c,r}^{(T)} > 0\}\left(\alpha_{n,p}^{(T)}\boldsymbol{v} + \boldsymbol{\zeta}_{n,p}^{(T)}\right),
\end{aligned} \tag{150}$$

Invoking Lemma F.3, we have that, for any $\tau \geq T+1$, the following inequality holds with probability at least $1 - O\left(\frac{mNP}{\text{poly}(d)}\right)$ for every $r \in [m]$ and noise patches,

$$\langle \Delta \boldsymbol{w}_{+,r}^{(T)}, \boldsymbol{x}_{n,p}^{(\tau)}\rangle + \Delta b_{+,r}^{(T)} < 0. \tag{151}$$

Consequently, for any $\tau \geq T+1$, the following inequality holds with probability at least $1 - O\left(\frac{mNP}{\text{poly}(d)}\right)$ for every $r \in [m]$ and noise patches $\boldsymbol{x}_{n,p}^{(\tau)} = \boldsymbol{\zeta}_{n,p}^{(\tau)}$:

$$\langle \boldsymbol{w}_{+,r}^{(T)} + \Delta \boldsymbol{w}_{+,r}^{(T)}, \boldsymbol{x}_{n,p}^{(\tau)}\rangle + b_{+,r}^{(T)} + \Delta b_{+,r}^{(T)} = \langle \boldsymbol{w}_{+,r}^{(T+1)}, \boldsymbol{x}_{n,p}^{(\tau)}\rangle + b_{+,r}^{(T+1)} < 0. \tag{152}$$

This finishes the inductive step for point 2.

*3. (Off-diagonal nonpositive growth)* Choose any $\boldsymbol{v}^* \in \{\boldsymbol{v}_-\} \cup \{\boldsymbol{v}_{-,c}\}_{c=1}^{k_-}$.

Choose any neuron with index $(+,r)$. Similar to our proof for point 2., we know that its update, when taken inner product with a $\boldsymbol{v}^*$-dominated patch $\boldsymbol{x}_{n,p}^{(\tau)} = \sqrt{1 \pm \iota}\boldsymbol{v}^* + \boldsymbol{\zeta}_{n,p}^{(\tau)}$, has to take the form

$$\begin{aligned}
&\langle \Delta \boldsymbol{w}_{+,r}^{(T)}, \sqrt{1 \pm \iota}\boldsymbol{v}^* + \boldsymbol{\zeta}_{n,p}^{(\tau)}\rangle \\
=&\frac{\eta}{NP} \sum_{\boldsymbol{v} \in \mathcal{M}} \sum_{n=1}^{N} \mathbb{1}\{|\mathcal{P}(\boldsymbol{X}_n^{(T)}; \boldsymbol{v})| > 0\}[\mathbb{1}\{y_n = +\} - \text{logit}_+^{(T)}(\boldsymbol{X}_n^{(T)})] \\
&\times \sum_{p \in \mathcal{P}(\boldsymbol{X}_n^{(T)}; \boldsymbol{v})} \mathbb{1}\{\langle \boldsymbol{w}_{+,r}^{(T)}, \alpha_{n,p}^{(T)}\boldsymbol{v} + \boldsymbol{\zeta}_{n,p}^{(T)}\rangle + b_{+,r}^{(T)} > 0\}\langle \alpha_{n,p}^{(T)}\boldsymbol{v} + \boldsymbol{\zeta}_{n,p}^{(T)}, \sqrt{1 \pm \iota}\boldsymbol{v}^* + \boldsymbol{\zeta}_{n,p}^{(\tau)}\rangle \\
=&\frac{\eta}{NP} \sum_{\boldsymbol{v} \in \mathcal{M}-\{\boldsymbol{v}^*\}} \sum_{n=1}^{N} \mathbb{1}\{|\mathcal{P}(\boldsymbol{X}_n^{(T)}; \boldsymbol{v})| > 0\}[\mathbb{1}\{y_n = +\} - \text{logit}_+^{(T)}(\boldsymbol{X}_n^{(T)})] \\
&\times \sum_{p \in \mathcal{P}(\boldsymbol{X}_n^{(T)}; \boldsymbol{v})} \mathbb{1}\{\langle \boldsymbol{w}_{+,r}^{(T)}, \alpha_{n,p}^{(T)}\boldsymbol{v} + \boldsymbol{\zeta}_{n,p}^{(T)}\rangle + b_{+,r}^{(T)} > 0\}\left(\langle \boldsymbol{\zeta}_{n,p}^{(T)}, \sqrt{1 \pm \iota}\boldsymbol{v}^*\rangle + \langle \alpha_{n,p}^{(T)}\boldsymbol{v} + \boldsymbol{\zeta}_{n,p}^{(T)}, \boldsymbol{\zeta}_{n,p}^{(\tau)}\rangle\right) \\
&- \frac{\eta}{NP} \sum_{n=1}^{N} \mathbb{1}\{|\mathcal{P}(\boldsymbol{X}_n^{(T)}; \boldsymbol{v}^*)| > 0\}[\text{logit}_+^{(T)}(\boldsymbol{X}_n^{(T)})] \\
&\times \sum_{p \in \mathcal{P}(\boldsymbol{X}_n^{(T)}; \boldsymbol{v})} \mathbb{1}\{\langle \boldsymbol{w}_{+,r}^{(T)}, \alpha_{n,p}^{(T)}\boldsymbol{v} + \boldsymbol{\zeta}_{n,p}^{(T)}\rangle + b_{+,r}^{(T)} > 0\}\langle \alpha_{n,p}^{(T)}\boldsymbol{v}^* + \boldsymbol{\zeta}_{n,p}^{(T)}, \sqrt{1 \pm \iota}\boldsymbol{v}^* + \boldsymbol{\zeta}_{n,p}^{(\tau)}\rangle
\end{aligned} \tag{153}$$

With probability at least $1 - O\left(\frac{NP}{\text{poly}(d)}\right)$, $\langle \alpha_{n,p}^{(T)} \boldsymbol{v}^* + \boldsymbol{\zeta}_{n,p}^{(T)}, \sqrt{1 \pm \iota} \boldsymbol{v}^* + \boldsymbol{\zeta}_{n,p}^{(\tau)} \rangle > 0$, and $\langle \boldsymbol{\zeta}_{n,p}^{(T)}, \sqrt{1 \pm \iota} \boldsymbol{v}^* \rangle + \langle \alpha_{n,p}^{(T)} \boldsymbol{v} + \boldsymbol{\zeta}_{n,p}^{(T)}, \boldsymbol{\zeta}_{n,p}^{(\tau)} \rangle < O(1/\log^9(d))$. Therefore,

$$
\begin{aligned}
\langle \Delta \boldsymbol{w}_{+,r}^{(T)}, \boldsymbol{v}^* \rangle < &\frac{\eta}{NP} \sum_{\boldsymbol{v} \in \mathcal{M} - \{\boldsymbol{v}^*\}} \sum_{n=1}^{N} \mathbb{1}\{|\mathcal{P}(\boldsymbol{X}_n^{(T)}; \boldsymbol{v})| > 0\}[\mathbb{1}\{y_n = +\} - \text{logit}_+^{(T)}(\boldsymbol{X}_n^{(T)})] \\
&\times \sum_{p \in \mathcal{P}(\boldsymbol{X}_n^{(T)}; \boldsymbol{v})} \mathbb{1}\{\langle \boldsymbol{w}_{+,r}^{(T)}, \alpha_{n,p}^{(T)} \boldsymbol{v} + \boldsymbol{\zeta}_{n,p}^{(T)} \rangle + b_{+,r}^{(T)} > 0\} O\left(\frac{1}{\log^9(d)}\right)
\end{aligned}
\tag{154}
$$

Invoking Lemma F.3, we know that

$$
\begin{aligned}
\Delta b_{+,r}^{(T)} \\
\leq &-\frac{1}{\log^5(d)} \frac{\eta}{NP} \left(\sqrt{1 - \iota} - \frac{1}{\log^9(d)}\right) \\
&\times \left(\sum_{\boldsymbol{v} \in \mathcal{M}} \sum_{n=1}^{N} \mathbb{1}\{|\mathcal{P}(\boldsymbol{X}_n^{(T)}; \boldsymbol{v})| > 0\} \left| \mathbb{1}\{y_n = +\} - \text{logit}_+^{(T)}(\boldsymbol{X}_n^{(T)}) \right| \right. \\
&\left. \times \sum_{p \in \mathcal{P}(\boldsymbol{X}_n^{(T)}; \boldsymbol{v})} \mathbb{1}\{\langle \boldsymbol{w}_{+,r}^{(T)}, \alpha_{n,p}^{(T)} \boldsymbol{v} + \boldsymbol{\zeta}_{n,p}^{(T)} \rangle + b_{+,r}^{(T)} > 0\} \right).
\end{aligned}
\tag{155}
$$

It follows that

$$
\begin{aligned}
&\langle \Delta \boldsymbol{w}_{+,r}^{(T)}, \sqrt{1 \pm \iota} \boldsymbol{v}^* + \boldsymbol{\zeta}_{n,p}^{(\tau)} \rangle + \Delta b_{+,r}^{(T)} \\
<& O\left(\frac{1}{\log^9(d)}\right) \frac{\eta}{NP} \left(\sum_{\boldsymbol{v} \in \mathcal{M} - \{\boldsymbol{v}^*\}} \sum_{n=1}^{N} \mathbb{1}\{|\mathcal{P}(\boldsymbol{X}_n^{(T)}; \boldsymbol{v})| > 0\}[\mathbb{1}\{y_n = +\} - \text{logit}_+^{(T)}(\boldsymbol{X}_n^{(T)})] \right. \\
&\times \left. \sum_{p \in \mathcal{P}(\boldsymbol{X}_n^{(T)}; \boldsymbol{v})} \mathbb{1}\{\langle \boldsymbol{w}_{+,r}^{(T)}, \alpha_{n,p}^{(T)} \boldsymbol{v} + \boldsymbol{\zeta}_{n,p}^{(T)} \rangle + b_{c,r}^{(T)} > 0\} \right) \\
&- \Omega\left(\frac{1}{\log^5(d)}\right) \frac{\eta}{NP} \left(\sum_{\boldsymbol{v} \in \mathcal{M}} \sum_{n=1}^{N} \mathbb{1}\{|\mathcal{P}(\boldsymbol{X}_n^{(T)}; \boldsymbol{v})| > 0\} \left| \mathbb{1}\{y_n = +\} - \text{logit}_+^{(T)}(\boldsymbol{X}_n^{(T)}) \right| \right. \\
&\times \left. \sum_{p \in \mathcal{P}(\boldsymbol{X}_n^{(T)}; \boldsymbol{v})} \mathbb{1}\{\langle \boldsymbol{w}_{+,r}^{(T)}, \alpha_{n,p}^{(T)} \boldsymbol{v} + \boldsymbol{\zeta}_{n,p}^{(T)} \rangle + b_{c,r}^{(T)} > 0\} \right) \\
<& 0.
\end{aligned}
\tag{156}
$$

Consequently,

$$
\begin{aligned}
&\sigma\left(\langle \boldsymbol{w}_{+,r}^{(T+1)}, \sqrt{1 \pm \iota} \boldsymbol{v}^* + \boldsymbol{\zeta}_{n,p}^{(\tau)} \rangle + b_{+,r}^{(T+1)}\right) \\
=& \sigma\left(\langle \boldsymbol{w}_{+,r}^{(T)}, \sqrt{1 \pm \iota} \boldsymbol{v}^* + \boldsymbol{\zeta}_{n,p}^{(\tau)} \rangle + b_{+,r}^{(T)} + \langle \Delta \boldsymbol{w}_{+,r}^{(T)}, \sqrt{1 \pm \iota} \boldsymbol{v}^* + \boldsymbol{\zeta}_{n,p}^{(\tau)} \rangle + \Delta b_{+,r}^{(T)}\right) \\
\leq& \sigma\left(\langle \boldsymbol{w}_{+,r}^{(T)}, \sqrt{1 \pm \iota} \boldsymbol{v}^* + \boldsymbol{\zeta}_{n,p}^{(\tau)} \rangle + b_{+,r}^{(T)}\right) \\
\leq& \sigma\left(\langle \boldsymbol{w}_{+,r}^{(0)}, \sqrt{1 \pm \iota} \boldsymbol{v}^* + \boldsymbol{\zeta}_{n,p}^{(\tau)} \rangle + b_{+,r}^{(0)}\right).
\end{aligned}
\tag{157}
$$

$\square$

**Corollary F.1.1** (Bias update upper bound). *Choose any $T_e \leq poly(d)$. With probability at least $1 - O\left(\frac{mk_+NPT_e}{poly(d)}\right)$, for all $t \in [0, T_e]$, any neuron $\boldsymbol{w}_{+,r}$, and any $\boldsymbol{v} \in \mathcal{U}_{+,r}^{(0)}$,*

$$
\Delta b_{+,r}^{(t)} < -\Omega\left(\frac{polylog(d)}{\log^5(d)}\right) \left|\langle \Delta \boldsymbol{w}_{+,r}^{(t)}, \boldsymbol{\zeta}^* \rangle\right|.
\tag{158}
$$

*Proof.* Conditioning on the high-probability events of Theorem F.1 above, we know that for any neuron indexed $(+, r)$, at any time $t \le T_e$, its update takes the form

$$
\begin{aligned}
\Delta \boldsymbol{w}_{+,r}^{(t)} = &\frac{\eta}{NP} \sum_{\boldsymbol{v} \in \mathcal{U}_{+,r}^{(0)}} \sum_{n=1}^{N} \mathbb{1}\{|\mathcal{P}(\boldsymbol{X}_n^{(t)}; \boldsymbol{v})| > 0\}[\mathbb{1}\{y_n = +\} - \mathrm{logit}_+^{(t)}(\boldsymbol{X}_n^{(t)})] \\
&\times \sum_{p \in \mathcal{P}(\boldsymbol{X}_n^{(t)}; \boldsymbol{v})} \mathbb{1}\{\langle \boldsymbol{w}_{+,r}^{(t)}, \alpha_{n,p}^{(t)} \boldsymbol{v} + \boldsymbol{\zeta}_{n,p}^{(t)}\rangle + b_{c,r}^{(t)} > 0\} \left(\alpha_{n,p}^{(t)} \boldsymbol{v} + \boldsymbol{\zeta}_{n,p}^{(t)}\right),
\end{aligned}
\tag{159}
$$

It follows that, with probability at least $1 - O\left(\frac{1}{\mathrm{poly}(d)}\right)$,

$$
\begin{aligned}
\left|\langle \Delta \boldsymbol{w}_{+,r}^{(t)}, \boldsymbol{\zeta}^*\rangle\right| = &\left|\frac{\eta}{NP} \sum_{\boldsymbol{v} \in \mathcal{U}_{+,r}^{(0)}} \sum_{n=1}^{N} \mathbb{1}\{|\mathcal{P}(\boldsymbol{X}_n^{(t)}; \boldsymbol{v})| > 0\}[\mathbb{1}\{y_n = +\} - \mathrm{logit}_+^{(t)}(\boldsymbol{X}_n^{(t)})]\right. \\
&\left.\times \sum_{p \in \mathcal{P}(\boldsymbol{X}_n^{(t)}; \boldsymbol{v})} \mathbb{1}\{\langle \boldsymbol{w}_{+,r}^{(t)}, \alpha_{n,p}^{(t)} \boldsymbol{v} + \boldsymbol{\zeta}_{n,p}^{(t)}\rangle + b_{c,r}^{(t)} > 0\}\langle \alpha_{n,p}^{(t)} \boldsymbol{v} + \boldsymbol{\zeta}_{n,p}^{(t)}, \boldsymbol{\zeta}^*\rangle\right| \\
\le &\frac{\eta}{NP} \sum_{\boldsymbol{v} \in \mathcal{U}_{+,r}^{(0)}} \sum_{n=1}^{N} \mathbb{1}\{|\mathcal{P}(\boldsymbol{X}_n^{(t)}; \boldsymbol{v})| > 0\} \left|\mathbb{1}\{y_n = +\} - \mathrm{logit}_+^{(t)}(\boldsymbol{X}_n^{(t)})\right| \\
&\times \sum_{p \in \mathcal{P}(\boldsymbol{X}_n^{(t)}; \boldsymbol{v})} \mathbb{1}\{\langle \boldsymbol{w}_{+,r}^{(t)}, \alpha_{n,p}^{(t)} \boldsymbol{v} + \boldsymbol{\zeta}_{n,p}^{(t)}\rangle + b_{c,r}^{(t)} > 0\} O\left(\frac{1}{\mathrm{polylog}(d)}\right)
\end{aligned}
\tag{160}
$$

On the other hand,

$$
\begin{aligned}
\left\|\Delta \boldsymbol{w}_{+,r}^{(t)}\right\|_2 \ge &\left\|\frac{\eta}{NP} \sum_{\boldsymbol{v} \in \mathcal{U}_{+,r}^{(0)}} \sum_{n=1}^{N} \mathbb{1}\{|\mathcal{P}(\boldsymbol{X}_n^{(t)}; \boldsymbol{v})| > 0\}[\mathbb{1}\{y_n = +\} - \mathrm{logit}_+^{(t)}(\boldsymbol{X}_n^{(t)})]\right. \\
&\left.\times \sum_{p \in \mathcal{P}(\boldsymbol{X}_n^{(t)}; \boldsymbol{v})} \mathbb{1}\{\langle \boldsymbol{w}_{+,r}^{(t)}, \alpha_{n,p}^{(t)} \boldsymbol{v} + \boldsymbol{\zeta}_{n,p}^{(t)}\rangle + b_{c,r}^{(t)} > 0\} \alpha_{n,p}^{(t)} \boldsymbol{v}\right\|_2 \\
&- \left\|\frac{\eta}{NP} \sum_{\boldsymbol{v} \in \mathcal{U}_{+,r}^{(0)}} \sum_{n=1}^{N} \mathbb{1}\{|\mathcal{P}(\boldsymbol{X}_n^{(t)}; \boldsymbol{v})| > 0\}[\mathbb{1}\{y_n = +\} - \mathrm{logit}_+^{(t)}(\boldsymbol{X}_n^{(t)})]\right. \\
&\left.\times \sum_{p \in \mathcal{P}(\boldsymbol{X}_n^{(t)}; \boldsymbol{v})} \mathbb{1}\{\langle \boldsymbol{w}_{+,r}^{(t)}, \alpha_{n,p}^{(t)} \boldsymbol{v} + \boldsymbol{\zeta}_{n,p}^{(t)}\rangle + b_{c,r}^{(t)} > 0\} \boldsymbol{\zeta}_{n,p}^{(t)}\right\|_2 \\
\ge &\frac{\eta}{NP} \sum_{\boldsymbol{v} \in \mathcal{U}_{+,r}^{(0)}} \sum_{n=1}^{N} \mathbb{1}\{|\mathcal{P}(\boldsymbol{X}_n^{(t)}; \boldsymbol{v})| > 0\} \left|\mathbb{1}\{y_n = +\} - \mathrm{logit}_+^{(t)}(\boldsymbol{X}_n^{(t)})\right| \\
&\times \sum_{p \in \mathcal{P}(\boldsymbol{X}_n^{(t)}; \boldsymbol{v})} \mathbb{1}\{\langle \boldsymbol{w}_{+,r}^{(t)}, \alpha_{n,p}^{(t)} \boldsymbol{v} + \boldsymbol{\zeta}_{n,p}^{(t)}\rangle + b_{c,r}^{(t)} > 0\} \left(\sqrt{1 - \iota} - O\left(\frac{1}{\log^9(d)}\right)\right)
\end{aligned}
\tag{161}
$$

Clearly,

$$
\left\|\Delta \boldsymbol{w}_{+,r}^{(t)}\right\|_2 \ge \Omega\left(\mathrm{polylog}(d) \left|\langle \Delta \boldsymbol{w}_{+,r}^{(t)}, \boldsymbol{\zeta}^*\rangle\right|\right).
\tag{162}
$$

The conclusion follows. □

**Lemma F.2** (Nonactivation invariance)**.** *Let the assumptions in Theorem D.1 hold.*

*Denote the set of features* $\mathcal{C}(\boldsymbol{v}^*) = \{\boldsymbol{v}_{+,c}\}_{c=1}^{k_+} \cup \{\boldsymbol{v}_{-,c}\}_{c=1}^{k_-} \cup \{\boldsymbol{v}_+, \boldsymbol{v}_-\} - \{\boldsymbol{v}^*\}$. *If the update term for neuron* $\boldsymbol{w}_{+,r}^{(t)}$ *can be written as follows*

$$
\begin{aligned}
\Delta\boldsymbol{w}_{+,r}^{(t)} = \frac{\eta}{NP} \sum_{\boldsymbol{v}\in\mathcal{C}(\boldsymbol{v}^*)} \sum_{n=1}^{N} &\mathbb{1}\{|\mathcal{P}(\boldsymbol{X}_n^{(t)};\boldsymbol{v})| > 0\}[\mathbb{1}\{y_n = +\} - logit_+^{(t)}(\boldsymbol{X}_n^{(t)})] \\
&\times \sum_{p\in\mathcal{P}(\boldsymbol{X}_n^{(t)};\boldsymbol{v})} \mathbb{1}\{\langle\boldsymbol{w}_{+,r}^{(t)}, \alpha_{n,p}^{(t)}\boldsymbol{v} + \boldsymbol{\zeta}_{n,p}^{(t)}\rangle + b_{c,r}^{(t)} > 0\}\left(\alpha_{n,p}^{(t)}\boldsymbol{v} + \boldsymbol{\zeta}_{n,p}^{(t)}\right),
\end{aligned}
\tag{163}
$$

*then given any* $\tau > t$, *the following inequality holds with probability at least* $1 - O\left(\frac{NP}{poly(d)}\right)$ *for all* $\boldsymbol{v}^*$*-dominated patch* $\boldsymbol{x}_{n,p}^{(\tau)}$:

$$
\langle\Delta\boldsymbol{w}_{+,r}^{(t)}, \boldsymbol{x}_{n,p}^{(\tau)}\rangle + \Delta b_{+,r}^{(t)} < 0
\tag{164}
$$

*Proof.* Let us fix a neuron $\boldsymbol{w}_{+,r}$ satisfying the update expression in the Lemma statement, and fix some $\tau > t$. Firstly, the bias update for this neuron can be upper bounded via the reverse triangle inequality:

$$
\begin{aligned}
\Delta b_{+,r}^{(t)} = &-\frac{\left\|\Delta\boldsymbol{w}_{+,r}^{(t)}\right\|_2}{\log^5(d)} \\
\leq &-\frac{1}{\log^5(d)}\frac{\eta}{NP}\left\|\sum_{\boldsymbol{v}\in\mathcal{C}(\boldsymbol{v}^*)}\sum_{n=1}^{N}\mathbb{1}\{|\mathcal{P}(\boldsymbol{X}_n^{(t)};\boldsymbol{v})| > 0\}[\mathbb{1}\{y_n = +\} - logit_+^{(t)}(\boldsymbol{X}_n^{(t)})]\right. \\
&\times \left.\sum_{p\in\mathcal{P}(\boldsymbol{X}_n^{(t)};\boldsymbol{v})}\mathbb{1}\{\langle\boldsymbol{w}_{+,r}^{(t)}, \alpha_{n,p}^{(t)}\boldsymbol{v} + \boldsymbol{\zeta}_{n,p}^{(t)}\rangle + b_{c,r}^{(t)} > 0\}\alpha_{n,p}^{(t)}\boldsymbol{v}\right\|_2 \\
&+\frac{1}{\log^5(d)}\frac{\eta}{NP}\left\|\sum_{\boldsymbol{v}\in\mathcal{C}(\boldsymbol{v}^*)}\sum_{n=1}^{N}\mathbb{1}\{|\mathcal{P}(\boldsymbol{X}_n^{(t)};\boldsymbol{v})| > 0\}[\mathbb{1}\{y_n = +\} - logit_+^{(t)}(\boldsymbol{X}_n^{(t)})]\right. \\
&\times \left.\sum_{p\in\mathcal{P}(\boldsymbol{X}_n^{(t)};\boldsymbol{v})}\mathbb{1}\{\langle\boldsymbol{w}_{+,r}^{(t)}, \alpha_{n,p}^{(t)}\boldsymbol{v} + \boldsymbol{\zeta}_{n,p}^{(t)}\rangle + b_{c,r}^{(t)} > 0\}\boldsymbol{\zeta}_{n,p}^{(t)}\right\|_2
\end{aligned}
\tag{165}
$$

Let us further upper bound the two $\|\cdot\|_2$ terms separately. Firstly,

$$
\begin{aligned}
& \left\| \sum_{\boldsymbol{v}\in\mathcal{C}(\boldsymbol{v}^*)} \sum_{n=1}^{N} \mathbb{1}\{|\mathcal{P}(\boldsymbol{X}_n^{(t)};\boldsymbol{v})| > 0\}[\mathbb{1}\{y_n = +\} - \text{logit}_+^{(t)}(\boldsymbol{X}_n^{(t)})] \right. \\
& \quad \times \left. \sum_{p\in\mathcal{P}(\boldsymbol{X}_n^{(t)};\boldsymbol{v})} \mathbb{1}\{\langle \boldsymbol{w}_{+,r}^{(t)}, \alpha_{n,p}^{(t)}\boldsymbol{v} + \boldsymbol{\zeta}_{n,p}^{(t)}\rangle + b_{c,r}^{(t)} > 0\}\alpha_{n,p}^{(t)}\boldsymbol{v} \right\|_2 \\
& = \sum_{\boldsymbol{v}\in\mathcal{C}(\boldsymbol{v}^*)} \left\| \sum_{n=1}^{N} \mathbb{1}\{|\mathcal{P}(\boldsymbol{X}_n^{(t)};\boldsymbol{v})| > 0\}[\mathbb{1}\{y_n = +\} - \text{logit}_+^{(t)}(\boldsymbol{X}_n^{(t)})] \right. \\
& \quad \times \left. \sum_{p\in\mathcal{P}(\boldsymbol{X}_n^{(t)};\boldsymbol{v})} \mathbb{1}\{\langle \boldsymbol{w}_{+,r}^{(t)}, \alpha_{n,p}^{(t)}\boldsymbol{v} + \boldsymbol{\zeta}_{n,p}^{(t)}\rangle + b_{c,r}^{(t)} > 0\}\alpha_{n,p}^{(t)}\boldsymbol{v} \right\|_2 \\
& = \sum_{\boldsymbol{v}\in\mathcal{C}(\boldsymbol{v}^*)} \sum_{n=1}^{N} \mathbb{1}\{|\mathcal{P}(\boldsymbol{X}_n^{(t)};\boldsymbol{v})| > 0\}\left| \mathbb{1}\{y_n = +\} - \text{logit}_+^{(t)}(\boldsymbol{X}_n^{(t)})\right| \\
& \quad \times \sum_{p\in\mathcal{P}(\boldsymbol{X}_n^{(t)};\boldsymbol{v})} \mathbb{1}\{\langle \boldsymbol{w}_{+,r}^{(t)}, \alpha_{n,p}^{(t)}\boldsymbol{v} + \boldsymbol{\zeta}_{n,p}^{(t)}\rangle + b_{c,r}^{(t)} > 0\}\alpha_{n,p}^{(t)}\|\boldsymbol{v}\|_2 \\
& \geq \sum_{\boldsymbol{v}\in\mathcal{C}(\boldsymbol{v}^*)} \sum_{n=1}^{N} \mathbb{1}\{|\mathcal{P}(\boldsymbol{X}_n^{(t)};\boldsymbol{v})| > 0\}\left| \mathbb{1}\{y_n = +\} - \text{logit}_+^{(t)}(\boldsymbol{X}_n^{(t)})\right| \\
& \quad \times \sum_{p\in\mathcal{P}(\boldsymbol{X}_n^{(t)};\boldsymbol{v})} \mathbb{1}\{\langle \boldsymbol{w}_{+,r}^{(t)}, \alpha_{n,p}^{(t)}\boldsymbol{v} + \boldsymbol{\zeta}_{n,p}^{(t)}\rangle + b_{c,r}^{(t)} > 0\}\sqrt{1 - \iota}
\end{aligned}
\tag{166}
$$

Secondly, with probability at least $1 - O\left(\frac{NP}{\text{poly}(d)}\right)$,

$$
\begin{aligned}
& \left\| \sum_{\boldsymbol{v}\in\mathcal{C}(\boldsymbol{v}^*)} \sum_{n=1}^{N} \mathbb{1}\{|\mathcal{P}(\boldsymbol{X}_n^{(t)};\boldsymbol{v})| > 0\}[\mathbb{1}\{y_n = +\} - \text{logit}_+^{(t)}(\boldsymbol{X}_n^{(t)})] \right. \\
& \quad \times \left. \sum_{p\in\mathcal{P}(\boldsymbol{X}_n^{(t)};\boldsymbol{v})} \mathbb{1}\{\langle \boldsymbol{w}_{+,r}^{(t)}, \alpha_{n,p}^{(t)}\boldsymbol{v} + \boldsymbol{\zeta}_{n,p}^{(t)}\rangle + b_{c,r}^{(t)} > 0\}\boldsymbol{\zeta}_{n,p}^{(t)} \right\|_2 \\
& \leq \sum_{\boldsymbol{v}\in\mathcal{C}(\boldsymbol{v}^*)} \sum_{n=1}^{N} \mathbb{1}\{|\mathcal{P}(\boldsymbol{X}_n^{(t)};\boldsymbol{v})| > 0\}\left| \mathbb{1}\{y_n = +\} - \text{logit}_+^{(t)}(\boldsymbol{X}_n^{(t)})\right| \\
& \quad \times \sum_{p\in\mathcal{P}(\boldsymbol{X}_n^{(t)};\boldsymbol{v})} \mathbb{1}\{\langle \boldsymbol{w}_{+,r}^{(t)}, \alpha_{n,p}^{(t)}\boldsymbol{v} + \boldsymbol{\zeta}_{n,p}^{(t)}\rangle + b_{c,r}^{(t)} > 0\}\left\|\boldsymbol{\zeta}_{n,p}^{(t)}\right\|_2 \\
& \leq \sum_{\boldsymbol{v}\in\mathcal{C}(\boldsymbol{v}^*)} \sum_{n=1}^{N} \mathbb{1}\{|\mathcal{P}(\boldsymbol{X}_n^{(t)};\boldsymbol{v})| > 0\}\left| \mathbb{1}\{y_n = +\} - \text{logit}_+^{(t)}(\boldsymbol{X}_n^{(t)})\right| \\
& \quad \times \sum_{p\in\mathcal{P}(\boldsymbol{X}_n^{(0)};\boldsymbol{v})} \mathbb{1}\{\langle \boldsymbol{w}_{+,r}^{(t)}, \alpha_{n,p}^{(t)}\boldsymbol{v} + \boldsymbol{\zeta}_{n,p}^{(t)}\rangle + b_{c,r}^{(t)} > 0\}\frac{1}{\log^9(d)}
\end{aligned}
\tag{167}
$$

Therefore, with probability at least $1 - O\left(\frac{NP}{\text{poly}(d)}\right)$, we can bound the update to the bias as follows:

$$
\begin{aligned}
\Delta b_{+,r}^{(t)} \\
\leq -\frac{1}{\log^5(d)} \frac{\eta}{NP} \left(\sqrt{1-\iota} - \frac{1}{\log^9(d)}\right) \\
\times \left(\sum_{\boldsymbol{v} \in \mathcal{C}(\boldsymbol{v}^*)} \sum_{n=1}^{N} \mathbb{1}\{|\mathcal{P}(\boldsymbol{X}_n^{(t)}; \boldsymbol{v})| > 0\} \left|\mathbb{1}\{y_n = +\} - \text{logit}_+^{(t)}(\boldsymbol{X}_n^{(t)})\right| \right. \\
\left. \times \sum_{p \in \mathcal{P}(\boldsymbol{X}_n^{(t)}; \boldsymbol{v})} \mathbb{1}\{\langle \boldsymbol{w}_{+,r}^{(t)}, \alpha_{n,p}^{(t)} \boldsymbol{v} + \boldsymbol{\zeta}_{n,p}^{(t)}\rangle + b_{c,r}^{(t)} > 0\}\right)
\end{aligned}
\tag{168}
$$

Furthermore, with probability at least $1 - e^{-\Omega(d)+O(\log(d))} > 1 - O\left(\frac{NP}{\text{poly}(d)}\right)$, the following holds for all $n, p$:

$$
\langle \alpha_{n,p}^{(t)} \boldsymbol{v}, \boldsymbol{\zeta}_{n,p}^{(\tau)}\rangle, \ \langle \boldsymbol{\zeta}_{n,p}^{(t)}, \alpha_{n,p}^{(\tau)} \boldsymbol{v}^*\rangle, \ \langle \boldsymbol{\zeta}_{n,p}^{(t)}, \boldsymbol{\zeta}_{n,p}^{(\tau)}\rangle < O\left(\frac{1}{\log^9(d)}\right).
\tag{169}
$$

Combining the above derivations, they imply that with probability at least $1 - O\left(\frac{NP}{\text{poly}(d)}\right)$, for any $\boldsymbol{x}_{n,p}^{(\tau)}$ dominated by $\boldsymbol{v}^*$,

$$
\begin{aligned}
&\langle \Delta \boldsymbol{w}_{+,r}^{(t)}, \boldsymbol{x}_{n,p}^{(\tau)}\rangle + \Delta b_{+,r}^{(t)} \\
=& \langle \Delta \boldsymbol{w}_{+,r}^{(t)}, \alpha_{n,p}^{(\tau)} \boldsymbol{v}^* + \boldsymbol{\zeta}_{n,p}^{(\tau)}\rangle + \Delta b_{+,r}^{(t)} \\
=& \frac{\eta}{NP} \sum_{\boldsymbol{v} \in \mathcal{C}(\boldsymbol{v}^*)} \sum_{n=1}^{N} \mathbb{1}\{|\mathcal{P}(\boldsymbol{X}_n^{(t)}; \boldsymbol{v})| > 0\}[\mathbb{1}\{y_n = +\} - \text{logit}_+^{(t)}(\boldsymbol{X}_n^{(t)})] \\
&\times \sum_{p \in \mathcal{P}(\boldsymbol{X}_n^{(t)}; \boldsymbol{v})} \mathbb{1}\{\langle \boldsymbol{w}_{+,r}^{(t)}, \alpha_{n,p}^{(t)} \boldsymbol{v} + \boldsymbol{\zeta}_{n,p}^{(t)}\rangle + b_{c,r}^{(t)} > 0\}\langle \alpha_{n,p}^{(t)} \boldsymbol{v} + \boldsymbol{\zeta}_{n,p}^{(t)}, \alpha_{n,p}^{(\tau)} \boldsymbol{v}^* + \boldsymbol{\zeta}_{n,p}^{(\tau)}\rangle + \Delta b_{+,r}^{(t)} \\
=& \frac{\eta}{NP} \sum_{\boldsymbol{v} \in \mathcal{C}(\boldsymbol{v}^*)} \sum_{n=1}^{N} \mathbb{1}\{|\mathcal{P}(\boldsymbol{X}_n^{(t)}; \boldsymbol{v})| > 0\}[\mathbb{1}\{y_n = +\} - \text{logit}_+^{(t)}(\boldsymbol{X}_n^{(t)})] \\
&\times \sum_{p \in \mathcal{P}(\boldsymbol{X}_n^{(t)}; \boldsymbol{v})} \mathbb{1}\{\langle \boldsymbol{w}_{+,r}^{(t)}, \alpha_{n,p}^{(t)} \boldsymbol{v} + \boldsymbol{\zeta}_{n,p}^{(t)}\rangle + b_{c,r}^{(t)} > 0\} \left(\langle \alpha_{n,p}^{(t)} \boldsymbol{v}, \boldsymbol{\zeta}_{n,p}^{(\tau)}\rangle + \langle \boldsymbol{\zeta}_{n,p}^{(t)}, \alpha_{n,p}^{(\tau)} \boldsymbol{v}^*\rangle + \langle \boldsymbol{\zeta}_{n,p}^{(t)}, \boldsymbol{\zeta}_{n,p}^{(\tau)}\rangle\right) \\
&+ \Delta b_{+,r}^{(t)} \\
\leq& \frac{\eta}{NP} \sum_{\boldsymbol{v} \in \mathcal{C}(\boldsymbol{v}^*)} \sum_{n=1}^{N} \mathbb{1}\{|\mathcal{P}(\boldsymbol{X}_n^{(t)}; \boldsymbol{v})| > 0\}\left|\mathbb{1}\{y_n = +\} - \text{logit}_+^{(t)}(\boldsymbol{X}_n^{(t)})\right| \\
&\times \sum_{p \in \mathcal{P}(\boldsymbol{X}_n^{(t)}; \boldsymbol{v})} \mathbb{1}\{\langle \boldsymbol{w}_{+,r}^{(t)}, \alpha_{n,p}^{(t)} \boldsymbol{v} + \boldsymbol{\zeta}_{n,p}^{(t)}\rangle + b_{c,r}^{(t)} > 0\} \times O\left(\frac{1}{\log^9(d)}\right) + \Delta b_{+,r}^{(t)} \\
\leq& \frac{\eta}{NP} \left(O\left(\frac{1}{\log^9(d)}\right) - \frac{1}{\log^5(d)}\left(\sqrt{1-\iota} - \frac{1}{\log^9(d)}\right)\right) \\
&\times \left(\sum_{\boldsymbol{v} \in \mathcal{C}(\boldsymbol{v}_+)} \sum_{n=1}^{N} \mathbb{1}\{|\mathcal{P}(\boldsymbol{X}_n^{(t)}; \boldsymbol{v})| > 0\}\left|\mathbb{1}\{y_n = +\} - \text{logit}_+^{(t)}(\boldsymbol{X}_n^{(t)})\right| \right. \\
&\left. \times \sum_{p \in \mathcal{P}(\boldsymbol{X}_n^{(t)}; \boldsymbol{v})} \mathbb{1}\{\langle \boldsymbol{w}_{+,r}^{(t)}, \alpha_{n,p}^{(t)} \boldsymbol{v} + \boldsymbol{\zeta}_{n,p}^{(t)}\rangle + b_{c,r}^{(t)} > 0\}\right) \\
<& 0.
\end{aligned}
\tag{170}
$$

This completes the proof. $\qquad\square$

**Lemma F.3** (Nonactivation on noise patches)**.** *Let the assumptions in Theorem D.1 hold.*

*Denote the set of features $\mathcal{M} = \{\boldsymbol{v}_{+,c}\}_{c=1}^{k_+} \cup \{\boldsymbol{v}_{-,c}\}_{c=1}^{k_-} \cup \{\boldsymbol{v}_+, \boldsymbol{v}_-\}$. If the update term for neuron $\boldsymbol{w}_{+,r}^{(t)}$ can be written as follows*

$$
\begin{aligned}
\Delta \boldsymbol{w}_{+,r}^{(t)} = &\frac{\eta}{NP} \sum_{\boldsymbol{v}\in\mathcal{M}} \sum_{n=1}^{N} \mathbb{1}\{|\mathcal{P}(\boldsymbol{X}_n^{(t)}; \boldsymbol{v})| > 0\}[\mathbb{1}\{y_n = +\} - logit_+^{(t)}(\boldsymbol{X}_n^{(t)})] \\
&\times \sum_{p\in\mathcal{P}(\boldsymbol{X}_n^{(t)};\boldsymbol{v})} \mathbb{1}\{\langle \boldsymbol{w}_{+,r}^{(t)}, \alpha_{n,p}^{(t)}\boldsymbol{v} + \boldsymbol{\zeta}_{n,p}^{(t)} \rangle + b_{c,r}^{(t)} > 0\} \left(\alpha_{n,p}^{(t)}\boldsymbol{v} + \boldsymbol{\zeta}_{n,p}^{(t)}\right),
\end{aligned}
\tag{171}
$$

*then*

$$
\begin{aligned}
\Delta b_{+,r}^{(t)} \\
\leq -&\frac{1}{\log^5(d)} \frac{\eta}{NP} \left(\sqrt{1-\iota} - \frac{1}{\log^9(d)}\right) \\
&\times \left(\sum_{\boldsymbol{v}\in\mathcal{M}} \sum_{n=1}^{N} \mathbb{1}\{|\mathcal{P}(\boldsymbol{X}_n^{(t)}; \boldsymbol{v})| > 0\} \left|\mathbb{1}\{y_n = +\} - logit_+^{(t)}(\boldsymbol{X}_n^{(t)})\right| \right. \\
&\times \left. \sum_{p\in\mathcal{P}(\boldsymbol{X}_n^{(t)};\boldsymbol{v})} \mathbb{1}\{\langle \boldsymbol{w}_{+,r}^{(t)}, \alpha_{n,p}^{(t)}\boldsymbol{v} + \boldsymbol{\zeta}_{n,p}^{(t)} \rangle + b_{c,r}^{(t)} > 0\}\right).
\end{aligned}
\tag{172}
$$

*Moreover, for any $\tau > t$, the following inequality holds with probability at least $1 - O\left(\frac{NP}{poly(d)}\right)$ for all noise patches $\boldsymbol{x}_{n,p}^{(\tau)} = \boldsymbol{\zeta}_{n,p}^{(\tau)}$:*

$$
\langle \Delta \boldsymbol{w}_{+,r}^{(t)}, \boldsymbol{x}_{n,p}^{(\tau)} \rangle + \Delta b_{+,r}^{(t)} < 0
\tag{173}
$$

*Proof.* Similar to the proof of Lemma F.2, we can estimate the update to the bias term

$$
\begin{aligned}
\Delta b_{+,r}^{(t)} \\
\leq -&\frac{1}{\log^5(d)} \frac{\eta}{NP} \left(\sqrt{1-\iota} - \frac{1}{\log^9(d)}\right) \\
&\times \left(\sum_{\boldsymbol{v}\in\mathcal{M}} \sum_{n=1}^{N} \mathbb{1}\{|\mathcal{P}(\boldsymbol{X}_n^{(t)}; \boldsymbol{v})| > 0\} \left|\mathbb{1}\{y_n = +\} - logit_+^{(t)}(\boldsymbol{X}_n^{(t)})\right| \right. \\
&\times \left. \sum_{p\in\mathcal{P}(\boldsymbol{X}_n^{(t)};\boldsymbol{v})} \mathbb{1}\{\langle \boldsymbol{w}_{+,r}^{(t)}, \alpha_{n,p}^{(t)}\boldsymbol{v} + \boldsymbol{\zeta}_{n,p}^{(t)} \rangle + b_{c,r}^{(t)} > 0\}\right)
\end{aligned}
\tag{174}
$$

Then for any $\boldsymbol{x}_{n,p}^{(\tau)} = \boldsymbol{\zeta}_{n,p}^{(\tau)}$ with $\tau > t$, with probability at least $1 - O\left(\frac{mNP}{\text{poly}(d)}\right)$,

$$
\begin{aligned}
&\langle \Delta \boldsymbol{w}_{+,r}^{(t)}, \boldsymbol{x}_{n,p}^{(\tau)} \rangle + \Delta b_{+,r}^{(t)} \\
=&\langle \Delta \boldsymbol{w}_{+,r}^{(t)}, \boldsymbol{\zeta}_{n,p}^{(\tau)} \rangle + \Delta b_{+,r}^{(t)} \\
=&\frac{\eta}{NP} \sum_{\boldsymbol{v}\in\mathcal{M}} \sum_{n=1}^{N} \mathbb{1}\{|\mathcal{P}(\boldsymbol{X}_n^{(t)};\boldsymbol{v})| > 0\}[\mathbb{1}\{y_n = +\} - \text{logit}_+^{(t)}(\boldsymbol{X}_n^{(t)})] \\
&\times \sum_{p\in\mathcal{P}(\boldsymbol{X}_n^{(t)};\boldsymbol{v})} \mathbb{1}\{\langle \boldsymbol{w}_{+,r}^{(t)}, \alpha_{n,p}^{(t)}\boldsymbol{v} + \boldsymbol{\zeta}_{n,p}^{(t)}\rangle + b_{c,r}^{(t)} > 0\}\langle \alpha_{n,p}^{(t)}\boldsymbol{v} + \boldsymbol{\zeta}_{n,p}^{(t)}, \boldsymbol{\zeta}_{n,p}^{(\tau)}\rangle + \Delta b_{+,r}^{(t)} \\
=&\frac{\eta}{NP} \sum_{\boldsymbol{v}\in\mathcal{M}} \sum_{n=1}^{N} \mathbb{1}\{|\mathcal{P}(\boldsymbol{X}_n^{(t)};\boldsymbol{v})| > 0\}[\mathbb{1}\{y_n = +\} - \text{logit}_+^{(t)}(\boldsymbol{X}_n^{(t)})] \\
&\times \sum_{p\in\mathcal{P}(\boldsymbol{X}_n^{(t)};\boldsymbol{v})} \mathbb{1}\{\langle \boldsymbol{w}_{+,r}^{(t)}, \alpha_{n,p}^{(t)}\boldsymbol{v} + \boldsymbol{\zeta}_{n,p}^{(t)}\rangle + b_{c,r}^{(t)} > 0\} \left(\langle \alpha_{n,p}^{(t)}\boldsymbol{v}, \boldsymbol{\zeta}_{n,p}^{(\tau)}\rangle + \langle \boldsymbol{\zeta}_{n,p}^{(t)}, \boldsymbol{\zeta}_{n,p}^{(\tau)}\rangle\right) + \Delta b_{+,r}^{(t)} \\
\leq&\frac{\eta}{NP} \sum_{\boldsymbol{v}\in\mathcal{M}} \sum_{n=1}^{N} \mathbb{1}\{|\mathcal{P}(\boldsymbol{X}_n^{(t)};\boldsymbol{v})| > 0\} \left|\mathbb{1}\{y_n = +\} - \text{logit}_+^{(t)}(\boldsymbol{X}_n^{(t)})\right| \\
&\times \sum_{p\in\mathcal{P}(\boldsymbol{X}_n^{(t)};\boldsymbol{v})} \mathbb{1}\{\langle \boldsymbol{w}_{+,r}^{(t)}, \alpha_{n,p}^{(t)}\boldsymbol{v} + \boldsymbol{\zeta}_{n,p}^{(t)}\rangle + b_{c,r}^{(t)} > 0\} \times O\left(\frac{1}{\log^9(d)}\right) + \Delta b_{+,r}^{(t)} \\
\leq&\frac{\eta}{NP}\left(O\left(\frac{1}{\log^9(d)}\right) - \frac{1}{\log^5(d)}\left(\sqrt{1-\iota} - \frac{1}{\log^9(d)}\right)\right) \\
&\times \left(\sum_{\boldsymbol{v}\in\mathcal{M}} \sum_{n=1}^{N} \mathbb{1}\{|\mathcal{P}(\boldsymbol{X}_n^{(t)};\boldsymbol{v})| > 0\} \left|\mathbb{1}\{y_n = +\} - \text{logit}_+^{(t)}(\boldsymbol{X}_n^{(t)})\right| \right. \\
&\left. \times \sum_{p\in\mathcal{P}(\boldsymbol{X}_n^{(t)};\boldsymbol{v})} \mathbb{1}\{\langle \boldsymbol{w}_{+,r}^{(t)}, \alpha_{n,p}^{(t)}\boldsymbol{v} + \boldsymbol{\zeta}_{n,p}^{(t)}\rangle + b_{c,r}^{(t)} > 0\}\right) \\
<&0.
\end{aligned}
\tag{175}
$$

$\square$

## G Fine-grained Learning

This section treats the learning dynamics of using fine-grained labels to train the NN; the analysis will be much simpler since the technical analysis overlaps significantly with that in the previous sections.

The training procedure is exactly the same as in the coarse-grained training setting. We explicitly write them out here to avoid any possible confusion.

The learner for fine-grained classification is written as follows for $c \in [k_+]$:

$$F_{+,c}(\boldsymbol{X}) = \sum_{r=1}^{m_{+,c}} a_{+,c,r} \sum_{p=1}^{P} \sigma(\langle \boldsymbol{w}_{+,c,r}, \boldsymbol{x}_p \rangle + b_{+,c,r}), \quad c \in [k_+] \tag{176}$$

with frozen linear classifier weights $a_{+,c,r} = 1$. Same definition applies to the $-$ classes.

The SGD dynamics induced by the training loss is now

$$\boldsymbol{w}_{+,c,r}^{(t+1)} = \boldsymbol{w}_{+,c,r}^{(t)} + \eta \frac{1}{NP} \sum_{n=1}^{N} \Bigg( \mathbb{1}\{y_n = (+,c)\}[1 - \mathrm{logit}_{+,c}^{(t)}(\boldsymbol{X}_n^{(t)})] \sum_{p \in [P]} \sigma'(\langle \boldsymbol{w}_{+,c,r}^{(t)}, \boldsymbol{x}_{n,p}^{(t)} \rangle + b_{c,r}^{(t)}) \boldsymbol{x}_{n,p}^{(t)} +$$
$$\mathbb{1}\{y_n \neq (+,c)\}[-\mathrm{logit}_{+,c}^{(t)}(\boldsymbol{X}_n^{(t)})] \sum_{p \in [P]} \sigma'(\langle \boldsymbol{w}_{+,c,r}^{(t)}, \boldsymbol{x}_{n,p}^{(t)} \rangle + b_{c,r}^{(t)}) \boldsymbol{x}_{n,p}^{(t)} \Bigg) \tag{177}$$

The bias is manually tuned according to the update rule

$$b_{+,c,r}^{(t+1)} = b_{+,c,r}^{(t)} - \frac{\|\Delta \boldsymbol{w}_{+,c,r}^{(t)}\|_2}{\log^5(d)} \tag{178}$$

We assign $m_{+,c} = \Theta(d^{1+2c_0})$ neurons to each subclass $(+,c)$. For convenience, we write $m = dm_{+,c}$.

The initialization scheme is identical to the coarse-training case, except we choose a slightly less negative $b_{c,r}^{(0)} = -\sigma_0\sqrt{2 + 2c_0}\sqrt{\log(d)}$.

The parameter choices remain the same as before.

### G.1 Initialization geometry

**Definition G.1.** Define the following sets of interest of the hidden neurons:

1. $\mathcal{U}_{+,c,r}^{(0)} = \{\boldsymbol{v} \in \mathcal{V} : \langle \boldsymbol{w}_{+,c,r}^{(0)}, \boldsymbol{v} \rangle \geq \sigma_0\sqrt{2 + 2c_0}\sqrt{\log(d) - \frac{1}{\log^5(d)}}\}$

2. Given $\boldsymbol{v} \in \mathcal{V}$, $S_{+,c}^{*(0)}(\boldsymbol{v}) \subseteq (+,c) \times [m_{+,c}]$ satisfies:

   (a) $\langle \boldsymbol{w}_{+,c,r}^{(0)}, \boldsymbol{v} \rangle \geq \sigma_0\sqrt{2 + 2c_0}\sqrt{\log(d) + \frac{1}{\log^5(d)}}$

   (b) $\forall \boldsymbol{v}' \in \mathcal{V}$ s.t. $\boldsymbol{v}' \perp \boldsymbol{v}$, $\langle \boldsymbol{w}_{+,c,r}^{(0)}, \boldsymbol{v}' \rangle < \sigma_0\sqrt{2 + 2c_0}\sqrt{\log(d) - \frac{1}{\log^5(d)}}$

3. Given $\boldsymbol{v} \in \mathcal{V}$, $S_{+,c}^{(0)}(\boldsymbol{v}) \subseteq (+,c) \times [m_{+,c}]$ satisfies:

   (a) $\langle \boldsymbol{w}_{+,c,r}^{(0)}, \boldsymbol{v} \rangle \geq \sigma_0\sqrt{2 + 2c_0}\sqrt{\log(d) - \frac{1}{\log^5(d)}}$

4. For any $(+,c,r) \in S_{+,c,reg}^{*(0)} \subseteq (+,c) \times [m_{+,c}]$:

   (a) $\langle \boldsymbol{w}_{+,c,r}^{(0)}, \boldsymbol{v} \rangle \leq \sigma_0\sqrt{10}\sqrt{\log(d)} \ \forall \boldsymbol{v} \in \mathcal{V}$

   (b) $\left|\mathcal{U}_{+,c,r}^{(0)}\right| \leq O(1)$

The same definitions apply to the $-$-class neurons.

**Proposition 2.** *At $t = 0$, for all $\boldsymbol{v} \in \mathcal{D}$, the following properties are true with probability at least $1 - d^{-2}$ over the randomness of the initialized kernels:*

1. $|S_{+,c}^{*(0)}(\boldsymbol{v})|, |S_{+,c}^{(0)}(\boldsymbol{v})| = \Theta\left(\frac{1}{\sqrt{\log(d)}}\right) d^{c_0}$

2. *In particular,* $\left|\frac{|S_y^{*(0)}(\boldsymbol{v})|}{|S_{y'}^{(0)}(\boldsymbol{v}')|} - 1\right| = O\left(\frac{1}{\log^5(d)}\right)$ *and* $\left|\frac{|S_y^{*(0)}(\boldsymbol{v})|}{|S_{y'}^{*(0)}(\boldsymbol{v}')|} - 1\right| = O\left(\frac{1}{\log^5(d)}\right)$ *for any* $y, y' \in \{(+,c)\}_{c=1}^{k_+} \cup \{(-,c)\}_{c=1}^{k_-}$ *and common or fine-grained features* $\boldsymbol{v}, \boldsymbol{v}'$.

3. $S_{+,c,reg}^{(0)} = [m_{+,c}]$

*The same properties apply to the $-$-class neurons.*

*Proof.* This proof proceeds in virtually the same way as in the proof of Proposition 1, so we omit it here. $\square$

## G.2 Poly-time properties

**Theorem G.1.** *Fix any $t \in [0, T_e]$, assuming $T_e \in poly(d)$.*

1. *(Non-activation invariance) For any $\tau \geq t$, with probability at least $1 - O\left(\frac{mk_+ + NPt}{poly(d)}\right)$, for any feature* $\boldsymbol{v} \in \{\boldsymbol{v}_{+,c}\}_{c=1}^{k_+} \cup \{\boldsymbol{v}_{-,c}\}_{c=1}^{k_-} \cup \{\boldsymbol{v}_+, \boldsymbol{v}_-\}$, *for every $t' \leq t$, $(+,c,r) \notin S_{+,c}^{(0)}(\boldsymbol{v})$ and $\boldsymbol{v}$-dominated patch sample $\boldsymbol{x}_{n,p}^{(\tau)} = \alpha_{n,p}^{(\tau)}\boldsymbol{v} + \boldsymbol{\zeta}_{n,p}^{(\tau)}$, the following holds:*

$$\sigma\left(\langle \boldsymbol{w}_{+,c,r}^{(t')}, \boldsymbol{x}_{n,p}^{(\tau)}\rangle + b_{+,c,r}^{(t')}\right) = 0 \tag{179}$$

2. *(Non-activation on noise patches) For any $\tau \geq t$, with probability at least $1 - O\left(\frac{mNPt}{poly(d)}\right)$, for every* $c \in [k_+]$, $r \in [m]$ *and noise patch $\boldsymbol{x}_{n,p}^{(\tau)} = \boldsymbol{\zeta}_{n,p}^{(\tau)}$, the following holds:*

$$\sigma\left(\langle \boldsymbol{w}_{+,c,r}^{(t)}, \boldsymbol{x}_{n,p}^{(\tau)}\rangle + b_{+,c,r}^{(t)}\right) = 0 \tag{180}$$

3. *(Off-diagonal nonpositive growth) Given fine-grained class $(+,c)$ and any $\tau \geq t$, with probability at least $1 - O\left(\frac{mk_+ + NPt}{poly(d)}\right)$, for any $t' \leq t$, any feature $\boldsymbol{v} \in \{\boldsymbol{v}_{-,c}\}_{c=1}^{k_-} \cup \{\boldsymbol{v}_-\} \cup \{\boldsymbol{v}_{+,c'}\}_{c' \neq c}$, any neuron* $\boldsymbol{w}_{+,c,r} \in S_{+,c}^{(0)}(\boldsymbol{v})$ *and any $\boldsymbol{v}$-dominated patch $\boldsymbol{x}_{n,p}^{(\tau)} = \alpha_{n,p}^{(\tau)}\boldsymbol{v} + \boldsymbol{\zeta}_{n,p}^{(\tau)}$, $\sigma\left(\langle \boldsymbol{w}_{+,c,r}^{(t')}, \boldsymbol{x}_{n,p}^{(\tau)}\rangle + b_{+,c,r}^{(t')}\right) \leq \sigma\left(\langle \boldsymbol{w}_{+,c,r}^{(0)}, \boldsymbol{x}_{n,p}^{(\tau)}\rangle + b_{+,c,r}^{(0)}\right)$.*

*Proof.* The proof of this theorem is similar to that of Theorem F.1, but with some subtle differences.

**Base case** $t = 0$.

*1. (Nonactivation invariance)*

Choose any $\boldsymbol{v}^*$ from the set $\{\boldsymbol{v}_{+,c}\}_{c=1}^{k_+} \cup \{\boldsymbol{v}_{-,c}\}_{c=1}^{k_-} \cup \{\boldsymbol{v}_+, \boldsymbol{v}_-\}$. We will work with neuron sets in the "+" class in this proof; the "$-$"-class case can be handled in the same way.

First, given $\tau \geq 0$, we need to show that, for every $n$ such that $|\mathcal{P}(\boldsymbol{X}_n^{(\tau)}; \boldsymbol{v}^*)| > 0$ and $p \in \mathcal{P}(\boldsymbol{X}_n^{(\tau)}; \boldsymbol{v}^*)$, for every $(+,c,r)$ neuron index,

$$\langle \boldsymbol{w}_{+,c,r}^{(0)}, \boldsymbol{v}^*\rangle < \sigma_0\sqrt{2 + 2c_0}\sqrt{\log(d) - \frac{1}{\log^5(d)}} \implies \sigma\left(\langle \boldsymbol{w}_{+,c,r}^{(0)}, \boldsymbol{x}_{n,p}^{(\tau)}\rangle + b_{+,c,r}^{(0)}\right) = 0 \tag{181}$$

This is indeed true. The following holds with probability at least $1 - O\left(\frac{mNP}{\text{poly}(d)}\right)$ for all $(+, r) \notin S_+^{(0)}(\boldsymbol{v})$ and all such $\boldsymbol{x}_{n,p}^{(\tau)}$:

$$
\begin{aligned}
&\langle \boldsymbol{w}_{+,c,r}^{(0)}, \boldsymbol{x}_{n,p}^{(\tau)}\rangle + b_{+,c,r}^{(0)} \\
\leq &\sigma_0\sqrt{1+\iota}\sqrt{(2+2c_0)(\log(d) - 1/\log^5(d))} + O\left(\frac{\sigma_0}{\log^9(d)}\right) - \sqrt{2+2c_0}\sqrt{\log(d)}\sigma_0 \\
= &\sigma_0\left(\frac{(2+2c_0)(1+\iota)(\log(d) - 1/\log^5(d)) - (2+2c_0)\log(d)}{\sqrt{(2+2c_0)(\log(d) - 1/\log^5(d))} + \sqrt{4+2c_0}\sqrt{\log(d)}} + O\left(\frac{1}{\log^9(d)}\right)\right) \\
= &\sigma_0\left(\frac{(2+2c_0)(\iota\log(d) - (1+\iota)/\log^5(d))}{\sqrt{(2+2c_0)(\log(d) - 1/\log^5(d))} + \sqrt{2+2c_0}\sqrt{\log(d)}} + O\left(\frac{1}{\log^9(d)}\right)\right) \\
< &0,
\end{aligned}
\tag{182}
$$

The first equality holds by utilizing the identity $a - b = \frac{a^2 - b^2}{a+b}$. As a consequence, $\sigma(\langle \boldsymbol{w}_{+,c,r}^{(0)}, \boldsymbol{x}_{n,p}^{(\tau)}\rangle + b_{+,r}^{(0)}) = 0$.

*2. (Non-activation on noise patches)* Invoking Lemma H.3, for any $\tau \geq 0$, with probability at least $1 - O\left(\frac{mNP}{\text{poly}(d)}\right)$, we have for all possible choices of $r \in [m]$ and the noise patches $\boldsymbol{x}_{n,p}^{(\tau)} = \boldsymbol{\zeta}_{n,p}^{(\tau)}$:

$$
\left|\langle \boldsymbol{w}_{+,c,r}^{(0)}, \boldsymbol{\zeta}_{n,p}^{(\tau)}\rangle\right| \leq O(\sigma_0\sigma_\zeta\sqrt{d\log(d)}) \leq O\left(\frac{\sigma_0}{\log^9(d)}\right) \ll b_{+,r}^{(0)}.
\tag{183}
$$

Therefore, no neuron can activate on the noise patches at time $t = 0$.

*3. (Off-diagonal nonpositive growth)* This point is trivially true at $t = 0$.

**Inductive step**: we assume the induction hypothesis for $t \in [0, T]$ (with $T < T_e$ of course), and prove the statements for $t = T + 1$.

*1. (Nonactivation invariance)*

Again, choose any $\boldsymbol{v}^*$ from the set $\{\boldsymbol{v}_{+,c}\}_{c=1}^{k_+} \cup \{\boldsymbol{v}_{-,c}\}_{c=1}^{k_-} \cup \{\boldsymbol{v}_+, \boldsymbol{v}_-\}$.

We need to prove that given $\tau \geq T + 1$, with probability at least $1 - O\left(\frac{mk_+NP(T+1)}{\text{poly}(d)}\right)$, for every $t' \leq T + 1$, $(+, c, r)$ neuron index and $\boldsymbol{v}^*$-dominated patch $\boldsymbol{x}_{n,p}^{(\tau)}$,

$$
(+, c, r) \notin S_{+,c}^{(0)}(\boldsymbol{v}^*) \implies \sigma\left(\langle \boldsymbol{w}_{+,c,r}^{(t')}, \boldsymbol{x}_{n,p}^{(\tau)}\rangle + b_{+,c,r}^{(t')}\right) = 0.
\tag{184}
$$

By the induction hypothesis of point 1., with probability at least $1 - O\left(\frac{mk_+NPT}{\text{poly}(d)}\right)$, the following is already true on all the $\boldsymbol{v}^*$-dominated patches at time $t' \leq T$:

$$
(+, c, r) \notin S_{+,c}^{(0)}(\boldsymbol{v}^*) \implies \sigma\left(\langle \boldsymbol{w}_{+,c,r}^{(t')}, \boldsymbol{x}_{n,p}^{(T)}\rangle + b_{+,c,r}^{(t')}\right) = 0.
\tag{185}
$$

In particular, $\sigma\left(\langle \boldsymbol{w}_{+,c,r}^{(T)}, \boldsymbol{x}_{n,p}^{(T)}\rangle + b_{+,c,r}^{(T)}\right) = 0$.

In other words, no $(+, c, r) \notin S_{+,c}^{(0)}(\boldsymbol{v}^*)$ can be updated on the $\boldsymbol{v}^*$-dominated patches at time $t = T$. Furthermore, the induction hypothesis of point 2. also states that the network cannot activate on any noise patch $\boldsymbol{x}_{n,p}^{(T)} = \boldsymbol{\zeta}_{n,p}^{(T)}$ with probability at least $1 - O\left(\frac{mNPT}{\text{poly}(d)}\right)$. Therefore, the neuron update for those

$(+, c, r) \notin S_{+,c}^{(0)}(\boldsymbol{v}^*)$ takes the form

$$
\begin{aligned}
\Delta \boldsymbol{w}_{+,c,r}^{(T)} = \frac{\eta}{NP} &\sum_{\boldsymbol{v} \in \mathcal{C}(\boldsymbol{v}^*)} \sum_{n=1}^{N} \mathbb{1}\{|\mathcal{P}(\boldsymbol{X}_n^{(T)}; \boldsymbol{v})| > 0\}[\mathbb{1}\{y_n = (+, c)\} - \mathrm{logit}_{+,c}^{(T)}(\boldsymbol{X}_n^{(T)})] \\
&\times \sum_{p \in \mathcal{P}(\boldsymbol{X}_n^{(T)}; \boldsymbol{v})} \mathbb{1}\{\langle \boldsymbol{w}_{+,c,r}^{(T)}, \alpha_{n,p}^{(T)} \boldsymbol{v} + \boldsymbol{\zeta}_{n,p}^{(T)} \rangle + b_{+,c,r}^{(T)} > 0\} \left( \alpha_{n,p}^{(T)} \boldsymbol{v} + \boldsymbol{\zeta}_{n,p}^{(T)} \right)
\end{aligned}
\tag{186}
$$

Conditioning on this high-probability event, we have

$$
\begin{aligned}
\Delta b_{+,c,r}^{(t)} = &- \frac{\left\| \Delta \boldsymbol{w}_{+,c,r}^{(t)} \right\|_2}{\log^5(d)} \\
\leq &- \frac{1}{\log^5(d)} \frac{\eta}{NP} \left\| \sum_{\boldsymbol{v} \in \mathcal{C}(\boldsymbol{v}^*)} \sum_{n=1}^{N} \mathbb{1}\{|\mathcal{P}(\boldsymbol{X}_n^{(t)}; \boldsymbol{v})| > 0\}[\mathbb{1}\{y_n = (+, c)\} - \mathrm{logit}_{+,c}^{(t)}(\boldsymbol{X}_n^{(t)})] \right. \\
&\times \left. \sum_{p \in \mathcal{P}(\boldsymbol{X}_n^{(t)}; \boldsymbol{v})} \mathbb{1}\{\langle \boldsymbol{w}_{+,c,r}^{(t)}, \alpha_{n,p}^{(t)} \boldsymbol{v} + \boldsymbol{\zeta}_{n,p}^{(t)} \rangle + b_{+,c,r}^{(t)} > 0\} \alpha_{n,p}^{(t)} \boldsymbol{v} \right\|_2 \\
&+ \frac{1}{\log^5(d)} \frac{\eta}{NP} \left\| \sum_{\boldsymbol{v} \in \mathcal{C}(\boldsymbol{v}^*)} \sum_{n=1}^{N} \mathbb{1}\{|\mathcal{P}(\boldsymbol{X}_n^{(t)}; \boldsymbol{v})| > 0\}[\mathbb{1}\{y_n = (+, c)\} - \mathrm{logit}_{+,c}^{(t)}(\boldsymbol{X}_n^{(t)})] \right. \\
&\times \left. \sum_{p \in \mathcal{P}(\boldsymbol{X}_n^{(t)}; \boldsymbol{v})} \mathbb{1}\{\langle \boldsymbol{w}_{+,c,r}^{(t)}, \alpha_{n,p}^{(t)} \boldsymbol{v} + \boldsymbol{\zeta}_{n,p}^{(t)} \rangle + b_{+,c,r}^{(t)} > 0\} \boldsymbol{\zeta}_{n,p}^{(t)} \right\|_2
\end{aligned}
\tag{187}
$$

Let us further upper bound the two $\| \cdot \|_2$ terms separately. Firstly,

$$
\begin{aligned}
&\left\| \sum_{\boldsymbol{v} \in \mathcal{C}(\boldsymbol{v}^*)} \sum_{n=1}^{N} \mathbb{1}\{|\mathcal{P}(\boldsymbol{X}_n^{(t)}; \boldsymbol{v})| > 0\}[\mathbb{1}\{y_n = (+, c)\} - \mathrm{logit}_{+,c}^{(t)}(\boldsymbol{X}_n^{(t)})] \right. \\
&\qquad \times \left. \sum_{p \in \mathcal{P}(\boldsymbol{X}_n^{(t)}; \boldsymbol{v})} \mathbb{1}\{\langle \boldsymbol{w}_{+,c,r}^{(t)}, \alpha_{n,p}^{(t)} \boldsymbol{v} + \boldsymbol{\zeta}_{n,p}^{(t)} \rangle + b_{+,c,r}^{(t)} > 0\} \alpha_{n,p}^{(t)} \boldsymbol{v} \right\|_2 \\
&= \sum_{\boldsymbol{v} \in \mathcal{C}(\boldsymbol{v}^*)} \sum_{n=1}^{N} \mathbb{1}\{|\mathcal{P}(\boldsymbol{X}_n^{(t)}; \boldsymbol{v})| > 0\} \left| \mathbb{1}\{y_n = (+, c)\} - \mathrm{logit}_{+,c}^{(t)}(\boldsymbol{X}_n^{(t)}) \right| \\
&\qquad \times \sum_{p \in \mathcal{P}(\boldsymbol{X}_n^{(t)}; \boldsymbol{v})} \mathbb{1}\{\langle \boldsymbol{w}_{+,c,r}^{(t)}, \alpha_{n,p}^{(t)} \boldsymbol{v} + \boldsymbol{\zeta}_{n,p}^{(t)} \rangle + b_{+,c,r}^{(t)} > 0\} \alpha_{n,p}^{(t)} \|\boldsymbol{v}\|_2 \\
&\geq \sum_{\boldsymbol{v} \in \mathcal{C}(\boldsymbol{v}^*)} \sum_{n=1}^{N} \mathbb{1}\{|\mathcal{P}(\boldsymbol{X}_n^{(t)}; \boldsymbol{v})| > 0\} \left| \mathbb{1}\{y_n = (+, c)\} - \mathrm{logit}_{+,c}^{(t)}(\boldsymbol{X}_n^{(t)}) \right| \\
&\qquad \times \sum_{p \in \mathcal{P}(\boldsymbol{X}_n^{(t)}; \boldsymbol{v})} \mathbb{1}\{\langle \boldsymbol{w}_{+,c,r}^{(t)}, \alpha_{n,p}^{(t)} \boldsymbol{v} + \boldsymbol{\zeta}_{n,p}^{(t)} \rangle + b_{+,c,r}^{(t)} > 0\} \sqrt{1 - \iota}
\end{aligned}
\tag{188}
$$

For the second $\| \cdot \|_2$ term consisting purely of noise, note that since all the $\boldsymbol{\zeta}_{n,p}^{(t)}$'s are independent Gaussian random vectors, the standard deviation of the sum is in fact

$$
\begin{aligned}
\left\{ \sum_{\boldsymbol{v} \in \mathcal{C}(\boldsymbol{v}^*)} \sum_{n=1}^{N} \sum_{p \in \mathcal{P}(\boldsymbol{X}_n^{(t)}; \boldsymbol{v})} \mathbb{1}\{|\mathcal{P}(\boldsymbol{X}_n^{(t)}; \boldsymbol{v})| > 0\} \mathbb{1}\{\langle \boldsymbol{w}_{+,c,r}^{(t)}, \alpha_{n,p}^{(t)} \boldsymbol{v} + \boldsymbol{\zeta}_{n,p}^{(t)} \rangle + b_{+,c,r}^{(t)} > 0\} \right. \\
\left. \times [\mathbb{1}\{y_n = (+, c)\} - \mathrm{logit}_{+,c}^{(t)}(\boldsymbol{X}_n^{(t)})]^2 \right\}^{1/2} \sigma_\zeta.
\end{aligned}
\tag{189}
$$

With the basic property that $\sqrt{\sum_j c_j^2} \leq \sum_j |c_j|$ for any sequence of real numbers $c_1, c_2, \ldots$, we know this standard deviation can be upper bounded by

$$
\sum_{\boldsymbol{v} \in \mathcal{C}(\boldsymbol{v}^*)} \sum_{n=1}^{N} \sum_{p \in \mathcal{P}(\boldsymbol{X}_n^{(t)}; \boldsymbol{v})} \mathbb{1}\{|\mathcal{P}(\boldsymbol{X}_n^{(t)}; \boldsymbol{v})| > 0\} \mathbb{1}\{\langle \boldsymbol{w}_{+,c,r}^{(t)}, \alpha_{n,p}^{(t)} \boldsymbol{v} + \boldsymbol{\zeta}_{n,p}^{(t)} \rangle + b_{+,c,r}^{(t)} > 0\}
$$
$$
\times \left| \mathbb{1}\{y_n = (+,c)\} - \text{logit}_{+,c}^{(t)}(\boldsymbol{X}_n^{(t)}) \right| \sigma_\zeta
\tag{190}
$$

It follows that with probability at least $1 - O\left(\frac{1}{\text{poly}(d)}\right)$,

$$
\left\| \sum_{\boldsymbol{v} \in \mathcal{C}(\boldsymbol{v}^*)} \sum_{n=1}^{N} \mathbb{1}\{|\mathcal{P}(\boldsymbol{X}_n^{(t)}; \boldsymbol{v})| > 0\} [\mathbb{1}\{y_n = (+,c)\} - \text{logit}_{+,c}^{(t)}(\boldsymbol{X}_n^{(t)})] \right.
$$
$$
\left. \times \sum_{p \in \mathcal{P}(\boldsymbol{X}_n^{(t)}; \boldsymbol{v})} \mathbb{1}\{\langle \boldsymbol{w}_{+,c,r}^{(t)}, \alpha_{n,p}^{(t)} \boldsymbol{v} + \boldsymbol{\zeta}_{n,p}^{(t)} \rangle + b_{+,c,r}^{(t)} > 0\} \boldsymbol{\zeta}_{n,p}^{(t)} \right\|_2
$$
$$
\leq \sum_{\boldsymbol{v} \in \mathcal{C}(\boldsymbol{v}^*)} \sum_{n=1}^{N} \mathbb{1}\{|\mathcal{P}(\boldsymbol{X}_n^{(t)}; \boldsymbol{v})| > 0\} \left| \mathbb{1}\{y_n = (+,c)\} - \text{logit}_{+,c}^{(t)}(\boldsymbol{X}_n^{(t)}) \right|
$$
$$
\times \sum_{p \in \mathcal{P}(\boldsymbol{X}_n^{(t)}; \boldsymbol{v})} \mathbb{1}\{\langle \boldsymbol{w}_{+,c,r}^{(t)}, \alpha_{n,p}^{(t)} \boldsymbol{v} + \boldsymbol{\zeta}_{n,p}^{(t)} \rangle + b_{+,c,r}^{(t)} > 0\} \frac{1}{\log^9(d)}
\tag{191}
$$

Therefore, we can upper bound the bias update as follows:

$$
\Delta b_{+,c,r}^{(t)} \leq -\frac{1}{\log^5(d)} \frac{\eta}{NP} \left( \sqrt{1-\iota} - \frac{1}{\log^9(d)} \right)
$$
$$
\times \left( \sum_{\boldsymbol{v} \in \mathcal{C}(\boldsymbol{v}^*)} \sum_{n=1}^{N} \mathbb{1}\{|\mathcal{P}(\boldsymbol{X}_n^{(t)}; \boldsymbol{v})| > 0\} \left| \mathbb{1}\{y_n = (+,c)\} - \text{logit}_{+,c}^{(t)}(\boldsymbol{X}_n^{(t)}) \right| \right.
$$
$$
\left. \times \sum_{p \in \mathcal{P}(\boldsymbol{X}_n^{(t)}; \boldsymbol{v})} \mathbb{1}\{\langle \boldsymbol{w}_{+,c,r}^{(t)}, \alpha_{n,p}^{(t)} \boldsymbol{v} + \boldsymbol{\zeta}_{n,p}^{(t)} \rangle + b_{+,c,r}^{(t)} > 0\} \right)
\tag{192}
$$

Furthermore, with probability at least $1 - O\left(\frac{NP}{\text{poly}(d)}\right)$, the following holds for all $n, p$:

$$
\langle \alpha_{n,p}^{(t)} \boldsymbol{v}, \boldsymbol{\zeta}_{n,p}^{(\tau)} \rangle, \ \langle \boldsymbol{\zeta}_{n,p}^{(t)}, \alpha_{n,p}^{(\tau)} \boldsymbol{v}^* \rangle, \ \langle \boldsymbol{\zeta}_{n,p}^{(t)}, \boldsymbol{\zeta}_{n,p}^{(\tau)} \rangle < O\left(\frac{1}{\log^9(d)}\right).
\tag{193}
$$

Combining the above derivations, they imply that with probability at least $1 - O\left(\frac{NP}{\text{poly}(d)}\right)$, for any $\boldsymbol{x}_{n,p}^{(\tau)}$ dominated by $\boldsymbol{v}^*$,

$$
\begin{aligned}
&\langle \Delta \boldsymbol{w}_{+,c,r}^{(t)}, \boldsymbol{x}_{n,p}^{(\tau)} \rangle + \Delta b_{+,c,r}^{(t)} \\
&= \langle \Delta \boldsymbol{w}_{+,c,r}^{(t)}, \alpha_{n,p}^{(\tau)} \boldsymbol{v}^* + \boldsymbol{\zeta}_{n,p}^{(\tau)} \rangle + \Delta b_{+,c,r}^{(t)} \\
&= \frac{\eta}{NP} \sum_{\boldsymbol{v} \in \mathcal{C}(\boldsymbol{v}^*)} \sum_{n=1}^{N} \mathbb{1}\{|\mathcal{P}(\boldsymbol{X}_n^{(t)}; \boldsymbol{v})| > 0\}[\mathbb{1}\{y_n = (+,c)\} - \text{logit}_{+,c}^{(t)}(\boldsymbol{X}_n^{(t)})] \\
&\quad \times \sum_{p \in \mathcal{P}(\boldsymbol{X}_n^{(t)}; \boldsymbol{v})} \mathbb{1}\{\langle \boldsymbol{w}_{+,c,r}^{(t)}, \alpha_{n,p}^{(t)} \boldsymbol{v} + \boldsymbol{\zeta}_{n,p}^{(t)} \rangle + b_{+,c,r}^{(t)} > 0\} \langle \alpha_{n,p}^{(t)} \boldsymbol{v} + \boldsymbol{\zeta}_{n,p}^{(t)}, \alpha_{n,p}^{(\tau)} \boldsymbol{v}^* + \boldsymbol{\zeta}_{n,p}^{(\tau)} \rangle + \Delta b_{+,c,r}^{(t)} \\
&= \frac{\eta}{NP} \sum_{\boldsymbol{v} \in \mathcal{C}(\boldsymbol{v}^*)} \sum_{n=1}^{N} \mathbb{1}\{|\mathcal{P}(\boldsymbol{X}_n^{(t)}; \boldsymbol{v})| > 0\}[\mathbb{1}\{y_n = (+,c)\} - \text{logit}_{+,c}^{(t)}(\boldsymbol{X}_n^{(t)})] \\
&\quad \times \sum_{p \in \mathcal{P}(\boldsymbol{X}_n^{(t)}; \boldsymbol{v})} \mathbb{1}\{\langle \boldsymbol{w}_{+,c,r}^{(t)}, \alpha_{n,p}^{(t)} \boldsymbol{v} + \boldsymbol{\zeta}_{n,p}^{(t)} \rangle + b_{+,c,r}^{(t)} > 0\} \left( \langle \alpha_{n,p}^{(t)} \boldsymbol{v}, \boldsymbol{\zeta}_{n,p}^{(\tau)} \rangle + \langle \boldsymbol{\zeta}_{n,p}^{(t)}, \alpha_{n,p}^{(\tau)} \boldsymbol{v}^* \rangle + \langle \boldsymbol{\zeta}_{n,p}^{(t)}, \boldsymbol{\zeta}_{n,p}^{(\tau)} \rangle \right) \\
&\quad + \Delta b_{+,c,r}^{(t)} \\
&\leq \frac{\eta}{NP} \sum_{\boldsymbol{v} \in \mathcal{C}(\boldsymbol{v}^*)} \sum_{n=1}^{N} \mathbb{1}\{|\mathcal{P}(\boldsymbol{X}_n^{(t)}; \boldsymbol{v})| > 0\} \left| \mathbb{1}\{y_n = (+,c)\} - \text{logit}_{+,c}^{(t)}(\boldsymbol{X}_n^{(t)}) \right| \\
&\quad \times \sum_{p \in \mathcal{P}(\boldsymbol{X}_n^{(t)}; \boldsymbol{v})} \mathbb{1}\{\langle \boldsymbol{w}_{+,c,r}^{(t)}, \alpha_{n,p}^{(t)} \boldsymbol{v} + \boldsymbol{\zeta}_{n,p}^{(t)} \rangle + b_{+,c,r}^{(t)} > 0\} \times O\left(\frac{1}{\log^9(d)}\right) + \Delta b_{+,c,r}^{(t)} \\
&\leq \frac{\eta}{NP} \left( O\left(\frac{1}{\log^9(d)}\right) - \frac{1}{\log^5(d)} \left( \sqrt{1-\iota} - \frac{1}{\log^9(d)} \right) \right) \\
&\quad \times \left( \sum_{\boldsymbol{v} \in \mathcal{C}(\boldsymbol{v}^*)} \sum_{n=1}^{N} \mathbb{1}\{|\mathcal{P}(\boldsymbol{X}_n^{(t)}; \boldsymbol{v})| > 0\} \left| \mathbb{1}\{y_n = (+,c)\} - \text{logit}_{+,c}^{(t)}(\boldsymbol{X}_n^{(t)}) \right| \right. \\
&\quad \left. \times \sum_{p \in \mathcal{P}(\boldsymbol{X}_n^{(t)}; \boldsymbol{v})} \mathbb{1}\{\langle \boldsymbol{w}_{+,c,r}^{(t)}, \alpha_{n,p}^{(t)} \boldsymbol{v} + \boldsymbol{\zeta}_{n,p}^{(t)} \rangle + b_{+,c,r}^{(t)} > 0\} \right) \\
&< 0.
\end{aligned}
\tag{194}
$$

Therefore, with probability at least $1 - O\left(\frac{mNP}{\text{poly}(d)}\right)$, the following holds for the relevant neurons and $\boldsymbol{v}^*$-dominated patches:

$$
\langle \Delta \boldsymbol{w}_{+,c,r}^{(T)}, \boldsymbol{x}_{n,p}^{(\tau)} \rangle + \Delta b_{+,c,r}^{(T)} < 0.
\tag{195}
$$

In conclusion, with $\tau \geq T + 1$, with probability at least $1 - O\left(\frac{mNP}{\text{poly}(d)}\right)$, for every $(+,c,r) \notin S_{+,c}^{(0)}(\boldsymbol{v}^*)$ and relevant $(n,p)$'s,

$$
\langle \boldsymbol{w}_{+,c,r}^{(T)} + \Delta \boldsymbol{w}_{+,c,r}^{(T)}, \boldsymbol{x}_{n,p}^{(\tau)} \rangle + b_{+,c,r}^{(T)} + \Delta b_{+,c,r}^{(T)} = \langle \boldsymbol{w}_{+,c,r}^{(T+1)}, \boldsymbol{x}_{n,p}^{(\tau)} \rangle + b_{+,c,r}^{(T+1)} < 0,
\tag{196}
$$

which leads to $\langle \boldsymbol{w}_{+,c,r}^{(t')}, \boldsymbol{x}_{n,p}^{(\tau)} \rangle + b_{+,c,r}^{(t')} < 0$ for all $t' \leq T+1$ with probability at least $1 - O\left(\frac{mk_+ NP(T+1)}{\text{poly}(d)}\right)$ (also by taking union bound over all the possible choices of $\boldsymbol{v}^*$ at time $T + 1$). This finishes the inductive step for point 1.

*2. (Non-activation on noise patches)*

The inductive step for this part is very similar to (and even simpler than) the inductive step of point 1, so we omit the calculations here.

*3. (Off-diagonal nonpositive growth)* By the induction hypothesis's high-probability event, we already have that, given any fine-grained class $(+, c)$, $\tau \geq T + 1$, for any feature $\boldsymbol{v}^* \in \{\boldsymbol{v}_{-,c}\}_{c=1}^{k} \cup \{\boldsymbol{v}_-\} \cup \{\boldsymbol{v}_{+,c'}\}_{c' \neq c}$ and any neuron $\boldsymbol{w}_{+,c,r}$, $\sigma\left(\langle \boldsymbol{w}_{+,c,r}^{(T)}, \boldsymbol{x}_{n,p}^{(\tau)}\rangle + b_{+,c,r}^{(T)}\right) \leq \sigma\left(\langle \boldsymbol{w}_{+,c,r}^{(0)}, \boldsymbol{x}_{n,p}^{(\tau)}\rangle + b_{+,c,r}^{(0)}\right)$. We just need to show that $\langle \Delta \boldsymbol{w}_{+,c,r}^{(t)}, \boldsymbol{x}_{n,p}^{(\tau)}\rangle + \Delta b_{+,r}^{(T)} \leq 0$ to finish the proof; the rest proceeds in a similar fashion to the induction step of point 3 in the proof of Theorem F.1.

Similar to the induction step of point 1, denoting $\mathcal{M}$ to be the set of all common and fine-grained features, the update expression of any neuron $(+, c, r)$ has to be

$$
\begin{aligned}
\Delta \boldsymbol{w}_{+,c,r}^{(T)} = &\frac{\eta}{NP} \sum_{\boldsymbol{v} \in \mathcal{M}} \sum_{n=1}^{N} \mathbb{1}\{|\mathcal{P}(\boldsymbol{X}_n^{(T)}; \boldsymbol{v})| > 0\}[\mathbb{1}\{y_n = (+,c)\} - \text{logit}_{+,c}^{(T)}(\boldsymbol{X}_n^{(T)})] \\
&\times \sum_{p \in \mathcal{P}(\boldsymbol{X}_n^{(T)}; \boldsymbol{v})} \mathbb{1}\{\langle \boldsymbol{w}_{+,c,r}^{(T)}, \alpha_{n,p}^{(T)}\boldsymbol{v} + \boldsymbol{\zeta}_{n,p}^{(T)}\rangle + b_{+,c,r}^{(T)} > 0\} \left(\alpha_{n,p}^{(T)}\boldsymbol{v} + \boldsymbol{\zeta}_{n,p}^{(T)}\right)
\end{aligned}
\tag{197}
$$

Written more explicitly,

$$
\begin{aligned}
\Delta \boldsymbol{w}_{+,c,r}^{(T)} = &\frac{\eta}{NP} \sum_{\boldsymbol{v} \in \mathcal{M} - \{\boldsymbol{v}^*\}} \sum_{n=1}^{N} \mathbb{1}\{|\mathcal{P}(\boldsymbol{X}_n^{(T)}; \boldsymbol{v})| > 0\} \mathbb{1}\{y_n = (+,c)\}[1 - \text{logit}_{+,c}^{(T)}(\boldsymbol{X}_n^{(T)})] \\
&\times \sum_{p \in \mathcal{P}(\boldsymbol{X}_n^{(T)}; \boldsymbol{v})} \mathbb{1}\{\langle \boldsymbol{w}_{+,c,r}^{(T)}, \alpha_{n,p}^{(T)}\boldsymbol{v} + \boldsymbol{\zeta}_{n,p}^{(T)}\rangle + b_{+,c,r}^{(T)} > 0\} \left(\alpha_{n,p}^{(T)}\boldsymbol{v} + \boldsymbol{\zeta}_{n,p}^{(T)}\right) \\
&- \frac{\eta}{NP} \sum_{n=1}^{N} \mathbb{1}\{y_n \neq (+,c)\} \mathbb{1}\{|\mathcal{P}(\boldsymbol{X}_n^{(T)}; \boldsymbol{v}^*)| > 0\}[\text{logit}_{+,c}^{(T)}(\boldsymbol{X}_n^{(T)})] \\
&\times \sum_{p \in \mathcal{P}(\boldsymbol{X}_n^{(T)}; \boldsymbol{v}^*)} \mathbb{1}\{\langle \boldsymbol{w}_{+,c,r}^{(T)}, \alpha_{n,p}^{(T)}\boldsymbol{v}^* + \boldsymbol{\zeta}_{n,p}^{(T)}\rangle + b_{+,c,r}^{(T)} > 0\} \left(\alpha_{n,p}^{(T)}\boldsymbol{v}^* + \boldsymbol{\zeta}_{n,p}^{(T)}\right)
\end{aligned}
\tag{198}
$$

It follows that with probability at least $1 - O\left(\frac{mNP}{\text{poly}(d)}\right)$, for relevant $n, p, r$, we have

$$
\begin{aligned}
&\langle \Delta \boldsymbol{w}_{+,c,r}^{(T)}, \alpha_{n,p}^{(\tau)}\boldsymbol{v}^* + \boldsymbol{\zeta}_{n,p}^{(\tau)}\rangle \\
&< \frac{\eta}{NP} \sum_{\boldsymbol{v} \in \mathcal{M} - \{\boldsymbol{v}^*\}} \sum_{n=1}^{N} \mathbb{1}\{|\mathcal{P}(\boldsymbol{X}_n^{(T)}; \boldsymbol{v})| > 0\} \mathbb{1}\{y_n = (+,c)\}[1 - \text{logit}_{+,c}^{(T)}(\boldsymbol{X}_n^{(T)})] \\
&\quad \times \sum_{p \in \mathcal{P}(\boldsymbol{X}_n^{(T)}; \boldsymbol{v})} \mathbb{1}\{\langle \boldsymbol{w}_{+,c,r}^{(T)}, \alpha_{n,p}^{(T)}\boldsymbol{v} + \boldsymbol{\zeta}_{n,p}^{(T)}\rangle + b_{+,c,r}^{(T)} > 0\} O\left(\frac{1}{\log^9(d)}\right)
\end{aligned}
\tag{199}
$$

Furthermore, similar to the induction step of point 1, we can estimate the bias update as follows:

$$
\begin{aligned}
&\Delta b_{+,c,r}^{(t)} \\
&\leq -\Omega\left(\frac{1}{\log^5(d)}\right) \frac{\eta}{NP} \left( \sum_{\boldsymbol{v} \in \mathcal{M}} \sum_{n=1}^{N} \mathbb{1}\{|\mathcal{P}(\boldsymbol{X}_n^{(t)}; \boldsymbol{v})| > 0\} \left| \mathbb{1}\{y_n = (+,c)\} - \text{logit}_{+,c}^{(t)}(\boldsymbol{X}_n^{(t)}) \right| \right. \\
&\quad \left. \times \sum_{p \in \mathcal{P}(\boldsymbol{X}_n^{(t)}; \boldsymbol{v})} \mathbb{1}\{\langle \boldsymbol{w}_{+,c,r}^{(t)}, \alpha_{n,p}^{(t)}\boldsymbol{v} + \boldsymbol{\zeta}_{n,p}^{(t)}\rangle + b_{+,c,r}^{(t)} > 0\} \right)
\end{aligned}
\tag{200}
$$

It follows that, indeed, $\langle \Delta \boldsymbol{w}_{+,c,r}^{(T)}, \boldsymbol{x}_{n,p}^{(\tau)}\rangle + \Delta b_{+,c,r}^{(T)} \leq 0$, which completes the induction step of point 3. $\qquad \square$

### G.3 Training

Choose an arbitrary constant $B \in [\Omega(1), \log(3/2)]$.

**Definition G.2.** Let $T_0(B) > 0$ be the first time that there exists some $\boldsymbol{X}_n^{(t)}$ and $c$ such that $F_y^{(T_0(B))}(\boldsymbol{X}_n^{(T_0(B))}) \geq B$ for any $n \in [N]$ and $y \in \{(+,c)\}_{c=1}^{k_+} \cup \{(-,c)\}_{c=1}^{k_-}$.

We write $T_0(B)$ as $T_0$ for simplicity of notation when the context is clear.

**Lemma G.2.** *With probability at least $1 - O\left(\frac{mk_+ NPT_0}{poly(d)}\right)$, the following holds for all $t \in [0, T_0)$:*

1. *(On-diagonal common-feature neuron growth) For every $c \in [k_+]$, every $(+,c,r), (+,c,r') \in S_{+,c}^{*(0)}(\boldsymbol{v}_+)$,*

$$\boldsymbol{w}_{+,c,r}^{(t)} - \boldsymbol{w}_{+,c,r}^{(0)} = \boldsymbol{w}_{+,c,r'}^{(t)} - \boldsymbol{w}_{+,c,r'}^{(0)} \tag{201}$$

   *Moreover,*

$$\Delta \boldsymbol{w}_{+,r}^{(t)} = [1/4, 2/3]\sqrt{1 \pm \iota}\left(1 \pm s^{*-1/3}\right)\eta \frac{s^*}{2k_+ P}\boldsymbol{v}_+ + \Delta \boldsymbol{\zeta}_{+,r}^{(t)} \tag{202}$$

   *where $\Delta \boldsymbol{\zeta}_{+,c,r}^{(t)} \sim \mathcal{N}(\boldsymbol{0}, \sigma_{\Delta \zeta_{+,c,r}}^{(t)2}\boldsymbol{I})$, $\sigma_{\Delta \zeta_{+,c,r}}^{(t)} = \Theta(1) \times \eta \sigma_\zeta \frac{\sqrt{s^*}}{P\sqrt{2N}}$.*
   *The bias updates satisfy*

$$\Delta b_{+,c,r}^{(t)} = -\Theta\left(\frac{\eta s^*}{k_+ P \log^5(d)}\right). \tag{203}$$

   *Furthermore, every $(+,r) \in S_+^{*(0)}(\boldsymbol{v}_+)$ activates on all the $\boldsymbol{v}_+$-dominated patches at time $t$.*

2. *(On-diagonal finegrained-feature neuron growth) For every $c \in [k_+]$ and every $(+,c,r), (+,c,r') \in S_{+,c}^{*(0)}(\boldsymbol{v}_{+,c})$,*

$$\boldsymbol{w}_{+,c,r}^{(t)} - \boldsymbol{w}_{+,c,r}^{(0)} = \boldsymbol{w}_{+,c,r'}^{(t)} - \boldsymbol{w}_{+,c,r'}^{(0)} \tag{204}$$

   *Moreover,*

$$\Delta \boldsymbol{w}_{+,c,r}^{(t)} = \left(1 \pm O\left(\frac{1}{k_+}\right)\right)\sqrt{1 \pm \iota}\left(1 \pm s^{*-1/3}\right)\eta \frac{s^*}{2k_+ P}\boldsymbol{v}_{+,c} + \Delta \boldsymbol{\zeta}_{+,r}^{(t)} \tag{205}$$

   *where $\boldsymbol{\zeta}_{+,c,r}^{(t)} \sim \mathcal{N}(\boldsymbol{0}, \sigma_{\Delta \zeta_{+,cr}}^{(t)2}\boldsymbol{I})$, and $\sigma_{\Delta \zeta_{+,r}}^{(t)} = \left(1 \pm O\left(\frac{1}{k_+}\right)\right)\left(1 \pm s^{*-1/3}\right)\eta \sigma_\zeta \frac{\sqrt{s^*}}{P\sqrt{2Nk_+}}$.*
   *The bias updates satisfy*

$$\Delta b_{+,c,r}^{(t)} = -\Theta\left(\frac{\eta s^*}{k_+ P \log^5(d)}\right). \tag{206}$$

   *Furthermore, every $(+,c,r) \in S_{+,c}^{*(0)}(\boldsymbol{v}_{+,c})$ activates on all the $\boldsymbol{v}_+$-dominated patches at time $t$.*

3. *The above results also hold with the "+" and "−" class signs flipped.*

*Proof.* The proof of this theorem proceeds in a similar fashion to Theorem D.1, with some variations for the common-feature neurons.

We shall prove the statements in this theorem via induction. We focus on the +-class neurons; −-class neurons' proofs are done in the same fashion.

First of all, relying on the (high-probability) event of Theorem G.1, we know that we can simplify the update expressions for the neurons in $S_{+,c}^{*(0)}(\boldsymbol{v}_{+,c})$ to the form

$$\Delta \boldsymbol{w}_{+,c,r}^{(t)} = \frac{\eta}{NP}\sum_{n=1}^{N}\mathbb{1}\{y_n = (+,c)\}[1 - \text{logit}_{+,c}^{(t)}(\boldsymbol{X}_n^{(t)})]$$
$$\times \sum_{p \in \mathcal{P}(\boldsymbol{X}_n^{(t)};\boldsymbol{v}_{+,c})}\mathbb{1}\{\langle \boldsymbol{w}_{+,c,r}^{(t)}, \alpha_{n,p}^{(t)}\boldsymbol{v}_{+,c} + \boldsymbol{\zeta}_{n,p}^{(t)}\rangle + b_{+,c,r}^{(t)} > 0\}\left(\alpha_{n,p}^{(t)}\boldsymbol{v}_{+,c} + \boldsymbol{\zeta}_{n,p}^{(t)}\right), \tag{207}$$

and for the neurons in $S_{+,c}^{*(0)}(\boldsymbol{v}_+)$, the updates take the form

$$
\begin{aligned}
&\Delta \boldsymbol{w}_{+,c,r}^{(t)} \\
&= \frac{\eta}{NP} \sum_{n=1}^{N} \left\{ \mathbb{1}\{y_n = (+,c)\}[1 - \mathrm{logit}_{+,c}^{(t)}(\boldsymbol{X}_n^{(t)})] + \sum_{c' \in [k_+]-\{c\}} \mathbb{1}\{y_n = (+,c')\}[-\mathrm{logit}_{+,c}^{(t)}(\boldsymbol{X}_n^{(t)})] \right\} \\
&\quad \times \sum_{p \in \mathcal{P}(\boldsymbol{X}_n^{(t)}; \boldsymbol{v}_+)} \mathbb{1}\{\langle \boldsymbol{w}_{+,c,r}^{(t)}, \alpha_{n,p}^{(t)} \boldsymbol{v}_+ + \boldsymbol{\zeta}_{n,p}^{(t)}\rangle + b_{+,c,r}^{(t)} > 0\} \left(\alpha_{n,p}^{(t)} \boldsymbol{v}_+ + \boldsymbol{\zeta}_{n,p}^{(t)}\right).
\end{aligned}
\tag{208}
$$

By definition of $T_0$ and the fact that $B \leq \log(3/2)$, for any $n \in [N]$ and $t < T_0$, we can write down a simple upper bound of $\mathrm{logit}_{+,c}^{(t)}(\boldsymbol{X}_n^{(t)})$:

$$
\begin{aligned}
\mathrm{logit}_{+,c}^{(t)}(\boldsymbol{X}_n^{(t)}) &= \frac{\exp(F_{+,c}(\boldsymbol{X}_n^{(t)}))}{\sum_{c'=1}^{k_+} \exp(F_{+,c'}(\boldsymbol{X}_n^{(t)})) + \sum_{c'=1}^{k_-} \exp(F_{-,c'}(\boldsymbol{X}_n^{(t)}))} \\
&\leq \frac{\frac{3}{2}}{2k_+} = \frac{3}{4k_+},
\end{aligned}
\tag{209}
$$

and we can lower bound it as follows

$$
\mathrm{logit}_{+,c}^{(t)}(\boldsymbol{X}_n^{(t)}) \geq \frac{1}{2k_+ \times \frac{3}{2}} = \frac{1}{3k_+},
\tag{210}
$$

The inductive proof for the fine-grained neurons $S_{+,c}^{*(0)}(\boldsymbol{v}_{+,c})$ is almost identical to that in the proof of Theorem D.1. The only notable difference here is that $[1 - \mathrm{logit}_{+,c}^{(t)}(\boldsymbol{X}_n^{(t)})]$ has the estimate $\left(1 \pm O\left(\frac{1}{k_+}\right)\right)$.

The inductive proof of the common-feature neurons $S_{+,c}^{*(0)}(\boldsymbol{v}_+)$ requires more care as its update expression equation 208 is qualitatively different from the coarse-grained training case in Theorem D.1, so we present the full proof here.

**Base case,** $t = 0$.

With probability at least $1 - O\left(\frac{mNP}{\mathrm{poly}(d)}\right)$, for every $c \in [k_+]$ and every $(+,c,r) \in S_{+,c}^{*(0)}(\boldsymbol{v}_+)$,

$$
\begin{aligned}
&\langle \boldsymbol{w}_{+,c,r}^{(0)}, \alpha_{n,p}^{(0)} \boldsymbol{v}_+ + \boldsymbol{\zeta}_{n,p}^{(0)}\rangle + b_{+,c,r}^{(0)} \\
&\geq \sigma_0 \left( \sqrt{(1-\iota)(2+2c_0)(\log(d) + 1/\log^5(d))} - \sqrt{(2+2c_0)\log(d)} - O\left(\frac{1}{\log^9(d)}\right) \right) \\
&= \sigma_0 \left( \frac{(1-\iota)(2+2c_0)(\log(d) + 1/\log^5(d)) - (2+2c_0)\log(d)}{\sqrt{(1-\iota)(2+2c_0)(\log(d) + 1/\log^5(d))} + \sqrt{(2+2c_0)\log(d)}} - O\left(\frac{1}{\log^9(d)}\right) \right) \\
&= \sigma_0 \left( \frac{(2+2c_0)(-\iota\log(d) + (1-\iota)/\log^5(d))}{\sqrt{(1-\iota)(2+2c_0)(\log(d) + 1/\log^5(d))} + \sqrt{(2+2c_0)\log(d)}} - O\left(\frac{1}{\log^9(d)}\right) \right) \\
&> 0.
\end{aligned}
\tag{211}
$$

This means all the $\boldsymbol{v}_+$-singleton neurons will be updated on all the $\boldsymbol{v}_+$-dominated patches at time $t = 0$. Therefore, we can write update expression equation 208 as follows

$$
\begin{aligned}
&\Delta\boldsymbol{w}_{+,c,r}^{(0)} \\
&= \frac{\eta}{NP}\sum_{n=1}^{N}\left\{\mathbb{1}\{y_n = (+,c)\}[1 - \mathrm{logit}_{+,c}^{(0)}(\boldsymbol{X}_n^{(0)})] + \sum_{c'\in[k_+]-\{c\}}\mathbb{1}\{y_n = (+,c')\}[-\mathrm{logit}_{+,c}^{(0)}(\boldsymbol{X}_n^{(0)})]\right\} \\
&\quad \times \sum_{p\in\mathcal{P}(\boldsymbol{X}_n^{(0)};\boldsymbol{v}_+)}\left(\alpha_{n,p}^{(0)}\boldsymbol{v}_+ + \boldsymbol{\zeta}_{n,p}^{(0)}\right).
\end{aligned}
\tag{212}
$$

By concentration of the binomial random variable, we know that with probability at least $1 - e^{-\Omega(\log^2(d))}$, for all $n$,

$$
\left|\mathcal{P}(\boldsymbol{X}_n^{(0)};\boldsymbol{v}_+)\right| = \left(1 \pm s^{*-1/3}\right)s^*.
\tag{213}
$$

Now, with the estimates we derived for $\mathrm{logit}_{+,c}^{(t)}(\boldsymbol{X}_n^{(t)})$ at the beginning of the proof and the independence of all the noise vectors $\boldsymbol{\zeta}_{n,p}^{(0)}$'s, we arrive at

$$
\Delta\boldsymbol{w}_{+,r}^{(0)} = [1/4, 2/3]\sqrt{1 \pm \iota}\left(1 \pm s^{*-1/3}\right)\eta\frac{s^*}{2k_+P}\boldsymbol{v}_+ + \Delta\boldsymbol{\zeta}_{+,r}^{(0)}
\tag{214}
$$

where $\sigma_{\Delta\zeta_{+,c,r}}^{(0)} = \Theta(1) \times \eta\sigma_\zeta\frac{\sqrt{s^*}}{P\sqrt{2N}}$.

Additionally, a byproduct of the above proof steps is that all the $S_{+,c}^{*(0)}(\boldsymbol{v}_+)$ neurons indeed activate on all the $\boldsymbol{v}_+$-dominated patches at $t = 0$ with high probability.

Now we examine the bias update. We first estimate $\left\|\Delta\boldsymbol{w}_{+,c,r}^{(0)}\right\|_2$. With probability at least $1 - O\left(\frac{m}{\mathrm{poly}(d)}\right)$ the following upper bound holds for all neurons in $S_{+,c}^{*(0)}(\boldsymbol{v}_+)$:

$$
\begin{aligned}
\left\|\Delta\boldsymbol{w}_{+,c,r}^{(0)}\right\|_2 &\leq O\left(\eta\frac{s^*}{k_+P}\right)\|\boldsymbol{v}_+\|_2 + \left\|\Delta\boldsymbol{\zeta}_{+,r}^{(0)}\right\|_2 \\
&\leq O\left(\eta\frac{s^*}{k_+P}\right) + O\left(\eta\sigma_\zeta\frac{\sqrt{s^*}}{P\sqrt{N}}\sqrt{d}\right) \\
&\leq O\left(\eta\frac{s^*}{k_+P}\right),
\end{aligned}
\tag{215}
$$

and the following lower bound holds (via the reverse triangle inequality):

$$
\begin{aligned}
\left\|\Delta\boldsymbol{w}_{+,c,r}^{(0)}\right\|_2 &\geq \Omega\left(\eta\frac{s^*}{k_+P}\right)\|\boldsymbol{v}_+\|_2 - \left\|\Delta\boldsymbol{\zeta}_{+,r}^{(0)}\right\|_2 \\
&\geq \Omega\left(\eta\frac{s^*}{k_+P}\right) - O\left(\eta\sigma_\zeta\frac{\sqrt{s^*}}{P\sqrt{N}}\sqrt{d}\right) \\
&\geq \Omega\left(\eta\frac{s^*}{k_+P}\right),
\end{aligned}
\tag{216}
$$

It follows that $\left\|\Delta\boldsymbol{w}_{+,c,r}^{(0)}\right\|_2 = \Theta\left(\eta\frac{s^*}{k_+P}\right)$, which means

$$
\Delta b_{+,c,r}^{(0)} = -\frac{\left\|\Delta\boldsymbol{w}_{+,c,r}^{(0)}\right\|_2}{\log^5(d)} = -\Theta\left(\frac{\eta s^*}{k_+P\log^5(d)}\right).
\tag{217}
$$

This completes the proof of the base case.

**Induction step**. Assume statements for time $[0, t]$, prove for $t + 1$.

First, by the induction hypothesis, we know that neurons in $S_{+,c}^{*(0)}(\boldsymbol{v}_+)$ must activate on all the $\boldsymbol{v}_+$-dominated patches at time $t$. Therefore, we can write the update expression equation 208 as follows:

$$
\begin{aligned}
&\Delta \boldsymbol{w}_{+,c,r}^{(t)} \\
&= \frac{\eta}{NP} \sum_{n=1}^{N} \left\{ \mathbb{1}\{y_n = (+, c)\}[1 - \mathrm{logit}_{+,c}^{(t)}(\boldsymbol{X}_n^{(t)})] + \sum_{c' \in [k_+] - \{c\}} \mathbb{1}\{y_n = (+, c')\}[-\mathrm{logit}_{+,c}^{(t)}(\boldsymbol{X}_n^{(t)})] \right\} \\
&\quad \times \sum_{p \in \mathcal{P}(\boldsymbol{X}_n^{(t)}; \boldsymbol{v}_+)} \left( \alpha_{n,p}^{(t)} \boldsymbol{v}_+ + \boldsymbol{\zeta}_{n,p}^{(t)} \right).
\end{aligned}
\tag{218}
$$

Following the same argument as in the base case, we have that with probability at least $1 - O\left(\frac{mNP}{\mathrm{poly}(d)}\right)$,

$$
\Delta \boldsymbol{w}_{+,c,r}^{(t)} = [1/4, 2/3]\sqrt{1 \pm \iota}\left(1 \pm s^{*-1/3}\right)\eta \frac{s^*}{2k_+ P} \boldsymbol{v}_+ + \Delta \boldsymbol{\zeta}_{+,c,r}^{(t)},
\tag{219}
$$

and $\sigma_{\Delta\zeta_{+,c,r}}^{(t)} = \Theta(1) \times \eta \sigma_\zeta \frac{\sqrt{s^*}}{P\sqrt{2N}}$.

Now we need to show that $\boldsymbol{w}_{+,c,r}^{(t+1)}$ indeed activate on all the $\boldsymbol{v}_+$-dominated patches at time $t + 1$ with high probability.

So far, we know that for $\tau \in [0, t + 1]$,

$$
\Delta \boldsymbol{w}_{+,c,r}^{(\tau)} = [1/4, 2/3]\sqrt{1 \pm \iota}\left(1 \pm s^{*-1/3}\right)\eta \frac{s^*}{2k_+ P} \boldsymbol{v}_+ + \Delta \boldsymbol{\zeta}_{+,c,r}^{(\tau)},
\tag{220}
$$

and $\sigma_{\Delta\zeta_{+,c,r}}^{(\tau)} = \Theta(1) \times \eta \sigma_\zeta \frac{\sqrt{s^*}}{P\sqrt{2N}}$. It follows that

$$
\boldsymbol{w}_{+,r}^{(t+1)} = \boldsymbol{w}_{+,c,r}^{(0)} + (t+1)[1/4, 2/3]\sqrt{1 \pm \iota}\left(1 \pm s^{*-1/3}\right)\eta \frac{s^*}{2k_+ P} \boldsymbol{v}_+ + \boldsymbol{\zeta}_{+,c,r}^{(t+1)},
\tag{221}
$$

where $\sigma_{\zeta_{+,c,r}}^{(t+1)} = \Theta(1) \times \sqrt{t+1}\eta \sigma_\zeta \frac{\sqrt{s^*}}{P\sqrt{2N}}$.

The following holds with probability at least $1 - O\left(\frac{mNP}{\mathrm{poly}(d)}\right)$ over all the $\boldsymbol{v}_+$-dominated patches $\boldsymbol{x}_{n,p}^{(t+1)} = \alpha_{n,p}^{(t+1)} \boldsymbol{v}_+ + \boldsymbol{\zeta}_{n,p}^{(t+1)}$ (which are independent of $\boldsymbol{w}_{+,r}^{(t+1)}$) and the $\boldsymbol{v}_+$-singleton neurons:

$$
\begin{aligned}
&\langle \boldsymbol{w}_{+,c,r}^{(t+1)}, \alpha_{n,p}^{(t+1)} \boldsymbol{v}_+ + \boldsymbol{\zeta}_{n,p}^{(t+1)} \rangle \\
&= \langle \boldsymbol{w}_{+,c,r}^{(0)} \alpha_{n,p}^{(t+1)} \boldsymbol{v}_+ + \boldsymbol{\zeta}_{n,p}^{(t+1)} \rangle + (t+1)[1/4, 2/3](1 \pm \iota)\left(1 \pm s^{*-1/3}\right)\left(1 \pm O\left(\frac{1}{\log^9(d)}\right)\right)\eta \frac{s^*}{2k_+ P} \\
&\quad + \langle \boldsymbol{\zeta}_{+,c,r}^{(t+1)}, \alpha_{n,p}^{(t+1)} \boldsymbol{v}_+ + \boldsymbol{\zeta}_{n,p}^{(t+1)} \rangle
\end{aligned}
\tag{222}
$$

Note that with probability at least $1 - O\left(\frac{1}{\mathrm{poly}(d)}\right)$,

$$
\langle \boldsymbol{\zeta}_{+,c,r}^{(t+1)}, \alpha_{n,p}^{(t+1)} \boldsymbol{v}_+ \rangle \leq O(1) \times \sqrt{T}\eta \sigma_\zeta \frac{\sqrt{s^*}}{P\sqrt{2N}}\sqrt{d \log(d)},
\tag{223}
$$

and since $\sqrt{t+1} \leq t+1$, $\sqrt{s^*} < s^*$, $\sigma_\zeta \sqrt{d \log(d)} < \frac{1}{\log^9(d)}$, and $N > dk_+$, we know that

$$
\langle \boldsymbol{\zeta}_{+,c,r}^{(t+1)}, \alpha_{n,p}^{(t+1)} \boldsymbol{v}_+ \rangle \leq O\left(\frac{1}{d}\right) \times (t+1)\eta \frac{s^*}{2k_+ P}.
\tag{224}
$$

Similarly, with probability at least $1 - O\left(\frac{1}{\text{poly}(d)}\right)$,

$$\langle \boldsymbol{\zeta}_{+,c,r}^{(t+1)}, \alpha_{n,p}^{(t+1)} \boldsymbol{v}_+ + \boldsymbol{\zeta}_{n,p}^{(t+1)} \rangle \leq O(1) \times \sqrt{T} \eta \sigma_\zeta^2 \frac{\sqrt{s^*}}{P\sqrt{2N}} \sqrt{d \log(d)} \leq O\left(\frac{1}{d}\right) \times (t+1) \eta \frac{s^*}{2k_+ P}. \tag{225}$$

It follows that with probability at least $1 - O\left(\frac{mNP}{\text{poly}(d)}\right)$,

$$\begin{aligned}
&\langle \boldsymbol{w}_{+,c,r}^{(t+1)}, \alpha_{n,p}^{(t+1)} \boldsymbol{v}_+ + \boldsymbol{\zeta}_{n,p}^{(t+1)} \rangle \\
&\geq \langle \boldsymbol{w}_{+,c,r}^{(0)}, \alpha_{n,p}^{(t+1)} \boldsymbol{v}_+ + \boldsymbol{\zeta}_{n,p}^{(t+1)} \rangle + \frac{1}{4}(t+1)(1-\iota)\left(1 - s^{*-1/3}\right)\left(1 - O\left(\frac{1}{\log^9(d)}\right)\right) \eta \frac{s^*}{2k_+ P}.
\end{aligned} \tag{226}$$

Next, let us estimate the bias updates for $\tau \in [0, t+1]$.

Estimating $\Delta b_{+,c,r}^{(t)}$ follows an almost identical argument as in the base case (with the only main difference being relying on Theorem G.1 for non-activation on non-$\boldsymbol{v}_+$-dominated patches), so we skip its calculations.

Therefore, $b_{+,c,r}^{(t+1)} = b_{+,c,r}^{(0)} + -\Theta\left(\frac{\eta s^*(t+1)}{k_+ P \log^5(d)}\right)$. This means

$$\begin{aligned}
&\langle \boldsymbol{w}_{+,c,r}^{(t+1)}, \alpha_{n,p}^{(t+1)} \boldsymbol{v}_+ + \boldsymbol{\zeta}_{n,p}^{(t+1)} \rangle + b_{+,c,r}^{(t+1)} \\
&\geq \langle \boldsymbol{w}_{+,c,r}^{(0)}, \alpha_{n,p}^{(t+1)} \boldsymbol{v}_+ + \boldsymbol{\zeta}_{n,p}^{(t+1)} \rangle + b_{+,c,r}^{(0)} \\
&\quad + \frac{1}{4}(t+1)(1-\iota)\left(1 - s^{*-1/3}\right)\left(1 - O\left(\frac{1}{\log^9(d)}\right)\right) \eta \frac{s^*}{2k_+ P} - O\left(\frac{\eta s^*(t+1)}{k_+ P \log^5(d)}\right) \\
&> 0.
\end{aligned} \tag{227}$$

This completes the inductive step. $\qquad\square$

**Corollary G.2.1.** *At time $t = T_0$, $\frac{\eta s^*}{k_+ P} \times s^* \left|S_{+,c}^{*(0)}(\boldsymbol{v}_+)\right|, \frac{\eta s^*}{k_+ P} \times s^* \left|S_{+,c}^{*(0)}(\boldsymbol{v}_{+,c})\right| = \Theta(1)$.*

*Proof.* Directly follows from Lemma G.2 and Theorem G.1. $\qquad\square$

### G.4 Model error after training

In this subsection, we show the model's error after fine-grained training. We also discuss that finetuning the model further increases its feature extractor's response to the true features, so it is even more robust/generalizing in downstream classification tasks.

**Theorem G.3.** *Define $\widehat{F}_+(\boldsymbol{X}) = \max_{c \in [k_+]} F_{+,c}(\boldsymbol{X})$, $\widehat{F}_-(\boldsymbol{X}) = \max_{c \in [k_-]} F_{-,c}(\boldsymbol{X})$.*

*With probability at least $1 - O\left(\frac{mk_+^2 NPT_0}{poly(d)}\right)$, the following events take place:*

1. *(Fine-grained easy & hard sample test accuracies are nearly perfect) Given an easy or hard fine-grained test sample $(\boldsymbol{X}, y)$ where $y \in \{(+,c)\}_{c=1}^{k_+} \cup \{(-,c)\}_{c=1}^{k_-}$, $\mathbb{P}\left[F_y^{(T_0)}(\boldsymbol{X}) \leq \max_{y' \neq y} F_{y'}^{(T_0)}(\boldsymbol{X})\right] \leq o(1)$.*

2. *(Coarse-grained easy & hard sample test accuracy are nearly perfect) Given an easy or hard coarse-grained test sample $(\boldsymbol{X}, y)$ where $y \in \{+1, -1\}$, $\mathbb{P}\left[\widehat{F}_y^{(T_0)}(\boldsymbol{X}) \leq \widehat{F}_{y'}^{(T_0)}(\boldsymbol{X})\right] \leq o(1)$.*

*Proof.* **Probability of mistake on easy samples**.

Without loss of generality, assume $\boldsymbol{X}$ is a $(+, c)$-class easy sample.

Conditioning on the events of Theorem G.1 and Lemma G.2, we know that for all $c' \in [k_-]$,

$$F_{-,c'}^{(T_0)} \leq O(m_{+,c'} \sigma_0 \sqrt{\log(d)}) \leq o(1), \tag{228}$$

and for all $c' \in [k_+] - \{c\}$,

$$
\begin{aligned}
F_{+,c'}^{(T_0)} &\leq \sum_{p \in \mathcal{P}(\boldsymbol{X};\boldsymbol{v}_+)} \sum_{(+,r) \in S_{+,c'}^{(0)}(\boldsymbol{v}_+)} \sigma\left(\langle \boldsymbol{w}_{+,r}^{(T_0)}, \alpha_{n,p}\boldsymbol{v}_+ + \boldsymbol{\zeta}_{n,p}\rangle + b_{+,c',r}^{(T_0)}\right) + O(m_{+,c'}\sigma_0\sqrt{\log(d)}) \\
&\leq s^*\left|S_{+,c'}^{(0)}(\boldsymbol{v}_+)\right|\frac{2}{3}(1+\iota)\left(1 + s^{*-1/3}\right)\left(1 + \left(\frac{1}{\log^9(d)}\right)\right)\eta T_0\frac{s^*}{2k_+P}
\end{aligned}
\tag{229}
$$

moreover,

$$
\begin{aligned}
F_{+,c}^{(T_0)} &\geq \sum_{p \in \mathcal{P}(\boldsymbol{X};\boldsymbol{v}_+)} \sum_{(+,r) \in S_{+,c}^{*(0)}(\boldsymbol{v}_+)} \sigma\left(\langle \boldsymbol{w}_{+,c,r}^{(T_0)}, \alpha_{n,p}\boldsymbol{v}_+ + \boldsymbol{\zeta}_{n,p}\rangle + b_{+,c,r}^{(T_0)}\right) \\
&\quad + \sum_{p \in \mathcal{P}(\boldsymbol{X};\boldsymbol{v}_{+,c})} \sum_{(+,r) \in S_{+,c}^{*(0)}(\boldsymbol{v}_{+,c})} \sigma\left(\langle \boldsymbol{w}_{+,c,r}^{(T_0)}, \alpha_{n,p}\boldsymbol{v}_{+,c} + \boldsymbol{\zeta}_{n,p}\rangle + b_{+,c,r}^{(T_0)}\right) \\
&\geq s^*\left|S_{+,c}^{*(0)}(\boldsymbol{v}_+)\right|\frac{1}{4}(1-\iota)\left(1 - s^{*-1/3}\right)\left(1 - \left(\frac{1}{\log^5(d)}\right)\right)\eta T_0\frac{s^*}{2k_+P} \\
&\quad + s^*\left|S_{+,c}^{*(0)}(\boldsymbol{v}_{+,c})\right|\left(1 - O\left(\frac{1}{k_+}\right)\right)(1-\iota)\left(1 - s^{*-1/3}\right)\left(1 - \left(\frac{1}{\log^5(d)}\right)\right)\eta T_0\frac{s^*}{2k_+P}
\end{aligned}
\tag{230}
$$

Relying on Proposition 2, we know $\left|S_{+,c'}^{(0)}(\boldsymbol{v}_+)\right| = \left(1 \pm \left(\frac{1}{\log^5(d)}\right)\right)\left|S_{+,c}^{*(0)}(\boldsymbol{v}_+)\right|$ and $\left|S_{+,c}^{*(0)}(\boldsymbol{v}_{+,c})\right| = \left(1 \pm \left(\frac{1}{\log^5(d)}\right)\right)\left|S_{+,c}^{*(0)}(\boldsymbol{v}_+)\right|$, therefore $F_{+,c}^{(T_0)}(\boldsymbol{X}) > \max_{c' \neq c} F_{+,c'}^{(T_0)}(\boldsymbol{X})$ has to be true. With Corollary G.2.1, we also have $F_{+,c}^{(T_0)}(\boldsymbol{X}) \geq \Omega(1) > o(1) \geq \max_{c' \in [k_-]} F_{-,c'}^{(T_0)}(\boldsymbol{X})$. It follows that the probability of mistake on an easy test sample is indeed at most $o(1)$.

**Probability of mistake on hard samples.** Without loss of generality, assume $\boldsymbol{X}$ is a $(+,c)$-class hard sample.

By Theorem G.1 (and its proof) and Lemma G.2, we know that for any $c' \in [k_+]$, the neurons $\boldsymbol{w}_{+,c',r}$ can only possibly receive update on $\boldsymbol{v}$-dominated patches for $\boldsymbol{v} \in \mathcal{U}_{+,c',r}^{(0)}$, and the updates to the neurons take the feature-plus-Gaussian-noise form of $\sum_{\boldsymbol{v}' \in \mathcal{U}_{+,c',r}^{(0)}} c(\boldsymbol{v}')\boldsymbol{v}' + \Delta\boldsymbol{\zeta}_{+,c',r}^{(t)}$, with $c(\boldsymbol{v}') \leq \sqrt{1+\iota}\left(1 + s^{*-1/3}\right)\eta\frac{s^*}{2k_+P}$ if $\boldsymbol{v}'$ is a fine-grained feature, or $c(\boldsymbol{v}') \leq \frac{2}{3}\sqrt{1+\iota}\left(1 + s^{*-1/3}\right)\eta\frac{s^*}{2k_+P}$ if $\boldsymbol{v}' = \boldsymbol{v}_+$ (because the $\boldsymbol{v}'$ component of a $\boldsymbol{v}'$-singleton neuron's update is already the maximum possible). Moreover, $\sigma_{\Delta\zeta_{+,c',r}}^{(t)} \leq O\left(\eta\sigma_\zeta\frac{\sqrt{s^*}}{P\sqrt{2N}}\right)$.

Relying on Theorem G.1, Lemma G.2, Corollary G.2.1 and previous observations, we have

$$
\begin{aligned}
F_{+,c}^{(T_0)}(\boldsymbol{X}) &\geq \sum_{p \in \mathcal{P}(\boldsymbol{X};\boldsymbol{v}_{+,c})} \sum_{(+,c,r) \in S_{+,c}^{*(0)}(\boldsymbol{v}_{+,c})} \sigma\left(\langle \boldsymbol{w}_{+,c,r}^{(T_0)}, \alpha_{n,p}\boldsymbol{v}_{+,c} + \boldsymbol{\zeta}_{n,p}\rangle + b_{+,c,r}^{(T_0)}\right) \\
&\geq s^*\left|S_{+,c}^{*(0)}(\boldsymbol{v}_{+,c})\right|\left(1 - O\left(\frac{1}{k_+}\right)\right)(1-\iota)\left(1 - s^{*-1/3}\right)\left(1 - O\left(\frac{1}{\log^5(d)}\right)\right)\eta T_0\frac{s^*}{2k_+P} \\
&\geq \Omega(1),
\end{aligned}
\tag{231}
$$

and for $c' \neq c$,

$$
\begin{aligned}
F_{+,c'}^{(T_0)}(\boldsymbol{X}) \leq & \sum_{r=1}^{m_{+,c'}} \sigma\left(\langle \boldsymbol{w}_{+,c',r}^{(T_0)}, \boldsymbol{\zeta}^* \rangle + b_{+,c',r}^{(T_0)}\right) \\
& + \sum_{p \in \mathcal{P}(\boldsymbol{X}; \boldsymbol{v}_{+,c})} \sum_{(+,c',r) \in S_{+,c'}^{(0)}(\boldsymbol{v}_{+,c})} \sigma\left(\langle \boldsymbol{w}_{+,c',r}^{(T_0)}, \alpha_{n,p}\boldsymbol{v}_{+,c} + \boldsymbol{\zeta}_{n,p} \rangle + b_{+,c',r}^{(T_0)}\right) \\
& + \sum_{p \in \mathcal{P}(\boldsymbol{X}; \boldsymbol{v}_-)} \sum_{(+,c',r) \in S_{+,c'}^{(0)}(\boldsymbol{v}_-)} \sigma\left(\langle \boldsymbol{w}_{+,c',r}^{(T_0)}, \alpha_{n,p}^{\dagger}\boldsymbol{v}_- + \boldsymbol{\zeta}_{n,p} \rangle + b_{+,c',r}^{(T_0)}\right) \\
\leq & O(1) \times \left( \sum_{(+,c',r) \in \mathcal{U}_{+,c',r}^{(0)}} \langle \sum_{\tau=0}^{T_0-1} \Delta\boldsymbol{w}_{+,c,'r}^{(\tau)}, \boldsymbol{\zeta}^* \rangle + \sum_{r \in [m_{+,c'}]} \langle \boldsymbol{w}_{+,c,'r}^{(0)}, \boldsymbol{\zeta}^* \rangle \right) \\
& + \sum_{p \in \mathcal{P}(\boldsymbol{X}; \boldsymbol{v}_{+,c})} \sum_{(+,c',r) \in S_{+,c'}^{(0)}(\boldsymbol{v}_{+,c})} \sigma\left(\langle \boldsymbol{w}_{+,c',r}^{(0)}, \alpha_{n,p}\boldsymbol{v}_{+,c} + \boldsymbol{\zeta}_{n,p} \rangle + b_{+,c',r}^{(0)}\right) \\
& + \sum_{p \in \mathcal{P}(\boldsymbol{X}; \boldsymbol{v}_-)} \sum_{(+,c',r) \in S_{+,c'}^{(0)}(\boldsymbol{v}_-)} \sigma\left(\langle \boldsymbol{w}_{+,c',r}^{(0)}, \alpha_{n,p}^{\dagger}\boldsymbol{v}_- + \boldsymbol{\zeta}_{n,p} \rangle + b_{+,c',r}^{(0)}\right) \\
\leq & O\left(\frac{1}{\mathrm{polylog}(d)}\right).
\end{aligned}
\tag{232}
$$

Moreover, for any $c' \in [k_-]$, similar to before,

$$
\begin{aligned}
F_{-,c'}^{(T_0)}(\boldsymbol{X}) \leq & \sum_{r=1}^{m_{-,c'}} \sigma\left(\langle \boldsymbol{w}_{-,c',r}^{(T_0)}, \boldsymbol{\zeta}^* \rangle + b_{-,c',r}^{(T_0)}\right) \\
& + \sum_{p \in \mathcal{P}(\boldsymbol{X}; \boldsymbol{v}_{+,c})} \sum_{(-,c',r) \in S_{-,c'}^{(0)}(\boldsymbol{v}_{+,c})} \sigma\left(\langle \boldsymbol{w}_{-,c',r}^{(T_0)}, \alpha_{n,p}\boldsymbol{v}_{+,c} + \boldsymbol{\zeta}_{n,p} \rangle + b_{-,c',r}^{(T_0)}\right) \\
& + \sum_{p \in \mathcal{P}(\boldsymbol{X}; \boldsymbol{v}_-)} \sum_{(-,c',r) \in S_{-,c'}^{(0)}(\boldsymbol{v}_-)} \sigma\left(\langle \boldsymbol{w}_{-,c',r}^{(T_0)}, \alpha_{n,p}^{\dagger}\boldsymbol{v}_- + \boldsymbol{\zeta}_{n,p} \rangle + b_{-,c',r}^{(T_0)}\right) \\
\leq & O(1) \times \left( \sum_{(-,c',r) \in \mathcal{U}_{-,c',r}^{(0)}} \langle \boldsymbol{w}_{-,c,'r}^{(T_0)}, \boldsymbol{\zeta}^* \rangle + \sum_{r \in [m_{-,c'}]} \langle \boldsymbol{w}_{-,c,'r}^{(0)}, \boldsymbol{\zeta}^* \rangle \right) \\
& + \sum_{p \in \mathcal{P}(\boldsymbol{X}; \boldsymbol{v}_{+,c})} \sum_{(-,c',r) \in S_{-,c'}^{(0)}(\boldsymbol{v}_{+,c})} \sigma\left(\langle \boldsymbol{w}_{-,c',r}^{(0)}, \alpha_{n,p}\boldsymbol{v}_{+,c} + \boldsymbol{\zeta}_{n,p} \rangle + b_{-,c',r}^{(0)}\right) \\
& + O(1) \times s^{\dagger} \left| S_{-,c'}^{(0)}(\boldsymbol{v}_-) \right| \times \left( \iota_{upper}^{\dagger} + O(\sigma_0 \log(d)) \right) \\
\leq & O\left(\frac{1}{\mathrm{polylog}(d)}\right) + O\left(\sigma_0 \sqrt{\log(d)}\right) + O\left(\frac{1}{\log(d)}\right) \\
\leq & o(1).
\end{aligned}
\tag{233}
$$

Therefore, $F_{+,c}^{(T_0)}(\boldsymbol{X}) > \max_{y \neq (+,c)} F_y^{(T_0)}(\boldsymbol{X})$, which means $\widehat{F}_+^{(T_0)}(\boldsymbol{X}) > \widehat{F}_-^{(T_0)}(\boldsymbol{X})$ indeed. $\qquad\square$

*Remark.* First of all, note that the feature extractor, after fine-grained training, is already well-performing, as it responds strongly ($\Omega(1)$ strength) to the true features, and very weakly ($o(1)$ strength) to any off-diagonal features and noise. In other words, we stop training when the margin is at least $\Omega(1)$, i.e. when we have $F_{y_n^{(T)}}^{(T)}(X_n^{(T)}) - \max_{y \neq y_n^{(T)}} F_y^{(T)}(X_n^{(T)}) \geq \Omega(1)$ for all $n$ at some $T \leq \mathrm{poly}(d)$, and with high probability, we just

need $T_0$ time to reach it. This can already help us explain the linear-probing result we saw on ImageNet21k in Appendix A.2, since linear probing does not alter the the feature extractor after fine-grained pretraining (on ImageNet21k), it only retrains a new linear classifier on top of the feature extractor for classifying on the target ImageNet1k dataset.

At a high level, *finetuning* $\widehat{F}$ can only further enhance the feature extractor's response to the features, therefore making the model even more robust for challenging downstream classification problems; it will not degrade the feature extractor's response to any true feature. A rigorous proof of this statement is almost a repetition of the proofs for fine-grained training, so we do not repeat them here. Intuitively speaking, we just need to note that the properties stated in Theorem G.1 will continue to hold during finetuning (as long as we stay in polynomial time), and with similar argument to those in the proof of Lemma G.2, we note that the neurons responsible for detecting fine-grained features, i.e. the $S_{+,c}^{*(0)}(\boldsymbol{v}_{+,c})$, will continue to only receive (positive) updates on the $\boldsymbol{v}_{+,c}$-dominated patches of the following form:

$$
\begin{aligned}
\Delta \boldsymbol{w}_{+,c,r}^{(t)} ={}& \frac{\eta}{NP} \sum_{n=1}^{N} \mathbb{1}\{y_n = (+,c)\}[1 - \mathrm{logit}_+^{(t)}(\boldsymbol{X}_n^{(t)})] \\
& \times \sum_{p \in \mathcal{P}(\boldsymbol{X}_n^{(t)}; \boldsymbol{v}_{+,c})} \mathbb{1}\{\langle \boldsymbol{w}_{+,c,r}^{(t)}, \alpha_{n,p}^{(t)} \boldsymbol{v}_{+,c} + \boldsymbol{\zeta}_{n,p}^{(t)} \rangle + b_{+,c,r}^{(t)} > 0\} \left( \alpha_{n,p}^{(t)} \boldsymbol{v}_{+,c} + \boldsymbol{\zeta}_{n,p}^{(t)} \right),
\end{aligned}
\tag{234}
$$

and similar update expression can be stated for the $S_{+,c}^{*(0)}(\boldsymbol{v}_+)$ neurons:

$$
\begin{aligned}
& \Delta \boldsymbol{w}_{+,c,r}^{(t)} \\
={}& \frac{\eta}{NP} \sum_{n=1}^{N} \mathbb{1}\{y_n = (+,c)\}[1 - \mathrm{logit}_+^{(t)}(\boldsymbol{X}_n^{(t)})] \\
& \times \sum_{p \in \mathcal{P}(\boldsymbol{X}_n^{(t)}; \boldsymbol{v}_+)} \mathbb{1}\{\langle \boldsymbol{w}_{+,c,r}^{(t)}, \alpha_{n,p}^{(t)} \boldsymbol{v}_+ + \boldsymbol{\zeta}_{n,p}^{(t)} \rangle + b_{+,c,r}^{(t)} > 0\} \left( \alpha_{n,p}^{(t)} \boldsymbol{v}_+ + \boldsymbol{\zeta}_{n,p}^{(t)} \right).
\end{aligned}
\tag{235}
$$

Indeed, these feature-detector neurons will continue growing in the direction of the features they are responsible for detecting instead of degrade in strength.

# H    Probability Lemmas

**Lemma H.1** (Laurent-Massart $\chi^2$ Concentration (Laurent & Massart (2000) Lemma 1)). *Let $\boldsymbol{g} \sim \mathcal{N}(\boldsymbol{0}, \boldsymbol{I}_d)$. For any vector $\boldsymbol{a} \in \mathbb{R}^d_{\geq 0}$, any $t > 0$, the following concentration inequality holds:*

$$\mathbb{P}\left[\sum_{i=1}^d a_i g_i^2 \geq \|\boldsymbol{a}\|_1 + 2\|\boldsymbol{a}\|_2\sqrt{t} + 2\|\boldsymbol{a}\|_\infty t\right] \leq e^{-t} \tag{236}$$

**Lemma H.2.** *Let $\boldsymbol{g} \sim \mathcal{N}(\boldsymbol{0}, \sigma^2 \boldsymbol{I}_d)$. Then,*

$$\mathbb{P}\left[\|\boldsymbol{g}\|_2^2 \geq 5\sigma^2 d\right] \leq e^{-d} \tag{237}$$

*Proof.* By Lemma H.1, setting $a_i = 1$ for all $i$ and $t = d$ yields

$$\mathbb{P}\left[\|\boldsymbol{g}\|_2^2 \geq \sigma^2 d + 2\sigma^2 d + 2\sigma^2 d\right] \leq e^{-d} \tag{238}$$

$\square$

**Lemma H.3** (Shen et al. (2022a)). *Let $\boldsymbol{g}_1 \sim \mathcal{N}(\boldsymbol{0}, \sigma_1^2 \boldsymbol{I}_d)$ and $\boldsymbol{g}_2 \sim \mathcal{N}(\boldsymbol{0}, \sigma_2^2 \boldsymbol{I}_d)$ be independent. Then, for any $\delta \in (0, 1)$ and sufficiently large $d$, there exist constants $c_1, c_2$ such that*

$$\mathbb{P}\left[|\langle \boldsymbol{g}_1, \boldsymbol{g}_2\rangle| \leq c_1 \sigma_1 \sigma_2 \sqrt{d \log(1/\delta)}\right] \geq 1 - \delta \tag{239}$$

$$\mathbb{P}\left[\langle \boldsymbol{g}_1, \boldsymbol{g}_2\rangle \geq c_2 \sigma_1 \sigma_2 \sqrt{d}\right] \geq \frac{1}{4} \tag{240}$$

