# OpenReview forum: "Why Fine-grained Labels in Pretraining Benefit Generalization?"
_TMLR — Accepted by TMLR_

### Review · Reviewer_kxTv · 2024-07-25

**Summary Of Contributions:**

This paper explains why the generalization performance of a DNN pre-trained using fine-grained labels is better than that of a DNN pre-trained using coarse-labeled data.

**Audience:**

Yes

**Broader Impact Concerns:**

This work does not require adding a Broader Impact Statement

**Claims And Evidence:**

No

**Requested Changes:**

Stated in Weakness.

**Strengths And Weaknesses:**

Strength:
1. This topic is very interesting.

Weakness:
1. Many definitions are not clarified clearly. For example, why and what is the meaning of two superclasses +1 and −1? Can authors take a vivid example to explain this? Moreover, in definition 4.1, what is the difference between "v_{+}" and "v_{-}?" In definition 4.2, what is the definition of "v_{y}?"

2. Why easy sample and hard sample can be defined through Definition 4.2? Any experimental results to support the definition of easy sample and hard sample?

3. This paper is not self-contained, since experimental results are reported in Appendix.

4. I suggest that authors can explain each definition/theorem immediately after introducing each definition/theorem.

---

> ### Author Response · Authors · 2024-07-26
> **Response to Reviewer kxTv**
>
> Thank you for the constructive feedback. Here is our response to your questions and comments.
>
> **Weakness**:
>
> **W1(i)**. “why and what is the meaning of the two superclasses $+1$ and $-1$”, “example to explain”?
>
> **A1(i)**. As described in the second paragraph of Section 4.1 (bottom of page 5), we consider two levels of label hierarchy for analytical tractability. The two superclasses $+1$ and $-1$ are at the root level of the hierarchy. The running example in this paper is the cat vs. dog problem, as discussed in Section 3.2 and illustrated in Figures 3 and 4: we can consider images of cats belonging to the superclass $+1$, and those of dogs belonging to the superclass $-1$.
>
> **W1(ii)**. “In definition 4.1, what is the difference between ‘$v_{+}$’ and ‘$v_{-}$’?"
>
> **A(ii)** The feature vector $v_+$ is the common feature of superclass $+1$, and $v_{-}$ is the common feature of superclass $-1$. As presented in Definitions 4.2 and 4.3, (easy) input samples belonging to superclass $+1$ contain patches dominated by $v_+$, and those of superclass $-1$ contain patches dominated by $v_-$. Please also refer to Figure 4 for a visual illustration of input samples, and how these definitions link to our intuition presented in Section 3.2 and Figure 3.
>
> **W1(iii)**. "In definition 4.2, what is the definition of $v_{y}$?”
>
> **A(iii)**. As stated in the first line of the definition, $y$ is either $+$ or $-$.
>
>
>
> **W2**. Justification for definition of easy and hard samples?
>
> **A2**. We follow our intuition presented in Section 3.2 to define easy and hard samples. The gist of it is that, given a target binary classification problem (e.g. cats vs. dogs), samples that contain discriminative patterns which are commonly observed in the two superclasses (e.g. the shape of cat's or dog's ears) should be easier to classify in comparison to samples which only contain fine-grained discriminative patterns (e.g. close-up shot of a cat's or dog's fur). This leads to the definition of the easy and hard samples in our theory: the easy samples contain both common and fine-grained features, while hard samples only contain fine-grained features. Additionally, we introduce some noise in the samples to simulate "distracting irrelevant patterns" which are often present in real images.
>
>
>
> **W3**. Experimental results in Appendix, not in main text. Explain definitions and theorems more immediately introducing them.
>
> **A3**. Thank you for the feedback. We do wish to highlight that this is a learning theory paper, therefore we dedicate the limited number of pages of the main text to explaining the framework of our theory and the core mathematical insights. We will do our best to revise and make the main text more self-contained and well-explained once all the reviews are received (as per TMLR's submission guideline).

---

### Review · Reviewer_zbHi · 2024-08-09

**Summary Of Contributions:**

The paper models and comes up with a theory for the effectiveness of pre-training on fine-grained labels. The intuitive argument (which is represented in their proofs) is that use of many (or an equal number of) subclass labels in pre-training ensures that an equal number of "detector" neurons for these subclass labels is produced during training. When only coarse labels are used, the common class labels are overrepresented.

The theoretical model considers a two-layer convolutional ReLU network and an orthonormal dictionary of feature vectors. The input is represented as a collection of patches and dataset modelling is considered as simple isotropic Gaussian noise on top of feature vectors. Hard test examples are generated by including feature-noise patches and "high-noise" patches, which serve as distracting patterns.

**Audience:**

Yes

**Broader Impact Concerns:**

N/A.

**Claims And Evidence:**

Yes

**Requested Changes:**

Overall, I am in favor of this article, as it seems to be a reasonable attempt at an analytical theory for explaining the effectiveness of fine-grained pretraining. The only thing I would ask for is further discussion of the weaknesses that I've noted above. If experimental evidence backing up the intuitive argument (as noted in point 3) was provided, I'd be willing to champion the article, I think.

**Strengths And Weaknesses:**

Strengths:
1. I found the writing in the main part of the text to be very clear. The authors clearly present the intuition behind the technical proofs in the appendices.
2. The model seems to be reasonably constructed and considered (though limited, necessarily; see below). The model allows for analysis, while being complicated enough to perhaps model the simplest relevant scenario for image classification.
3. I did not have time to check the theoretical proofs carefully, so I must give the benefit of the doubt on their correctness. At a quick glance the arguments seem to be well-structured and nontrivial.

Weaknesses:
1. There are many simplifying assumptions in the model that could limit the applicability of the analysis (and the authors admit as much), e.g., the model used is a simple two-layer convolutional network, and both classes and subclasses are represented with a single feature vector each. In reality, models are far more complicated (and would be harder to analytically characterize) and the features that represent a class may be a collection of local patch features in conjunction.
2. More of a question: The relationship between the common feature vectors and the subclass feature vectors is simply that of orthogonal unit vectors. This does not seem to capture any explicit hierarchical relationship between the common feature vectors and the subclass vectors, e.g., the common vectors being some linear combination of subclass vectors. Was any such modelling attempted, or can you give justification for the choice made here?
3. The empirical results simply show the effectiveness of pre-training with fine-grained labels, but do not aim to verify the intuitive argument. In particular, it'd be nice to try and show the presence of such "detector" neurons and to measure their approximate relative growth.
4. While the theoretical results are on a simple two-layer ReLU network, the experiments run are with a transformer vision model VitB.

---

> ### Author Response · Authors · 2024-09-03
> **Response to Reviewer zbHi**
>
> Thank you for your insightful comments and suggestions! Our response to your questions and comments is as follows.
>
> Weaknesses:
>
> **W1a & W4**. “ many simplifying assumptions in the model that could limit the applicability of the analysis”, “In reality, models are far more complicated … harder to analytically characterize… “, “While the theoretical results are on a simple two-layer ReLU network, the experiments run are with a transformer vision model VitB”
>
> As we discussed in the third Q&A in Section 6 of the main text, our theoretical setting is consistent with the existing literature on analyzing the features learnt by a neural network (and the process of feature learning): we all focus on understanding how two-layer nonlinear neural networks learn certain simple distributions, for the sake of analytical tractability. While our current analysis is confined to two layers, it is possible to extend it to deeper networks and our findings are expected to remain valid. However, it is important to note that the complexity of the analysis will increase considerably with additional layers.
>
>
> **W1b**. “features that represent a class may be a collection of local patch features in conjunction”
>
> Since the data model of this work is primarily driven by the goal of understanding the connections between features learnt when the label granularity varies (hence the importance of the _hierarchical_ multi-view property), we choose to keep the feature-label correspondence with a hierarchy level minimal for greater clarity of exposition. It is straightforward to extend the current analysis to the setting in which a collection of local patches in conjunction represents a “feature”, or the setting where a (sub-)class can have multiple features, etc.
>
> **W2**. “relationship between the common feature vectors and the subclass feature vectors… orthogonal unit vectors”, “does not seem to capture any explicit hierarchical relationship between the common feature vectors and the subclass vectors, e.g., the common vectors being some linear combination of subclass vectors”, “justification?”
>
> Our decision to model common and subclass feature vectors as orthogonal vectors is mainly driven by our intuition of the nature of these features in reality.
>
> Consider a very simple example: suppose the coarse-grained task is to classify “cars” vs. “pedestrians”, and the fine-grained subclasses are “Honda”, “BMW”, “Camry”, “Adults”, “Children”, etc. Let us think about the superclass of “cars”. One can easily distinguish them from “pedestrians” by recognizing _common features_ such as “wheels”, “front light” and so on. These common features are visually _very different_ from the _car brand logos_, which are _fine-grained features_ useful for knowing whether a car is a Honda, a BMW, or some other brand. Modeling the visual feature of a “wheel” as one which does not correlate much with the feature of a “BMW logo” seems _more_ reasonable to us than, for example, assuming that the common “wheel” feature is a linear combination of “Honda”, “BMW” and other car brand logos.
>
>
> **W3**. “ it'd be nice to try and show the presence of such "detector" neurons and to measure their approximate relative growth”
>
> Thank you for this comment. In practice, it is extremely difficult to know (1) what exactly is counted as a “feature” inside a deep neural network, and (2) what neurons are responsible for forming such features. In fact, single neurons simply might not be mono-semantic at all in practical DNNs  which solve real and challenging problems (such as ImageNet21k): meaningful features are not necessarily captured by individual neurons (or even small groups of neurons), since the number of discriminating “features” in the input distribution can be far greater than the number of neurons in any layer of the network. Although such hypotheses can be challenged or refuted with further evidence (although more formalized forms of this hypothesis already exists in the literature, for instance, see [1]), the complexity of the situation is apparent: the correspondence between “features” and neuron (groups) is a problem deserving an independent investigation, and beyond the scope of this learning theory paper. This paper’s more modest goal is to illustrate, in a analytically tractable scenario where the number of discriminative features _do not far exceed the number of neurons in the hidden layer of the network_, that there indeed exists an imbalance in feature learning inside the neural network induced by the label complexity.
>
> [1] Nelson Elhage, Tristan Hume, Catherine Olsson, Nicholas Schiefer, Tom Henighan, Shauna Kravec, Zac Hatfield-Dodds, Robert Lasenby, Dawn Drain, Carol Chen, et al. Toy models of superposition. arXiv preprint arXiv:2209.10652, 2022.

---

### Review · Reviewer_bHAS · 2024-08-21

**Summary Of Contributions:**

This paper provides a theoretical framework to explain why pretraining deep neural networks (DNNs) with fine-grained labels enhances generalization when fine-tuning on coarse-labeled downstream tasks. The authors introduce a "hierarchical multi-view" data model and prove that fine-grained pretraining enables DNNs to learn both common and rare features, leading to better accuracy on challenging test samples. They also show that higher label complexity in pretraining increases the complexity of learned representations, which is key to improved generalization. These theoretical insights are supported by large-scale experiments on datasets like ImageNet and iNaturalist, confirming the practical benefits of fine-grained pretraining.

**Audience:**

Yes

**Broader Impact Concerns:**

I do not have concerns regarding the broader impact of this paper.

**Claims And Evidence:**

Yes

**Requested Changes:**

See weaknesses.

**Strengths And Weaknesses:**

Strengths:
1. This paper studies an intriguing phenomenon on neural network generalization, where pretraining with fine-grained labels lead to better representations. The authors provided a theoretical analysis on explaining the reason behind this phenomenon.


2. The authors provided detailed background introduction for the theoretical framework and summarize the results in plain and useful insights.


Weaknesses:
1. The authors should have more discussions on the practical implications of the results. Given that pretraining on more labels are beneficial, should we be using this more often in practice? Since unsupervised pretraining is the standard of representation learning nowadays, can the results of supervised learning in this paper provide any insights in the current regime?

2. Are the learning dynamics the same between pretraining with coarse-grained labels and fine-grained labels? Given that the supervised tasks are different, is one necessarily harder than the others? Some analysis comparing the training dynamics of these two settings would be better.

3. The studied setting in this paper is that we should do pretraining with fine-grained labels and fine-tuning with coarse-grained labels. I am confused about the fine-tuning part. Why not always fine-tune using the same label space as pretraining? Is it necessarily inferior to fine-tuning with coarse-grained labels? I think the authors should have more discussions on this part or add clarifications on the settings where the claim holds.

4. Another relevant question would be about whether we can gradually interpolate from coarse-grained labels to fine-grained labels during pretraining, or the reversed way. Given the results presented in this paper, do the authors have any intuition about how one should do this interleaved style training? How does it compare to only training in either coarse-grained or fine-grained labels?

---

> ### Author Response · Authors · 2024-09-03
>
> Thank you for your valuable time and constructive feedback. The following is our response to your questions and comments.
>
> Weaknesses.
>
> **W1a**. “Practical implications of the results”. “Pretraining on more labels … using this more often in practice?”
>
> As we have discussed in Sections 1.1 and 2.3 of the paper, the practice of pretraining (large) vision models with a large label space is already quite common in supervised learning: we are mainly providing a theoretical justification for this common practice.
>
> **W1b**. “Unsupervised pretraining … results of supervised learning in this paper provide any insights…?”
>
> At its essence, supervised and unsupervised training of DNNs is about learning generalizing representations. In this regard, our proposed data model of hierarchical multi-view, and our theoretical analysis technique which establishes a connection between label and representation complexity, should be transferable to the unsupervised regime.
>
>
> **W2a**. “Are the learning dynamics the same between pretraining with coarse-grained labels and fine-grained labels?”, “some analysis comparing the training dynamics… would be better”
>
> The learning dynamics of the two tasks are different. We discuss the learning dynamics of the model in detail in Section 5.2 in the main text, and the corresponding Appendix sections C to G. The major difference between the two, at a very high level, lies in the complexity of the representations developed during learning: finer-grained pretraining tends to induce more complex representations of the underlying input data distribution, thus leading to better generalization.
>
> **W2b**.  “... is one necessarily harder than the others?”
>
> We wish to clarify that the focus of the analysis is not about comparing the hardness of the two tasks, but about showing why a neural network which is pretrained with fine-grained labels (and finetuned on the coarse-grained task) tends to generalize better on the coarse-grained task than one trained purely on that task.
>
>
> **W3**. “Why not always fine-tune using the same label space as pretraining?”, “clarification”
>
> As we discussed in Sections 1 and 2.3 of this paper, pretraining with highly fine-grained labels and finetuning on a downstream task that has lower label granularity is already commonly adopted in practice. We are mainly providing theoretical justifications for such practice.
>
>
> **W4**. “ interpolate from coarse-grained labels to fine-grained labels during pretraining, or the reversed way”, “ …interleaved style training?”, “...compare to only training in either coarse-grained or fine-grained labels?”
>
> Thank you for this interesting question.
>
> Outside of the idealized setting in our theory, some form of interleaved training with coarse and fine-grained labels (if we have access to such label hierarchy) might indeed be more beneficial to generalization than training on a fixed granularity. We consider two factors here:
>
> 1. In practice, higher granularity does not always imply better generalization: the more classes we have, the more information the model is presented with, but also the fewer samples per class. As discussed in Section 6, in the extreme case of one unique class per training sample, the DNN is not going to learn much meaningful information.
>
> 2. In practice, it can be hard to know the exact pretraining granularity that is the most suitable for a downstream task.
> Therefore, some interleaved/adaptive training which prevents sample starvation while still offering sufficiently detailed information about the image distribution could be ideal.

---

### Comment · Action_Editor_W7oP · 2024-10-13
**One minor change**

Dear authors,

Thank you very much for addressing the comment, I've accepted the changes. One final small request: could you please fix the reference to iNaturalist dataset?

Many thanks,
The Action Editor

---

> ### Author Response · Authors · 2024-10-23
> **Final revision**
>
> Dear Action Editor,
>
> Thank you for accepting the changes! We cannot upload a new revision of our paper now that the "Camera Ready Revision" button is gone. Is there another place for us to upload such revisions?
>
> Thanks,
>
> Authors of paper 2978

---

### Decision · Action_Editor_W7oP · 2024-09-07

**Recommendation:** Accept with minor revision

**Comment:**

I suggest the paper should be accepted with minor revision. This is because of the outstanding concern that the paper is not self-contained as the experimental results are entirely in the Appendix. Also, it would be important that the authors emphasise in the text how the provided empirical setting reflects the assumptions of the theoretical analysis (Q1 and Q4 of Reviewer zbHi).

**Audience:**

All reviewers agreed that the manuscript has audience.

**Claims And Evidence:**

The paper asks simple but interesting question: what are the theoretical reasons for improvement of the generalisation performance when pretraining using fine-grained data labels as opposed to coarse-grained ones. The authors provide theoretical argument and, in the appendix, the experimental studies. The articles claims receive mostly positive feedback, as the rebuttal addresses most of the concerns about the experimental evidence.